# DISENTANGLE, GATE, AND OPTIMIZE: CROSS DOMAIN TRANSFER POWER BY MULTI OBJECTIVE BAYESIAN OPTIMIZATION

## ABSTRACT

Prompt Tuning (PT) has recently shown remarkable success in diverse Natural Language Processing (NLP) tasks, providing an efficient knowledge transfer paradigm to textually instruct models with domain-level guidance. However, existing PT approaches often struggle to accurately distinguish between domain-invariant and domain-specific knowledge of input texts, thereby inducing negative transfer that harms model performances across various domains. To mitigate this, recent studies have introduced the concept of adversarial training to highlight domain-specific nuances for improving the model's adaptation ability, but often rely on overly complex parameter optimization, which hinders smooth generalization. Motivated by this, we propose a novel prefix tuning framework, named **A**daptive **R**obust **P**refix **O**ptimization (**ARPO**), in which adaptive representation disentanglement precisely decouples domain-specific information from invariant knowledge, while Multi-Objective Bayesian Optimization (MOBO) dynamically adjusts adversarial strategies for improved model robustness. Specifically, we first develop disentangled representation learning based on Information Bottleneck theory with dynamic orthogonality and conditional independence constraints, combined with adaptive adversarial training driven by dynamic thresholds. We then employ MOBO for efficient search within the high-dimensional strategy space. We theoretically prove that the proposed MOBO approach is feasible and guaranteed to converge under reasonable assumptions. Extensive evaluations on GLUE, Super GLUE, MRQA 2019, GSM8K, and HumanEval show that ARPO achieves around 6% improvement in two experimental settings, highlighting its robust cross-domain generalization.

## 1 INTRODUCTION

Large Language Models (LLMs) have significantly advanced natural language processing Radford et al. (2019); Raffel et al. (2020); Brown et al. (2020). Despite these successes, efficiently adapting LLMs across diverse tasks and domains remains challenging, particularly when computational resources are limited Houlsby et al. (2019); Lester et al. (2021); Hu et al. (2021). Prompt Tuning (PT) partially alleviates this issue by selectively updating only a small subset of parameters, facilitating efficient cross-domain adaptation Houlsby et al. (2019); Lester et al. (2021); Hu et al. (2021); Zaken et al. (2021). In particular, PT methods often fail to accurately separate domain-invariant from domain-specific information, leading to negative transfer and reduced generalization. Although recent adversarial training schemes have been proposed to address these issues, their task-oriented nature significantly limits the ability of LLMs to adapt stably across diverse learning scenarios. Moreover, traditional hyperparameter optimization approaches Laumanns & Ocenasek (2002) are insufficiently efficient in managing complex multi-objective optimization problems, further limiting the performance of PT in practical cross-domain applications.

Previous research in PT (e.g., Adapters Houlsby et al. (2019), LoRA Hu et al. (2021), and SPoT Vu et al. (2021)) mainly focused on reducing computational overhead by training only a subset of model parameters. However, these methods often fail to adequately handle substantial semantic and lexical domain variations, resulting in negative transfer and limited generalization. Meanwhile, methods employing disentangled representation learning(e.g., DVIB Bao (2021) and DisTIB Dang et al. (2024)) attempt to address these issues by isolating domain-specific and invariant features, yet

they lack robust adaptive constraints, causing representation redundancy. Moreover, although Multi-Objective Bayesian Optimization (MOBO)(e.g., EVHI Daulton et al. (2020a), MBO Suzuki et al. (2020)) effectively optimize multiple goals, it struggles with complex, high-dimensional parameter spaces common in cross-domain scenarios. These limitations highlight the need for adaptive, efficient optimization strategies tailored explicitly for robust cross-domain LLM adaptation.

To this end, we propose **Adaptive Robust Prefix Optimization (ARPO)**, a cross-domain prefix-tuning approach that integrates adaptive representation disentanglement with multi-objective Bayesian optimization (MOBO). Specifically, we construct a dynamically adaptive prefix disentanglement framework by combining Information Bottleneck (IB) theory Tishby et al. (2000) with orthogonality and conditional independence constraints, effectively separating domain-invariant from domain-specific knowledge. Furthermore, we also present a dynamic threshold-controlled adversarial training method and use MOBO to automatically search and optimize a high-dimensional mixed discrete-continuous adversarial strategy space, boosting model generalization and robustness.

Extensive experiments including GLUE Wang et al. (2018), SuperGLUE Wang et al. (2019), and MRQA 2019 Fisch et al. (2019) as well as reasoning and coding benchmarks GSM8K Cobbe et al. (2021) and HumanEval Chen et al. (2021), together with robustness and ablation analyses, demonstrate the effectiveness of our approach. Our main contributions are summarized as follows:

(1) We propose a clear **domain-information disentanglement strategy** using Information Bottleneck constraints, effectively mitigating negative transfer caused by domain-specific features;

(2) We design a **dynamic threshold-based adversarial gate mechanism** to prevent premature interference in primary task training, significantly improving training stability;

(3) We introduce a **MOBO-based global decision maker** that uses noisy-$q$EHVI on a mixed discrete–continuous, gate-aware space to jointly tune perturbation structure, strength, and timing, directly maximizing the accuracy–robustness–cost Pareto hypervolume and replacing ad-hoc heuristics with a transferable, interpretable, sample-efficient strategy.

## 2  RELATED WORKS

**Domain Adaptation and Prompt Tuning.** Traditional domain adaptation methods typically fine-tune all parameters Radford et al. (2021); Touvron et al. (2023), incurring computational overhead and overfitting risks Han et al. (2024); Zaken et al. (2021). Prompt Tuning (PT) strategies, such as Adapters Houlsby et al. (2019), LoRA Hu et al. (2021), and AdapterDrop Rücklé et al. (2020), alleviate resource consumption but can introduce latency or fail under large semantic shifts Houlsby et al. (2019); Zhong et al. (2021). Further parameter-efficient PT methods Lester et al. (2021); Liu et al. (2021c) remain vulnerable to vocabulary and input perturbations Ma et al. (2022). Multi-part decomposition techniques Vu et al. (2021); Asai et al. (2022) and DePT Shi & Lipani (2023) provide modularity but lack systematic cross-domain reuse. Differently, our solution reuses modular prompts to address semantic diversity and computational efficiency.

**Adversarial Training and Multi-Domain Robustness.** Adversarial training significantly enhances NLP model robustness Miyato et al. (2018), but embedding-level methods struggle against discrete textual perturbations. Token-aware adversarial training partially mitigates this Li & Qiu (2021), though at increased complexity. Recent advancements, including contrastive learning Rim et al. (2021), curriculum methods Yoo & Qi (2021), Adversarial Distributional Training (ADT) Dong et al. (2020), aim for improved generalization. Adversarial Self-Training (AST) Shi & Liu (2023) specifically applies adversarial techniques for domain adaptation tasks. Our proposed method innovatively integrates adversarial training within domain-specific prompt tuning, significantly improving cross-domain robustness with minimal computational overhead.

**Multi-Objective Bayesian Optimization.** Efficiently balancing conflicting objectives, such as accuracy and computation, is crucial in hyperparameter optimization Snoek et al. (2012); Shahriari et al. (2015). Bayesian Optimization (BO) with Gaussian Processes provides systematic exploration and sample efficiency Frazier (2018); Jin et al. (2018); Eriksson et al. (2019). Recent Multi-Objective Bayesian Optimization (MOBO) methods utilize Expected Hypervolume Improvement (EHVI) Emmerich et al. (2011); Daulton et al. (2020a) and adapt to mixed variables Ru et al. (2020). However, existing MOBO primarily targets hyperparameter selection rather than systematic prompt or adapter

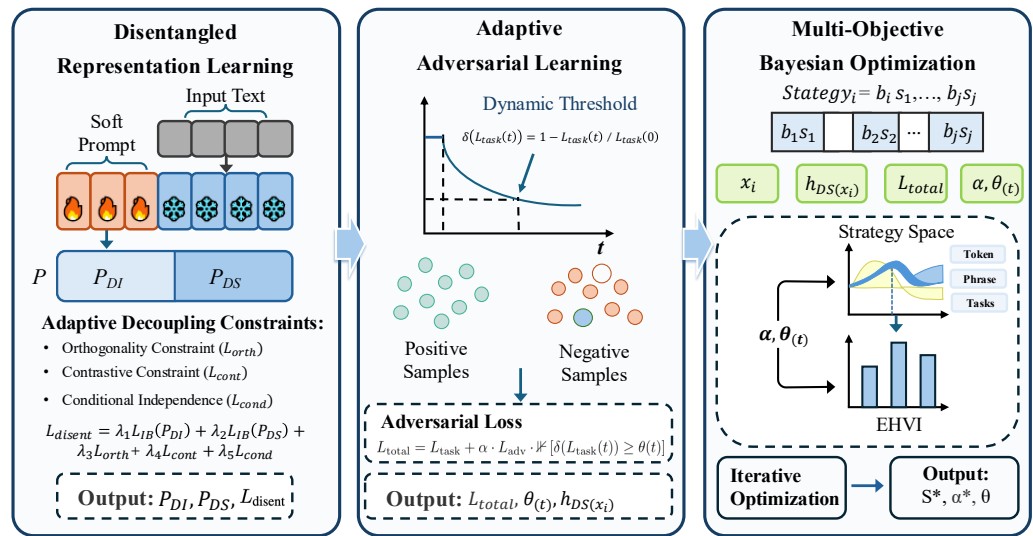

Figure 1: The framework of **ARPO**. The model learns disentangled prefix representations $P_{DI}$ and $P_{DS}$ via adaptive information bottleneck and multi-level constraints, selectively applies adversarial adaptation with dynamic thresholds $\theta(t)$, and efficiently searches optimal adversarial strategies $S^*$ using MOBO for robust cross-domain adaptation.

reuse Lester et al. (2021); Rücklé et al. (2020). Our novel MOBO method reuses effective prompts to provide strong adaptation and efficient generalization across many NLP domains.

## 3 METHODOLOGY

In this section, we introduce ARPO (Figure 1, Algorithm 1), a three-part method for robust cross-domain transfer. An information-bottleneck prefix module (Sec 3.1) disentangles representations into $P_{DI}$ (domain-invariant) and $P_{DS}$ (domain-specific), producing clean features for transfer. A dynamic gate (Sec 3.2) applies adversarial signals only when training is stable, improving robustness without slowing convergence. A MOBO module (Sec 3.3) with noisy-qEHVI jointly tunes perturbation structure, strength, and trigger timing to balance accuracy, robustness, and compute. Appendix B.1 provides notation, problem setup, and assumptions, Appendix B.2 gives the soft prompt rationale and implementation, Appendix C develops the theoretical and implementation details for adaptive prefix disentanglement, Appendix D introduces task-aware gating, EMA bounds, spectral regularization, and Appendix E details the MOBO objectives, surrogate, and EHVI.

### 3.1 ADAPTIVE REPRESENTATION LEARNING OF DISENTANGLED PREFIX

In prefix tuning, the prefix $P$ shapes the conditional representation $h(x; P)$. Domain-invariant (DI) knowledge is the part that does not change across domains $D$ and remains predictive of labels $Y$. We target $I(P_{DI}; D) \approx 0$ and large $I(P_{DI}; Y)$. Domain-specific (DS) knowledge is the part that encodes domain traits such as style, vocabulary, format, and noise. We target large $I(P_{DS}; D)$ and keep $I(P_{DS}; Y)$ moderate so it does not dominate the task signal.

We split the learnable prefix $P \in \mathbb{R}^{L \times d}$ into domain-invariant and domain-specific segments, $P_{DI} \in \mathbb{R}^{L_{DI} \times d}$ and $P_{DS} \in \mathbb{R}^{L_{DS} \times d}$ with $L = L_{DI} + L_{DS}$. Domain-invariant knowledge should be predictive across domains while carrying minimal domain cues, whereas domain-specific knowledge should capture domain traits without overwhelming the task signal. We therefore optimize two information-bottleneck objectives

$$\mathcal{L}_{IB}(P_{DI}) = I(P_{DI}; \text{Domain}) - \beta_1 I(P_{DI}; \text{Task}),$$
$$\mathcal{L}_{IB}(P_{DS}) = -I(P_{DS}; \text{Domain}) + \beta_2 I(P_{DS}; \text{Task}). \tag{1}$$

Mutual informations are estimated by a neural estimator with stabilization (Appendix C), which turns information-theoretic targets into practical training losses.

To geometrically separate the two subspaces, we penalize cross-subspace overlap via an orthogonality term

$$\mathcal{L}_{orth} = \|P_{DI}^{\top} P_{DS}\|_F^2, \tag{2}$$

and further reduce residual dependence by a HSIC-based penalty (Appendix C.2. Discriminative decoupling is encouraged by a contrastive loss that pulls together $P_{DI}$ for the same task across domains and pushes apart $P_{DS}$,

$$\mathcal{L}_{cont} = -\log \frac{\exp(\mathrm{sim}(P_{DI}^i, P_{DI}^j)/\tau)}{\sum_k \exp(\mathrm{sim}(P_{DI}^i, P_{DI}^k)/\tau)} - \log \frac{\exp(\mathrm{sim}(P_{DS}^i, P_{DS}^j)/\tau)}{\sum_k \exp(\mathrm{sim}(P_{DS}^i, P_{DS}^k)/\tau)}, \tag{3}$$

with temperature adaptation and quantile pairing detailed in Appendix C.3. We explicitly control redundant information flow by a conditional-independence term

$$\mathcal{L}_{cond} = D_{KL}\Big(p(P_{DI}, P_{DS} \mid \mathrm{Task}) \,\big\|\, p(P_{DI} \mid \mathrm{Task})\, p(P_{DS} \mid \mathrm{Task})\Big), \tag{4}$$

whose variational estimator and consistency are given in Appendix C.4.

The overall disentanglement loss is

$$\mathcal{L}_{disent} = \lambda_1 \mathcal{L}_{IB}(P_{DI}) + \lambda_2 \mathcal{L}_{IB}(P_{DS}) + \lambda_3 \mathcal{L}_{orth} + \lambda_4 \mathcal{L}_{cont} + \lambda_5 \mathcal{L}_{cond}. \tag{5}$$

This combination aligns information-theoretic, geometric, and statistical criteria so that task-relevant transferable factors concentrate in $P_{DI}$ and domain cues in $P_{DS}$. Under mild boundedness and local-Lipschitz assumptions, the expected gradient directions of these terms are compatible rather than antagonistic, improving separability without sacrificing task signal; the statement and supporting inequalities are summarized in Appendix C.5. The resulting encoder $[P_{DI}; P_{DS}]$ becomes the common input for the gated adversarial module and the MOBO controller, enabling safe perturbation and strategy search downstream.

## 3.2 Adversarial Adaptation of Cross-domain Knowledge

Motivated by the risk that adversarial updates can harm convergence when applied too early after Section 3.1, and by heterogeneous convergence across tasks and domains, we design a dynamic gate for adversarial training. The gate combines three signals, main loss progress, gradient stability, and task difficulty, and activates only when the score exceeds a threshold. When active, we shape the $P_{DS}$ subspace with distance based positive and negative pairs. The loss pulls same domain pairs together and pushes cross domain pairs apart, trained jointly with the main and disentanglement losses. We couple the gate with MOBO to auto tune thresholds and adversarial strength.

### 3.2.1 Dynamic Threshold Determination

We measure training progress using the relative improvement of the main task loss

$$\delta(\mathcal{L}_{\mathrm{task}}(t)) = 1 - \mathcal{L}_{\mathrm{task}}(t)/\mathcal{L}_{\mathrm{task}}(0) \in [0, 1], \tag{6}$$

where larger values indicate safer timing for adversarial updates. This aligns with Curriculum Learning, ensuring foundational task mastery before introducing increased complexity. However, a fixed threshold is too rigid for multi-task training. Therefore, we define a time-varying base threshold,

$$\theta(t) = \theta_0 \cdot \left(1 - \exp\left(-\gamma \cdot \frac{\mathrm{var}(\nabla \mathcal{L}_{\mathrm{task}}(t-w))}{\mathrm{mean}(|\nabla \mathcal{L}_{\mathrm{task}}(t-w)|)}\right)\right), \tag{7}$$

computed over a short window $w$: when recent gradients are volatile, $\theta(t)$ increases to postpone activation; when gradients stabilize, $\theta(t)$ decreases to allow earlier activation. Estimator choices (EMA vs. fixed window), outlier handling, trigger properties, and per-task scaling by difficulty are summarized in Appendix D.1. The Appendix D.2 for the gate's sublinear flip count in $T$ under EMA-driven sub-Gaussian increments, supporting a locally slowly-varying objective between flips.

### 3.2.2 Adaptive Adversarial Training with Dynamic Thresholding

To enhance stability and cross-domain generalization, we propose adaptive adversarial training using a dynamic threshold. Specifically, after encoding inputs $x_i$ via the T5 encoder to obtain hidden

representations $\mathbf{h}_i$, we isolate domain-specific features $\mathbf{h}_{DS}(x_i)$. Without proper constraints, these domain-specific representations may overlap, reducing discriminability across domains.

To address this, we dynamically generate positive and negative sample pairs in each training batch by calculating pairwise Euclidean distances: $d(\mathbf{h}_{DS}(x_i), \mathbf{h}_{DS}(x_j)) = \|\mathbf{h}_{DS}(x_i) - \mathbf{h}_{DS}(x_j)\|_2$. Sample pairs with distances below a predefined threshold are grouped as positives (Pos), reflecting similar domain attributes, while those exceeding this threshold form the negatives (Neg). This adaptive pairing method captures modest intra- and inter-domain differences and accurately supervises this; its batchwise quantile implementation and finite-sample guarantees follow from DKW–Massart bounds, detailed in Appendix D.3.

Leveraging these dynamically constructed pairs, we design an adversarial loss inspired by contrastive learning principles to effectively enhance domain discriminability within the representation space. Formally, the adversarial loss is

$$\mathcal{L}_{\text{adv}} = \sum_{(i,j)\in\text{Pos}} d(\mathbf{h}_{DS}(x_i), \mathbf{h}_{DS}(x_j)) - \sum_{(i,k)\in\text{Neg}} d(\mathbf{h}_{DS}(x_i), \mathbf{h}_{DS}(x_k)), \tag{8}$$

which encourages domain-specific representations of similar samples to cluster while pushing apart dissimilar domains.

To balance the primary task objective with adversarial optimization effectively, we integrate these components into a unified loss function modulated by a dynamic thresholding strategy:

$$\mathcal{L}_{\text{total}} = \mathcal{L}_{\text{task}} + \alpha \cdot \mathcal{L}_{\text{adv}} \cdot \mathbb{K}\left[\delta(\mathcal{L}_{\text{task}}(t)) \geq \theta(t)\right], \tag{9}$$

where $\mathcal{L}_{\text{task}}$ is the primary task loss, $\alpha \in [0, 1]$ is a weighting parameter, and the indicator enables adversarial training selectively. Adversarial optimization activates only when the current improvement $\delta(\mathcal{L}_{\text{task}}(t))$ surpasses the adaptive threshold $\theta(t)$; otherwise, the model optimizes solely toward the primary task, ensuring stable foundations before adding adversarial complexity across domains.

## 3.3 ADAPTIVE MULTI-BAYESIAN ADVERSARIAL STRATEGIES

MOBO acts as the global controller in ARPO, jointly choosing structure (binary switches $b_j$), strength/frequency (continuous magnitudes $s_j$ and global weight $\alpha$), and schedule (gate threshold $\theta$). This controller is necessary because the decision space is mixed discrete–continuous with temporal coupling and the objectives $[\text{Acc}, \text{Robust}, -\text{Cost}]$ are inherently conflicting, making heuristic or single-objective tuning inadequate. MOBO maps validation feedback through GP surrogates and EHVI to propose the next $(\alpha, \theta, \text{Strategy})$, which perturbs inputs encoded by $[P_{DI}; P_{DS}]$ and interfaces with the dynamic gate. The resulting train–validate–acquire loop advances the Pareto front in a sample-efficient manner and supplies updated surrogates for subsequent proposals.

### 3.3.1 CONSTRUCTION AND AUTOMATIC ASSEMBLY OF ADVERSARIAL STRATEGY SPACE

We build a hierarchical library of parameterized atomic operations across task, phrase, and token levels, $\{\mathcal{O}_1, \ldots, \mathcal{O}_m\}$. Task-level operators (e.g., cross-domain task swaps) modify global supervision signals, phrase-level operators (e.g., Phrase-Swap-A/B) perturb local semantic spans, and token-level operators (e.g., Token-FGSM/Token-PGD) adjust embeddings along gradient-guided directions.The multi-granularity design reveals complementary inductive biases from coarse to fine scales, engaging with the prefixed representation $[P_{DI}; P_{DS}]$ by simultaneously regularizing domain-invariant transfer (via stable cues to $P_{DI}$) and enhancing domain-specific separability (via discriminative cues to $P_{DS}$).

Each atomic operation $\mathcal{O}_j$ is governed by a binary activation $b_j \in \{0, 1\}$ and a continuous strength $s_j \in \mathbb{R}$, and a concrete strategy is encoded as $\text{Strategy} = \{(b_1, s_1), \ldots, (b_m, s_m)\}$. During training, the assembled perturbation acts on inputs by superposing the atomic effects:

$$\mathcal{A}(x; \text{Strategy}) = \sum_{j=1}^{m} b_j \, s_j \, \mathcal{O}_j(x), \tag{10}$$

so only active operators contribute and their magnitudes scale with the corresponding strengths. This operator $\mathcal{A}(\cdot; \text{Strategy})$ is applied to inputs already encoded with $[P_{DI}; P_{DS}]$, before task and disentanglement losses are evaluated.

The decision vector encodes structure, strength, and schedule as

$$x = \big(x_{\text{struct}}, \alpha, \theta\big), \qquad x_{\text{struct}} = \{(b_j, s_j)\}_{j=1}^m, \tag{11}$$

where $\alpha$ is the global adversarial weight and $\theta$ the dynamic-gate threshold. The resulting search space combines discrete choices with continuous parameters and is optimized via MOBO in Sec. 3.3.2.

We fit Gaussian-process surrogates with a product kernel that respects both continuous magnitudes and discrete structure:

$$k(x, x') = k_{\text{cont}}(z, z') \cdot k_{\text{disc}}(u, u'), \tag{12}$$

where $z = (s_1, \ldots, s_m, \alpha, \theta)$ and $u = b \in \{0, 1\}^m$. For the continuous part, $k_{\text{cont}}$ is RBF or Matérn; for the discrete part we use a Hamming-based categorical kernel

$$k_{\text{disc}}(u, u') = \exp\big(-\lambda \operatorname{Ham}(u, u')\big), \tag{13}$$

with a $\delta$-kernel as an alternative when a strictly categorical metric is required. Equations 12 and 13 induce an interpretable similarity: strategies differing in more activation bits are farther apart than those differing only in strengths, enabling Lipschitz-style regularity checks on the product domain.(Appendix E.5) Together with the assembly operator in equation 10, this completes the reduction from hierarchical perturbations to a GP-ready decision space used in Sec. 3.3.2.

### 3.3.2 Adaptive Strategy Search via Multi-Objective Bayesian Optimization

At iteration $t$, ARPO follows a train$\to$validate$\to$acquire loop: using the current decision $x = (x_{\text{struct}}, \alpha, \theta)$, a full training round is executed under the gate schedule $\theta$; validation then yields noisy observations $y_j = f_j(x) + \varepsilon_j$ with $\varepsilon_j \sim \mathcal{N}(0, \sigma_{n,j}^2)$ for $\mathbf{f}(x) = [\operatorname{Acc}(x), \operatorname{Robust}(x), -\operatorname{Cost}(x)]$. These observations update Gaussian–process (GP) surrogates over the mixed decision space, which reuse the product kernel defined in Sec. 3.3.1. Full GP basics and multi-objective extensions are deferred to Appendix E.1.

For each objective $j \in \{\operatorname{Acc}, \operatorname{Robust}, -\operatorname{Cost}\}$, the GP posterior mean at any candidate $x$ is

$$\mu_{j,t}(x) = m_j(x) + k_j(x, X_t)\big[K_j(X_t, X_t) + \sigma_{n,j}^2 I\big]^{-1}\big(\mathbf{y}_j - m_j(X_t)\big), \tag{14}$$

and the posterior variance $\sigma_{j,t}^2(x)$ follows the standard closed form. $\mu_{j,t}(x)$ predicts the expected validation outcome after training under decision $x$, while $\sigma_{j,t}^2(x)$ measures epistemic uncertainty from limited evaluations in the mixed discrete–continuous space in ARPO. The summaries balance exploitation ($\mu_{j,t}$) and exploration ($\sigma_{j,t}^2$) to drive acquisition.

Candidates are scored via hypervolume-based acquisition with respect to a reference point $\mathbf{r}_t \in \mathbb{R}^3$. Let $\operatorname{ND}(\cdot)$ return the non-dominated set. The batch of size $q$ is chosen by maximizing expected hypervolume improvement (EHVI) under the joint GP posterior $\mathcal{P}_t$:

$$X_{t+1} = \arg \max_{X \subset \Omega, \, |X|=q} \operatorname{EHVI}\big(X \mid \mathcal{D}_t, \mathbf{r}_t\big), \tag{15}$$

where we use noisy-$q$EHVI with Monte Carlo and common random numbers for variance reduction. The reference point is updated monotonically by componentwise 5th percentiles to stabilize estimates:

$$\mathbf{r}_{t+1} = \min\Big(\mathbf{r}_t, \operatorname{Percentile}_{5\%}\big(\{\mathbf{f}(x^{(i)})\}_{i \le n_{t+1}}\big)\Big). \tag{16}$$

Implementation details, including the hypervolume definition and Monte Carlo estimators, are provided in Appendix E.2–E.3, and differentiability/gradient schemes in Appendix E.4.

The selected batch $X_{t+1}$ is decoded to $(\alpha, \theta, \operatorname{Strategy})$ and executed under the gate; validation outcomes are appended to $\mathcal{D}_{t+1}$, GP surrogates are refit, and the acquisition in Eq. equation 15 is re-optimized to propose the next batch. This closes a gate-aware MOBO loop that combines a product-kernel GP surrogate with noisy-$q$EHVI to directly advance Pareto hypervolume; hypervolume consistency under standard GP regularity and slow variation holds as shown in Appendix E.5.

**Putting It All Together.** Iterating the pipeline yields a closed-loop adaptation scheme: Sec. 3.1 disentangles $[P_{DI}; P_{DS}]$ with information-theoretic regularizers grounded by variational MI estimation and conditional KL (Appendix C, C.4) and stabilized independence control via HSIC (Appendix C.2); Sec. 3.2 deploys a gate that injects adversarial signals only under measured stability, aligning with

the multi-loss synergy guarantees (Appendix C.5); Sec. 3.3 runs MOBO over the mixed decision space with product-kernel GPs and noisy-$q$EHVI to propose $(\alpha, \theta, \text{Strategy})$, with acquisition, differentiability, and convergence analyses in Appendices E.2–E.5. The result is a principled, data-driven controller that harmonizes disentanglement, safe adversarial adaptation, and multi-objective search; Sec. 4 demonstrates consistent Pareto improvements in accuracy–robustness–compute, while the appendices provide the theoretical scaffolding that underwrites these empirical gains.

# 4 EXPERIMENTS

In this section, we present experiments that validate our approach. We evaluate cross-domain performance on standard NLP benchmarks and compare against strong baselines. Appendix B.1 summarizes the preliminaries, and Appendix B.3 details the experimental setup and the datasets.

## 4.1 MAIN RESULTS

| Method | NQ→SQA | | Yelp→SciTail | | News→HP | | MNLI→QQP | | CoLA→PAWS | | Mean | |
|---|---|---|---|---|---|---|---|---|---|---|---|---|
| | Q | L | Q | L | Q | L | Q | L | Q | L | Q | L |
| **Fine-tuning** | 73.2 | 76.2 | 91.5 | 97.2 | 69.8 | 71.8 | 79.5 | 83.6 | 70.9 | 73.3 | 77.0 | 80.1 |
| **Adapter** | 76.8 | 78.8 | 93.3 | 96.9 | 70.3 | 71.9 | 82.0 | 85.0 | 72.8 | 74.5 | 79.0 | 81.3 |
| **BitFit** | 72.9 | 76.9 | 88.1 | 90.1 | 71.3 | 73.2 | 80.3 | 87.1 | 72.3 | 74.3 | 76.9 | 80.7 |
| **PT-2** | 70.1 | 76.1 | 91.4 | 93.1 | 72.8 | 75.5 | 81.4 | 84.9 | 70.7 | 74.1 | 77.3 | 81.1 |
| **SPoT** | 70.2 | 75.2 | 89.8 | 92.7 | 72.0 | 74.5 | 83.1 | 87.1 | 71.9 | 74.8 | 77.4 | 81.3 |
| **ATTEMPT** | 74.2 | 79.4 | 90.6 | 93.7 | 73.2 | 75.2 | 83.3 | 87.3 | 72.9 | 75.2 | 78.8 | 82.6 |
| **XPROMPT** | 75.2 | 78.2 | 90.3 | 93.3 | 75.4 | 76.1 | 84.4 | 89.3 | 73.8 | 75.8 | 79.8 | 82.9 |
| **InfoPrompt** | 76.7 | 80.2 | 91.5 | 94.5 | 74.1 | 75.8 | 85.8 | 90.5 | 75.2 | 76.1 | 80.7 | 83.8 |
| **DEPT** | 77.4 | 80.3 | 93.1 | 96.5 | 74.5 | 78.2 | 86.8 | 91.3 | 75.1 | 77.0 | 81.4 | 85.0 |
| **Udapter** | 77.9 | 80.9 | 93.6 | 96.7 | 74.8 | 78.5 | 87.2 | 91.7 | 75.4 | 77.3 | 81.8 | 85.4 |
| **DAdEE** | 78.5 | 81.5 | 93.9 | 96.8 | 75.1 | 78.9 | 87.8 | 92.3 | 76.1 | 77.9 | 82.3 | 85.9 |
| **ARPO** | **80.5** | **84.5** | **96.7** | **98.2** | **77.5** | **80.8** | **89.8** | **94.5** | **77.5** | **80.4** | **84.4** | **88.0** |

Table 1: Comparison of methods evaluated on cross-domain transfer tasks using the Qwen3-4B (Q) and LLama2-7B (L) models. Mean accuracy across all tasks is also reported.

| Method | CoLA→QQP | | GSM8K→HEval | | BoolQ→NQ | | HP→SciTail | | MNLI→SQA | | Mean | |
|---|---|---|---|---|---|---|---|---|---|---|---|---|
| | Q | L | Q | L | Q | L | Q | L | Q | L | Q | L |
| **Fine-tuning** | 66.0 | 68.9 | 12.5 | 18.0 | 70.5 | 73.4 | 71.5 | 74.6 | 73.0 | 76.4 | 58.7 | 62.3 |
| **Adapter** | 68.8 | 70.4 | 16.8 | 24.5 | 72.2 | 74.6 | 73.9 | 76.1 | 74.1 | 78.3 | 61.2 | 64.8 |
| **BitFit** | 67.6 | 69.8 | 14.0 | 20.5 | 71.4 | 74.9 | 73.2 | 75.8 | 74.0 | 78.1 | 60.0 | 63.8 |
| **PT-2** | 67.2 | 68.9 | 16.2 | 24.0 | 71.6 | 75.0 | 73.6 | 76.0 | 75.1 | 78.0 | 60.7 | 64.4 |
| **SPoT** | 67.9 | 69.1 | 18.0 | 26.5 | 72.5 | 74.8 | 75.4 | 77.8 | 77.0 | 79.6 | 62.2 | 65.6 |
| **ATTEMPT** | 68.4 | 71.0 | 19.5 | 28.8 | 73.0 | 75.2 | 76.0 | 78.6 | 77.5 | 80.5 | 62.9 | 66.8 |
| **XPROMPT** | 69.1 | 72.1 | 20.2 | 29.6 | 73.5 | 75.8 | 76.6 | 79.1 | 78.2 | 81.0 | 63.5 | 67.5 |
| **InfoPrompt** | 69.6 | 72.5 | 21.0 | 30.5 | 73.9 | 76.5 | 77.0 | 80.1 | 78.8 | 81.7 | 64.1 | 68.3 |
| **DEPT** | 70.5 | 74.3 | 22.5 | 31.0 | 73.0 | 75.1 | 77.6 | 80.8 | 79.1 | 82.7 | 64.5 | 68.8 |
| **Udapter** | 72.4 | 75.1 | 24.5 | 33.8 | 74.6 | 76.9 | 81.5 | 85.2 | 76.0 | 78.9 | 65.8 | 70.0 |
| **DAdEE** | 73.1 | 75.7 | 25.3 | 34.6 | 75.2 | 77.5 | 82.1 | 85.8 | 76.6 | 79.6 | 66.5 | 70.6 |
| **ARPO** | **74.6** | **77.9** | **32.1** | **40.9** | **77.1** | **79.6** | **81.9** | **84.5** | **83.1** | **85.8** | **69.8** | **73.7** |

Table 2: Comparison of methods evaluated on different-task, different-domain transfer scenarios using Qwen3-4B (Q) and LLama2-7B (L) models. Mean accuracy across all tasks is also presented.

**Robust Prefix Adaptation Across Domains.** We investigate our method's ability to transfer across domains under a single-source training setup with zero modification when moving to the target dataset. Table 1 covers same-task and different-domain transfers, and Table 2 covers different-task and different-domain transfers, both on Qwen3-4B (Q) and Llama2-7B (L). In Table 1, ARPO ranks first on all pairs with Mean 84.4/88.0 (Q/L), outperforming DAdEE at 82.3/85.9 by +2.1/+2.1. On MNLI→QQP, ARPO reaches 89.8/94.5 while DAdEE obtains 87.8/92.3, a gain of +2.0/+2.2. In Table 2, the gap is larger under domain-task shift; ARPO records Mean 69.8/73.7 compared with 66.5/70.6, a gain of +3.3/+3.1. On GSM8K→HEval, scores are 32.1/40.9 compared with 25.3/34.6,

improving by +6.8/+6.3. These gains follow from IB-driven disentanglement of domain-invariant and domain-specific prefixes, dynamically gated adversarial training that reduces errors on hard cases, and MOBO that balances accuracy, robustness, and efficiency.

**Efficient Accuracy-Robustness Trade-off Optimization.** Figure 2a illustrates the Pareto frontier, clearly highlighting accuracy-robustness trade-offs among various approaches, while Figure 2b offers a detailed performance and computational cost comparison. Our method notably outperforms the baselines (DePT Shi & Lipani (2023),XPrompt Ma et al. (2022),PT Lester et al. (2021),Radom Search Bergstra & Bengio (2012)), achieving approximately 5% higher accuracy (87% vs. DePT's 82%) and 8% improved robustness (83% vs. DePT's 75%), while reducing computational overhead (0.64 vs. DePT's 0.73 normalized cost). This substantial enhancement is mainly due to our disentangled prefix learning framework, effectively isolating domain-invariant and domain-specific representations through information bottleneck optimization, orthogonality constraints, and contrastive regularization. Furthermore, our dynamic adversarial adaptation mechanism selectively applies adversarial training only where needed, significantly boosting efficiency. Lastly, our multi-objective Bayesian Optimization systematically balances accuracy, robustness, and computational efficiency, enabling stable and robust cross-domain generalization.

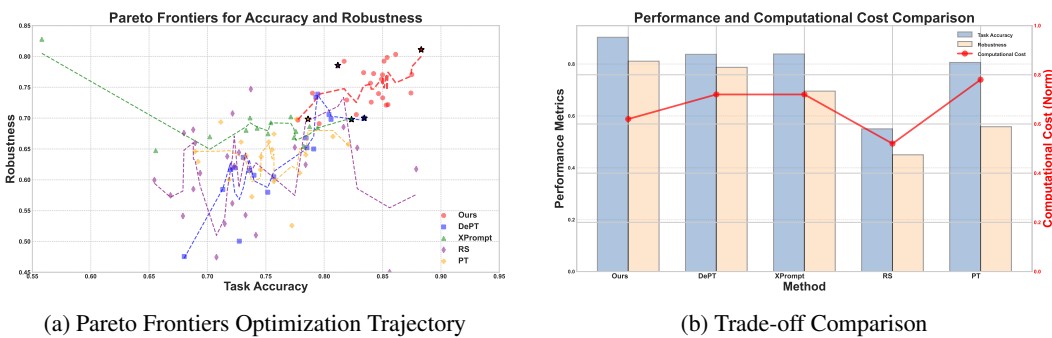

(a) Pareto Frontiers Optimization Trajectory      (b) Trade-off Comparison

Figure 2: Multi-objective optimization performance tested on T5-base model: Figure(a) illustrates Pareto frontiers revealing accuracy-robustness trade-offs across different techniques, while Figure(b) compares the optimal performance metrics (bars) alongside computational cost (red), demonstrating our method's superior balance between task accuracy, robustness, and efficiency.

**Supplementary Analysis of ARPO.** We replicate experiments on T5-base and T5-large. Appendix B.4.1 shows that the module is plug-and-play and integrates cleanly with LLMs of different sizes and architectures. We also perform statistical significance tests on Appendix B.4.3, which show consistent cross-domain gains with narrow uncertainty across tasks. In addition, we list hyperparameters and run sensitivity studies on Appendix B.4.2. With MOBO-based joint search, we efficiently find robust configurations in a large space, leading to stable transfer performance.

## 4.2 ROBUSTNESS ANALYSIS

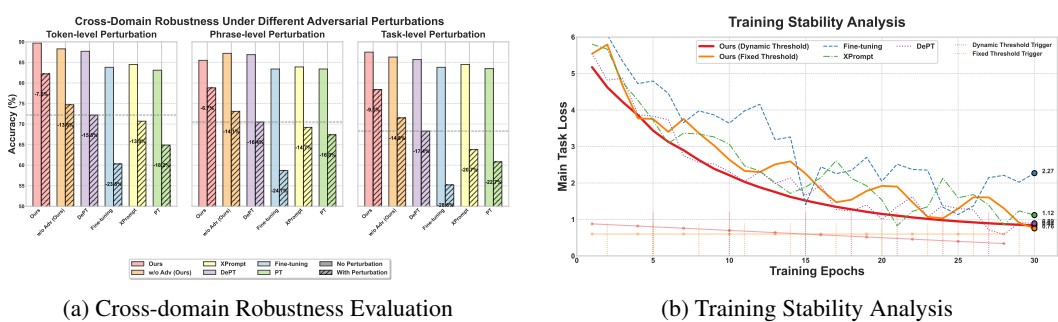

(a) Cross-domain Robustness Evaluation      (b) Training Stability Analysis

Figure 3: Comparison of cross-domain robustness in Figure (a) and training stability in Figure (b), illustrating our method's superior accuracy under adversarial perturbations and enhanced convergence stability relative to baseline methods, evaluated using the T5-base model.

Figure 3a shows cross-domain robustness on T5-base under token-, phrase-, and task-level perturbations; our method attains the best accuracy in all settings, with gains of about 10% over DePT and nearly 22% over standard fine-tuning. Figure 3b shows training stability; our method reduces final training loss by ∼23% vs. DePT and ∼58% vs. standard fine-tuning. The gains come from a dynamic adversarial schedule that triggers only when gradient variance is stable, and from MOBO which searches a mixed discrete–continuous strategy space to cover hard cases. We also learn disentangled prefixes with an IB objective plus orthogonality and conditional independence constraints, and we use independent train and test perturbation protocols to ensure a fair robustness assessment. Our method delivers stronger cross-domain robustness and faster, steadier convergence; full efficiency and robustness analyses, with statistical tests and sensitivity studies, are in Appendix B.4.3 B.4.4.

## 4.3 ABLATION STUDY

**DI:DS prefix length analysis.** Table 3 presents an ablation study exploring how different ratios between domain-invariant (DI) and domain-specific (DS) prefixes affect our method's performance with T5-base and LLaMA-7B models under CombA (same task, cross-domain) and CombB (different tasks and domains) settings. Our optimal 5:5 ratio consistently surpasses the second-best (6:4) by 1.9% (CombA) and 2.4% (CombB), and greatly outperforms the weakest ratio (1:9) by approximately 14.1% and 14.4%, respectively. These improvements highlight the effectiveness of our balanced information bottleneck strategy, promoting an ideal trade-off between domain-invariant and domain-specific information. Additionally, incorporating orthogonality constraints, contrastive disentanglement, and conditional independence ensures clear separation between prefixes, substantially enhancing cross-domain generalization.

| DI:DS | CombA | | CombB | |
|---|---|---|---|---|
| | Base (%) | 7B (%) | Base (%) | 7B (%) |
| 9:1 | 69.5 | 70.4 | 67.3 | 69.1 |
| 8:2 | 72.3 | 74.6 | 71.7 | 73.5 |
| 7:3 | 76.7 | 77.5 | 75.5 | 76.3 |
| 6:4 | 78.2 | 80.8 | 77.2 | 78.0 |
| **5:5** | **80.1** | **84.2** | **79.5** | **83.2** |
| 4:6 | 78.7 | 80.2 | 76.8 | 77.3 |
| 3:7 | 76.5 | 77.1 | 74.2 | 76.2 |
| 2:8 | 71.9 | 73.9 | 71.6 | 73.1 |
| 1:9 | 69.2 | 70.1 | 67.6 | 68.8 |

Table 3: Ablation on DI:DS prefix length (60) ratios for T5-base and LLama-7B, incorporating comparisons of Same Task, Different Domains (Comb A) and Different Tasks, Different Domains (Comb B) with mean scores.

**Impact of Disentanglement Constraints.** Table 4 displays T5-Base and LLaMA-7B ablations for CombA (same task, cross-domain) and CombB (different tasks and domains). In all circumstances, the entire model outperforms w/o all by +9.0/+10.9 on CombA and +8.7/+11.0 on CombB. Removing $L_{\text{disent}}$ decreases CombA to 4.9/6.7 and CombB to 5.2/6.7, showing IB-driven separation as the primary cause Removing $L_{\text{adv}}$ results in 3.3/3.6 and 3.6/3.9 reductions, indicating greater resilience to mismatch. On average, deleting $L_{\text{cons}}$ lowers scores by 5.5, while removing any one restriction causes a 4.4-6.5 reduction. The larger drops on 7B and CombB show that better disentanglement and adversarial regularization improve capacity and cross-task transfer. This obvious separation greatly enhances model resilience and generalization across domain-task combinations.

| Method | CombA | | CombB | |
|---|---|---|---|---|
| | Base (%) | 7B (%) | Base (%) | 7B (%) |
| w/o all | 71.1 | 71.9 | 70.8 | 71.3 |
| w/o $\mathcal{L}_{\text{orth}}$ | 75.2 | 76.8 | 74.2 | 75.9 |
| w/o $\mathcal{L}_{\text{cons}}$ | 75.5 | 76.9 | 74.7 | 75.8 |
| w/o $\mathcal{L}_{\text{cond}}$ | 75.3 | 76.5 | 75.1 | 76.1 |
| w/o $\mathcal{L}_{\text{disent}}$ | 75.2 | 76.1 | 74.3 | 75.6 |
| w/o $\mathcal{L}_{\text{adv}}$ | 76.8 | 79.2 | 75.9 | 78.4 |
| **Our** | **80.1** | **82.8** | **79.5** | **82.3** |

Table 4: Ablation on disentanglement constraints for Base and 7B models, including comparisons of Same Task, Different Domains (Comb A) and Different Tasks, Different Domains (Comb B) with mean scores.

## 5 CONCLUSION

To summarize, we propose ARPO that unifies prefix disentanglement, dynamic adversarial gating, and multi objective Bayesian optimization into a unified pipeline for robust cross domain transfer. In particular, we split the prefix into $P_{DI}$ and $P_{DS}$ with information bottleneck and geometric constraints, trigger adversarial updates by a stability driven threshold, and use product kernel GP surrogates with EHVI to tune strategy structure, strength, and schedule for a better accuracy robustness cost tradeoff. In the future, we will scale the surrogate and search space, add safety and latency objectives, and strengthen theory for convergence under dynamic gating and shifting domains.

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

---

**Algorithm 1:** Training Procedure for ARPO

---

**1 Input:** Base LM $f_\theta$; train set $\mathcal{D}_{train} = \{(x, y, d)\}$ with domain $d$; validation set $\mathcal{D}_{val}$; prefix sizes $(L_{DI}, L_{DS})$, dim $d$; weights $\lambda_{1:5}$; gate params $(\theta_0, \gamma, w)$; adversarial operator library $\{\mathcal{O}_j\}_{j=1}^m$; MOBO budget $T_{BO}$, per-round steps $T_{train}$; initial mixed-decision set $X_0$, where $x = (x_{\text{struct}}, \alpha, \theta)$ and $x_{\text{struct}} = \{(b_j, s_j)\}_{j=1}^m$.

**2 Output:** Learned prefixes $P_{DI}^*, P_{DS}^*$; MOBO-optimal decision $x^* = (x_{\text{struct}}^*, \alpha^*, \theta^*)$; trained model $f_{\theta, P^*}$; Pareto archive $\mathcal{A}_T$ for $(\text{Acc}, \text{Robust}, -\text{Cost})$.

**3 Initialize:** Randomly initialize $P_{DI} \in \mathbb{R}^{L_{DI} \times d}$, $P_{DS} \in \mathbb{R}^{L_{DS} \times d}$; set $\mathcal{D}_0 \leftarrow \emptyset$, $\mathcal{A}_0 \leftarrow \emptyset$.

**4 (I) Initial design evaluation**

**5 foreach** $x = (x_{struct}, \alpha, \theta) \in X_0$ **do**

  **6**    **for** $t = 1$ **to** $T_{train}$ **do**

    **7**      Sample minibatch $\mathcal{B} \subset \mathcal{D}_{train}$; forward $h(\cdot; [P_{DI}; P_{DS}])$ and extract $h_{DS}$

    **8**      *Task loss:* compute $\mathcal{L}_{task}$ on $\mathcal{B}$

    **9**      *Disentanglement loss:*
         $\mathcal{L}_{\text{disent}} = \lambda_1 \mathcal{L}_{IB}(P_{DI}) + \lambda_2 \mathcal{L}_{IB}(P_{DS}) + \lambda_3 \mathcal{L}_{orth} + \lambda_4 \mathcal{L}_{cont} + \lambda_5 \mathcal{L}_{cond}$

    **10**     *Gate score:* $\delta \leftarrow 1 - \mathcal{L}_{task}(t)/\mathcal{L}_{task}(0)$,

    **11**     $\hat{\theta}(t) \leftarrow \theta_0 \cdot (1 - \exp(-\gamma \cdot \text{Var}/\text{Mean}))$ using last $w$ gradient steps

    **12**     **if** $\delta \geq \hat{\theta}(t)$ **then**

    **13**       Build Pos/Neg in $h_{DS}$ via batch distance quantiles;
           $\mathcal{L}_{adv} = \sum_{(i,j) \in Pos} \|h_{DS,i} - h_{DS,j}\|_2 - \sum_{(i,k) \in Neg} \|h_{DS,i} - h_{DS,k}\|_2$;

    **14**     **else**

    **15**       $\mathcal{L}_{adv} \leftarrow 0$;

    **16**     **end**

    **17**     *Total:* $\mathcal{L}_{total} = \mathcal{L}_{task} + \alpha \mathcal{L}_{adv} + \mathcal{L}_{\text{disent}}$; update $\{\theta, P_{DI}, P_{DS}\}$

  **18**   **end**

  **19**   Evaluate $\mathbf{f}(x) = [\text{Acc}, \text{Robust}, -\text{Cost}]$ on $\mathcal{D}_{val}$ with a robustness protocol *independent* from training perturbations; append $(x, \mathbf{f}(x))$ to $\mathcal{D}_0$ and update $\mathcal{A}_0$

**20 end**

**21** Fit GP surrogates on $\mathcal{D}_0$ with a product kernel over mixed $(b, s, \alpha, \theta)$; set reference $\mathbf{r}_0$ (e.g., monotone 5th percentile)

**22 (II) MOBO loop (noisy-$q$ EHVI)**

**23 for** $t = 1$ **to** $T_{BO}$ **do**

  **24**   $X_t \leftarrow \arg\max_{|X|=q} \text{EHVI}(X \mid \mathcal{D}_{t-1}, \mathbf{r}_{t-1})$

  **25**   **foreach** $x = (x_{struct}, \alpha, \theta) \in X_t$ **do**

    **26**    **for** $s = 1$ **to** $T_{train}$ **do**

    **27**     Repeat the inner training of (I) under decision $x$

    **28**    **end**

    **29**    Evaluate $\mathbf{f}(x)$ on $\mathcal{D}_{val}$; append to $\mathcal{D}_t$ and update $\mathcal{A}_t$

  **30**   **end**

  **31**   Refit GP surrogates; update $\mathbf{r}_t$

**32 end**

**33 Return:** select $x^*$ from $\mathcal{A}_T$ (e.g., Pareto knee or scalarization) and output $P_{DI}^*, P_{DS}^*, x^*, f_{\theta, P^*}$.

---

## A   STATEMENT ON THE USE OF LARGE LANGUAGE MODELS

We used a large language model (e.g. ChatGPT, Claude) solely to aid and polish writing (grammar, phrasing, and clarity). The model did not generate technical content, analyses, or results, and all outputs were reviewed and verified by the authors.

## B   TECHNICAL APPENDICES AND SUPPLEMENTARY MATERIAL

### B.1   PRELIMINARY

**Prefix Tuning and Cross-Domain Transfer.** Prefix tuning enables efficient fine-tuning by inserting learnable prefix vectors into pretrained Transformers. Given input $\mathbf{x} = (x_1, x_2, \ldots, x_n)$, the model

computes outputs as Transformer$(\mathbf{x}) =$ Attention$(\mathbf{x} + \mathbf{P}_{\text{prefix}})$, updating only the prefix parameters $\mathbf{P}_{\text{prefix}}$ during training. This improves parameter efficiency and transferability. Traditional domain adaptation minimizes differences between source $P_S(X, Y)$ and target $P_T(X, Y)$ distributions using a domain-invariant extractor $\phi$: $D(P_S(X), P_T(X)) = \min_\phi \text{Div}(\phi(X_S), \phi(X_T))$. We extend this approach to multi-domain scenarios, enhancing cross-domain robustness.

**Information Bottleneck and Mutual Information Estimation.** The Information Bottleneck (IB) theory treats deep neural networks as mechanisms that compress irrelevant information while preserving task-relevant details. Its objective is to maximize $I(\phi(X); Y) - \beta I(X; \phi(X))$, where $I(X; Y)$ denotes mutual information and $\beta$ controls the compression level. Directly computing $I(X; Y)$ is challenging, so Mutual Information Neural Estimation (MINE) provides a variational lower bound:

$$I(X; Y) \geq \hat{I}_{\text{MINE}}(X, Y) = \mathbb{E}_{P_{XY}}[T_\theta(x, y)] - \log(\mathbb{E}P_X P_Y[e^{T\theta(x,y)}]), \tag{17}$$

where $T_\theta$ is a parameterized neural network. To further decouple domain-invariant ($P_{DI}$) and domain-specific ($P_{DS}$) representations, an orthogonality constraint $L_{\text{orth}} = |P_{DI}^T \cdot P_{DS}|_F^2$ is imposed, ensuring approximate orthogonality to mitigate redundancy and negative transfer

**Adversarial Training and Multi-Level Perturbations.** Adversarial training enhances robustness by introducing perturbations $\delta$ into input or intermediate representations. The objective is:

$$\min_\theta \mathbb{E}_{(x,y)\sim D}\left[\max_{\|\delta\|\leq\epsilon} \mathcal{L}\big(f_\theta(x + \delta), y\big)\right], \tag{18}$$

where $\epsilon$ determines perturbation strength. Token-level perturbations alter individual tokens via $x'_{\text{token}} = x_{\text{token}} + \delta$, constrained by $\|\delta\| \leq \epsilon_{\text{token}}$. Task-level perturbations mix samples from different tasks as $x'_{\text{task}} = \alpha x_{\text{task1}} + (1 - \alpha)x_{\text{task2}}$, where $0 < \alpha < 1$. Perturbations at higher levels create larger distribution shifts, enabling stronger evaluations of generalization.

**Multi-Objective Bayesian Optimization.** When dealing with high-dimensional hyperparameter tuning for performance, robustness, and efficiency, methods like manual tuning often struggle to find good solutions. Multi-Objective Bayesian Optimization (MOBO) addresses this by modeling multiple objectives $f_i(x)$ with Gaussian Processes, written as $\mathbf{f}(x) \sim GP(\mu(x), K(x, x'))$. Under limited resources, MOBO iteratively refines the Pareto frontier, defined as

$$\{x \mid \nexists x', \forall i : f_i(x') \leq f_i(x), \exists j : f_j(x') < f_j(x)\}. \tag{19}$$

By strategically selecting hyperparameter configurations that maximize information gain, MOBO automates optimization and balances trade-offs among objectives.

## B.2 RATIONALE FOR USING SOFT PROMPTS

We use soft prompts for control, interpretation, and reproducibility. They confine trainable parameters to the prefix space and scale linearly with length and dimension, far below full fine tuning. Prior work shows strong performance with few vectors and gains with scale (Lester et al., 2021), effective layer control under frozen backbones (Li & Liang, 2021), and results comparable to full fine tuning (Liu et al., 2021c). This compact channel lets us measure mutual information, HSIC, conditional dependence, and geometric margins in the prefix space without changing backbone weights. Attribution therefore stays on our design rather than backbone drift.

Soft prompts fit ARPO and bring engineering gains. Module one performs information bottleneck domain disentanglement at the prefix layer. Module two shapes the domain specific subspace and gates the start time using progress, stability, and task difficulty. The outer loop uses multi objective Bayesian optimization to search a Pareto front over structure, strength, and timing. The small parameter count enables frequent gating and MOBO probing under realistic compute, which reduces adversarial instability and hyperparameter cost. The pipeline thus performs joint adaptation over time, space, and strength in a parameter efficient channel.

This choice remains general. The three ARPO mechanisms are orthogonal to the adaptation layer and transfer to LoRA, Adapters, or full fine tuning (Hu et al., 2021; Houlsby et al., 2019). MOBO is even more valuable with larger models and complex architectures. Soft prompts provide a clean testbed to combine domain disentanglement, dynamic adversarial training, and MOBO strategy search, yielding reproducible evidence and a reusable foundation for PEFT and larger models.

### B.3 EXPERIMENTAL SETTING

In this experiment, we trained and evaluated T5-base, T5-large Raffel et al. (2020), LLaMA-2-7B Touvron et al. (2023), and Qwen3-4B Yang et al. (2025) on a single NVIDIA A6000 GPU with 48 GB memory. The prefix length was 60 and training ran for 50 epochs. Hyperparameters were tuned with Multi-Objective Bayesian Optimization (MOBO) in a low-dimensional latent space using 500 intrinsic dimensions and 60 virtual tokens. The initial MOBO configuration used a learning rate of $1 \times 10^{-4}$, a linear warm-up over $10\%$ of training, dropout 0.1, batch size 32, and weight decay 0.01, targeting stable and efficient optimization. Random projections were initialized with dynamic scaling factors $\alpha$ and standard deviations $\sigma$. We first collected five Sobol points, then performed 30 Bayesian iterations in batches of 64, iteratively refining the prompt embeddings without retraining the full model to maintain parameter efficiency and robust cross-task performance.

Additionally, we evaluated our model's ability to generalize and transfer knowledge across tasks and domains using datasets from GLUE Wang et al. (2018), Super GLUE Fisch et al. (2019), MRQA 2019 shared tasks Fisch et al. (2019), and additional datasets including Yelp Zhang et al. (2015), SciTail Khot et al. (2018), PAWS-Wiki Zhang et al. (2019), GSM8K Cobbe et al. (2021), and HumanEval Chen et al. (2021). We compared our method against several baselines, including Fine-tuning Radford et al. (2019), Adapter Houlsby et al. (2019), BitFit Zaken et al. (2021), SPoT Vu et al. (2021), ATTEMPT Asai et al. (2022), XPROMPT Ma et al. (2022), InfoPrompt Wu et al. (2024), DePT Shi & Lipani (2023), DAdEE Bajpai & Hanawal (2024), and Udapter Malik et al. (2023).

#### B.3.1 SAME TASKS AND DOMAIN DIFFERENCES DATASETS EXPLANATION:

**NQ → SQA**: Both datasets involve question-answering tasks but originate from distinct domains. The Natural Questions (NQ) dataset comprises real user queries sourced from the Google search engine, whereas the Sequential Question Answering (SQA) dataset includes sequentially dependent questions based on Wikipedia paragraphs. Thus, the two datasets differ significantly in their contextual backgrounds and query formats.

**Yelp → SciTail**: Both datasets focus on text classification or entailment tasks. The Yelp dataset contains everyday scenarios such as restaurant reviews, whereas the SciTail dataset consists of textual entailment examples extracted from scientific literature, highlighting substantial domain differences.

**News → HP**: Both datasets involve text classification tasks. The News dataset contains general news articles, while the Hyperpartisan (HP) dataset specifically targets classification of news articles based on their partisan political orientation. Consequently, they differ considerably in content style and topical characteristics.

**MNLI → QQP**: Both tasks involve natural language inference or semantic similarity judgment. Multi-Genre Natural Language Inference (MNLI) emphasizes inference across various textual genres, whereas Quora Question Pairs (QQP) concentrates specifically on evaluating semantic similarity between pairs of questions. Thus, their task contexts and application scenarios differ markedly.

**CoLA → PAWS**: Both datasets pertain to linguistic acceptability or semantic analysis tasks. The Corpus of Linguistic Acceptability (CoLA) dataset is utilized for grammatical acceptability judgments, while the Paraphrase Adversaries from Word Scrambling (PAWS) dataset aims to detect whether sentences retain meaning after word reordering. These datasets exhibit significant domain and task-specific differences.

#### B.3.2 DIFFERENCES TASKS AND DOMAIN DIFFERENCES DATASETS EXPLANATION:

**CoLA → QQP:** Regarding task differences, CoLA (Corpus of Linguistic Acceptability) is designed for grammatical acceptability judgments, primarily assessing whether sentences adhere to linguistic correctness. Conversely, QQP (Quora Question Pairs) involves evaluating the semantic similarity between pairs of questions, representing a fundamentally distinct task. Concerning domain differences, CoLA deals with linguistic analysis typically within an academic linguistic framework, while QQP encompasses colloquial and everyday user-generated questions.

**GSM8K → HumanEval:** GSM8K has natural language math problems that require multistep numerical reasoning and a single numeric answer. HumanEval asks for executable Python functions from text specifications that must pass unit tests; the output shifts from a scalar to a program, and

the metric from accuracy to pass@k. The domains differ: GSM8K is educational arithmetic with everyday language, while HumanEval is software engineering that relies on code syntax, libraries, and algorithmic patterns. Inputs, reasoning, and errors also diverge: narrative prompts vs function signatures, arithmetic chains vs program planning and control flow, and calculation or unit mistakes vs syntax, semantic, or edge case bugs.

**BoolQ → NQ:** BoolQ involves binary yes-or-no question-answering, requiring models to provide definitive affirmative or negative responses. In contrast, NQ (Natural Questions) demands extractive question answering, where models must identify and extract precise answer spans from documents. Regarding domains, BoolQ questions are typically closed-ended and clearly structured, while NQ questions are sourced from actual user queries on Google, featuring greater openness and diversity.

**HP → SciTail:** For task differences, HP (Hyperpartisan) pertains to classifying news articles according to partisan political alignment, whereas SciTail addresses textual entailment recognition specifically within scientific contexts. Concerning domain differences, HP data is centered on subjective political viewpoints and biases in news reporting, while SciTail content is derived from objective scientific literature characterized by rigorous logic and structured reasoning, thus representing distinctly separate domains.

**MNLI → SQA:** Regarding task distinctions, MNLI (Multi-Genre Natural Language Inference) involves inference-based evaluations of logical relationships between text pairs across multiple genres. Conversely, SQA (Sequential Question Answering) is concerned with extractive answering of sequentially dependent questions based on provided contexts. Domain-wise, MNLI spans diverse text genres and styles, whereas SQA specifically targets continuous information extraction from Wikipedia articles, underscoring notable differences in both textual nature and application context.

## B.4 SUPPLEMENTARY EXPERIMENTS

### B.4.1 EXPERIMENTS ON T5 BASE & T5 LARGE MODELS

| Method | NQ→SQA | | Yelp→SciTail | | News→HP | | MNLI→QQP | | CoLA→PAWS | | Mean | |
|---|---|---|---|---|---|---|---|---|---|---|---|---|
| | B | L | B | L | B | L | B | L | B | L | B | L |
| **Fine-tuning** | 68.9 | 71.2 | 87.3 | 90.5 | 65.2 | 67.8 | 75.3 | 78.8 | 66.5 | 69.3 | 72.6 | 75.5 |
| **Adapter** | 72.5 | 73.8 | 89.1 | 91.3 | 65.7 | 67.9 | 77.8 | 80.2 | 68.4 | 70.5 | 74.7 | 76.7 |
| **BitFit** | 68.6 | 71.9 | 83.9 | 87.0 | 66.7 | 69.2 | 76.1 | 82.3 | 67.9 | 70.3 | 72.7 | 76.1 |
| **PT** | 64.1 | 69.3 | 86.7 | 89.8 | 69.3 | 69.7 | 75.4 | 78.7 | 64.1 | 69.5 | 71.9 | 75.4 |
| **PT-2** | 65.8 | 71.1 | 87.2 | 90.1 | 68.2 | 71.5 | 77.2 | 80.1 | 66.3 | 70.1 | 72.9 | 76.6 |
| **SPoT** | 65.9 | 70.2 | 85.6 | 89.7 | 67.4 | 70.5 | 78.9 | 82.3 | 67.5 | 70.8 | 73.1 | 76.7 |
| **ATTEMPT** | 69.9 | 74.4 | 86.4 | 90.7 | 68.6 | 71.2 | 79.1 | 82.5 | 68.5 | 71.2 | 74.5 | 78.0 |
| **XPROMPT** | 70.9 | 73.2 | 86.1 | 90.3 | 70.8 | 72.1 | 80.2 | 84.5 | 69.4 | 71.8 | 75.5 | 78.4 |
| **InfoPrompt** | 72.4 | 75.2 | 87.3 | 91.5 | 69.5 | 71.8 | 81.6 | 85.7 | 70.8 | 72.1 | 76.3 | 79.3 |
| **DEPT** | 73.6 | 75.9 | 89.4 | 93.7 | 70.2 | 74.5 | 83.0 | 86.9 | 71.0 | 73.3 | 77.4 | 80.9 |
| **ARPO** | **76.2** | **79.5** | **92.5** | **96.7** | **72.9** | **76.8** | **85.6** | **89.7** | **73.1** | **76.4** | **80.1** | **83.8** |

Table 5: Comparison of methods evaluated on cross-domain transfer tasks using the T5-base (B) and T5-large (L) models. Mean accuracy across all tasks is also reported.

Section B.4.1 reports cross domain transfer on T5 base and T5 large. Table 5 shows that our method attains a mean of 80.1 on base and 83.8 on large, which exceeds DePT by 2.7 and 2.9 points. The advantage holds on every transfer. NQ→SQA improves by 2.6 on base and 3.6 on large. Yelp→SciTail improves by 3.1 and 3.0. News→HP improves by 2.7 and 2.3. MNLI→QQP improves by 2.6 and 2.8. CoLA→PAWS improves by 2.1 and 3.1. Yelp→SciTail reaches 96.7 on T5 large, while News→HP is the hardest but still gains. Scaling from base to large adds 3.7 points for our method and 3.5 for DePT, so the margin remains. These results arise because the disentangled prefix separates domain invariant and domain specific signals, which reduces negative transfer and preserves task cues. The dynamic threshold schedules adversarial updates only when the task loss improvement and gradient statistics indicate value, which avoids early noise and focuses learning on the domain specific space. The MOBO search finds effective strategy settings in a small number of trials, which balances accuracy, robustness, and cost better than manual tuning. The parameter efficient design lowers the risk of overfitting and stabilizes training, so gains persist when model capacity increases.

| Method | CoLA→QQP | | RTE→SST-2 | | BoolQ→NQ | | HP→SciTail | | MNLI→SQA | | Mean | |
|---|---|---|---|---|---|---|---|---|---|---|---|---|
| | B | L | B | L | B | L | B | L | B | L | B | L |
| **Fine-tuning** | 64.2 | 66.7 | 77.4 | 80.2 | 68.1 | 71.5 | 69.7 | 73.1 | 71.9 | 75.3 | 70.3 | 73.4 |
| **Adapter** | 67.9 | 69.3 | 79.4 | 81.8 | 70.8 | 71.9 | 72.1 | 74.8 | 71.1 | 77.5 | 72.3 | 75.1 |
| **BitFit** | 66.4 | 68.8 | 79.1 | 82.5 | 69.5 | 72.9 | 71.8 | 74.2 | 71.3 | 77.4 | 71.6 | 75.2 |
| **PT** | 65.5 | 69.5 | 78.5 | 82.9 | 68.3 | 72.2 | 71.6 | 74.7 | 70.9 | 76.5 | 71.0 | 75.2 |
| **PT-2** | 66.2 | 68.0 | 79.2 | 83.1 | 69.2 | 72.8 | 72.1 | 75.2 | 72.2 | 77.2 | 71.8 | 75.3 |
| **SPoT** | 66.8 | 68.5 | 79.5 | 83.3 | 70.9 | 73.1 | 74.2 | 77.1 | 74.9 | 79.9 | 73.3 | 76.4 |
| **ATTEMPT** | 67.2 | 70.6 | 80.8 | 83.8 | 71.4 | 73.8 | 74.8 | 77.9 | 75.5 | 80.6 | 73.9 | 77.3 |
| **XPROMPT** | 68.5 | 72.1 | 81.1 | 83.9 | 71.9 | 74.2 | 75.5 | 78.5 | 76.2 | 81.1 | 74.6 | 78.0 |
| **InfoPrompt** | 68.9 | 72.5 | 81.7 | 84.1 | 72.2 | 74.9 | 75.9 | 79.7 | 76.8 | 81.9 | 75.1 | 78.6 |
| **DEPT** | 69.4 | 73.8 | 82.5 | 85.9 | 72.8 | 75.2 | 76.6 | 80.3 | 77.7 | 83.3 | 75.8 | 79.7 |
| **ARPO** | 73.5 | 76.8 | 85.1 | 88.4 | 76.3 | 79.6 | 80.1 | 83.5 | 82.5 | 86.2 | 79.5 | 82.9 |

Table 6: Comparison of methods evaluated on different-task, different-domain transfer scenarios using T5-base (B) and T5-large (L) models. Mean accuracy across all tasks is also presented.

Table 6 reports different task and different domain transfer on T5 base and T5 large, where our method achieves the best mean accuracy in both settings, with 79.5 on base and 82.9 on large, exceeding DePT by 3.7 and 3.2 points. The gains are consistent across all transfers. For CoLA→QQP the improvements over DePT are 4.1 on base and 3.0 on large. For RTE→SST–2 the improvements are 2.6 and 2.5. For BoolQ→NQ the improvements are 3.5 and 4.4. For HP→SciTail the improvements are 3.5 and 3.2. For MNLI→SQA the improvements are 4.8 and 2.9. These results arise because the disentangled prefix separates domain invariant and domain specific information, which limits negative transfer while preserving task signals. The dynamic threshold schedules adversarial updates only when loss progress and gradient statistics indicate value, which prevents early noise and focuses adaptation on domain specific features. The multi objective Bayesian search selects effective strategy settings with few evaluations, which balances accuracy, robustness, and cost better than manual tuning. The parameter efficient design reduces overfitting risk and stabilizes training across tasks and model sizes, so the advantage persists when scaling from base to large.

### B.4.2 EXPERIMENTS ON HYPERPARAMETER SENSITIVITY

| Parameter | Search Range | Value 1 | Value 2 | Value 3 | Value 4 | Value 5 | Value 6 | Optimal Range |
|---|---|---|---|---|---|---|---|---|
| $\lambda_1$ | $[0.1, 1.0]$ | $76.2 \pm 1.6$ (0.1) | $78.4 \pm 1.5$ (0.3) | $80.1 \pm 1.2$ **(0.5)** | $79.8 \pm 1.2$ (0.7) | $79.2 \pm 1.3$ (0.9) | $77.8 \pm 1.5$ (1.0) | $[0.5, 0.7]$ |
| $\lambda_2$ | $[0.1, 1.0]$ | $75.8 \pm 1.5$ (0.1) | $77.9 \pm 1.3$ (0.3) | $80.1 \pm 1.2$ **(0.5)** | $79.3 \pm 1.1$ (0.7) | $79.4 \pm 1.2$ (0.9) | $78.1 \pm 1.4$ (1.0) | $[0.5, 0.7]$ |
| $\lambda_3$ | $[0.01, 0.5]$ | $79.4 \pm 1.5$ (0.01) | $79.8 \pm 1.2$ (0.05) | $80.1 \pm 1.2$ **(0.1)** | $79.6 \pm 1.1$ (0.2) | $78.9 \pm 1.3$ (0.3) | $77.2 \pm 1.6$ (0.5) | $[0.05, 0.1]$ |
| $\lambda_4$ | $[0.05, 0.3]$ | $79.2 \pm 1.3$ (0.05) | $79.7 \pm 1.3$ (0.1) | $79.9 \pm 1.1$ (0.15) | $80.1 \pm 1.2$ **(0.2)** | $79.5 \pm 1.2$ (0.25) | $78.8 \pm 1.4$ (0.3) | $[0.15, 0.25]$ |
| $\lambda_5$ | $[0.01, 0.2]$ | $79.8 \pm 1.5$ (0.01) | $80.0 \pm 1.1$ (0.03) | $80.1 \pm 1.2$ **(0.05)** | $79.9 \pm 1.0$ (0.08) | $79.4 \pm 1.2$ (0.12) | $78.6 \pm 1.5$ (0.2) | $[0.03, 0.08]$ |

Table 7: Hyperparameter sensitivity results (mean ± standard deviation). Bold candidate values in parentheses indicate the best-performing setting within each row.

Table 7 shows that performance is most sensitive to the Information Bottleneck weights in Equation (6), with peaks at $\lambda_1 = 0.5$ and $\lambda_2 = 0.5$ where the mean reaches $80.1 \pm 1.2$ and then declines toward both ends of the search ranges. The orthogonality term works best around $\lambda_3 = 0.1$ with a stable region in $[0.05, 0.10]$. The contrastive term prefers moderate strength with $\lambda_4 = 0.2$ and remains strong in $[0.15, 0.25]$. The conditional independence term is most effective near $\lambda_5 = 0.05$ with a stable region in $[0.03, 0.08]$. These results indicate that the main gains come from balancing domain invariant extraction and domain specific retention, since $\lambda_1$ and $\lambda_2$ control the trade off between removing domain cues and preserving task signals; too small values under regularize and allow leakage across prefixes, while too large values over regularize and remove useful information. The orthogonality weight avoids representation mixing and reduces redundancy, but if it is too high it limits capacity and hurts alignment. The contrastive and conditional independence terms improve structure and separability when set to moderate values; if they are too weak they fail to

| Notation | Component | Update Method | Description (Range/Setting) |
|---|---|---|---|
| $L$ | Total prefix length | Backprop (Adam) | Total tokens, split as $L = L_{DI} + L_{DS}$, range: 10–50. |
| $L_{DI}$ | Domain-invariant prefix | Backprop (Adam) | Prefix tokens for domain-invariant features, integer $[0, L]$. |
| $L_{DS}$ | Domain-specific prefix | Backprop (Adam) | Prefix tokens for domain-specific features, integer $[0, L]$. |
| $d$ | Embedding dimension | Fixed by backbone | Embedding size (e.g., T5-base: $d = 768$). |
| $\beta_1, \beta_2$ | IB weighting factors | Backprop (Adam) | Balances mutual information objectives, real $\geq 0$. |
| $\tau$ | Contrastive temperature | Backprop (Adam) | Contrastive similarity scaling, typical range: 0.01–0.2. |
| $\lambda_0$ | Orthogonality penalty base | Backprop (Adam) | Base factor for adaptive orthogonality, real $> 0$. |
| $\lambda_1, \ldots, \lambda_5$ | Disentanglement weights | Backprop (Adam) | Weights for losses $L_{IB}, L_{orth}, L_{cont}, L_{cond}$, real $\geq 0$. |
| $\alpha$ | Adversarial balance | Backprop (Adam) | Balances $\mathcal{L}_{adv}$ and $\mathcal{L}_{task}$, real in [0,1]. |
| $\delta(\mathcal{L}_{task}(t))$ | Improvement ratio | Computed per iteration | Task loss improvement ratio, real in [0,1]. |
| $\theta(t)$ | Dynamic threshold | Computed per iteration | Adversarial activation threshold, real in [0,1]. |
| $\theta_0$ | Base threshold | Hyperparameter | Initial threshold for $\theta(t)$, real in [0,1]. |
| $\gamma$ | Threshold sensitivity | Hyperparameter | Gradient variance sensitivity, real $> 0$. |
| $\beta$ | Task-difficulty sensitivity | Hyperparameter | Scales threshold by task difficulty, real $\geq 0$. |
| $w$ | Lookback window | Fixed integer | Recent batches for variance calculation, typical: 5–20. |
| Pos/Neg threshold | Pair distance threshold | Hyperparameter | Threshold for positive/negative pairs, real $> 0$. |
| $\mathcal{L}_{adv}$ | Adversarial loss weight | Backprop (Adam) | Loss computed from positive/negative pairs. |
| $T$ | Max training iterations | Outer loop | Optimization steps, integer $> 0$. |
| $\{\mathcal{O}_j\}$ | Atomic adversarial ops | BO-chosen | Basic ops (FGSM, PGD, swaps), each with toggle $b_j$, strength $s_j$. |
| $b_j \in \{0,1\}$ | Discrete op toggle | BO (GP model) | Operation active/inactive binary indicator. |
| $s_j \in \mathbb{R}$ | Continuous op strength | BO (GP model) | Magnitude of perturbation, typical range [0,5]. |
| Strategy | Adversarial strategy | MOBO-selected | Combination of $(b_j, s_j)$ pairs, updated per iteration. |
| EHVI | Acquisition function | MOBO/GP | Expected Hypervolume Improvement for strategy selection. |

Table 8: Key Parameters and Hyperparameters in the Adaptive Robust Prefix Optimization

guide the split, and if they are too strong they force overly rigid clusters. The observed optima match the automated search in Section 3.3.2, where the multi objective Bayesian optimization treats $x = (\alpha, \theta, \lambda_1, \ldots, \lambda_5, \text{Strategy})$ as decision variables, starts from Sobol initialization, and converges within about 30 iterations; ablations attribute about 60% of the total gain to $\lambda_1$ and $\lambda_2$.

As shown in the Table 8, it summarizes the key parameters guiding our method's core functionalities. The prefix lengths ($L, L_{DI}, L_{DS}$) separate domain-invariant and domain-specific features, while mutual information and orthogonality weights ($\beta_1, \beta_2, \lambda_1, \ldots, \lambda_5$) control disentanglement strength. The dynamic threshold parameters ($\theta(t), \theta_0, \gamma, \beta$) manage when adversarial training begins, preventing early interference. Adversarial strategies (e.g., $\{\mathcal{O}_j\}, b_j, s_j$) and multi-objective Bayesian optimization (EHVI) collaborate to tune both discrete and continuous components, enhancing cross-domain robustness and reducing manual efforts.

### B.4.3 EXPERIMENT ON SIGNIFICANCE STUDY

The significance study in Table 9 indicates consistent cross domain transfer performance with narrow uncertainty across tasks. On the dev set, we report means and 95% confidence intervals over 5 independent runs: the overall mean is $80.1 \pm 1.2$ with a 95% interval of [78.0, 82.2] and a 1.49% coefficient of variation; Yelp→SciTail attains $92.5 \pm 1.2$, MNLI→QQP reaches $85.6 \pm 1.2$, NQ→SQA and CoLA→PAWS yield $76.2 \pm 1.3$ and $73.1 \pm 1.1$, while News→HP is the hardest at $72.9 \pm 1.3$, indicating stable training and limited run to run variance. These outcomes stem from three design choices. Information bottleneck

| Task Transfer | Mean | 95% CI | CV |
|---|---|---|---|
| NQ→SQA | $76.2 \pm 1.3$ | [74.6, 77.8] | 1.44% |
| Yelp→SciTail | $92.5 \pm 1.2$ | [91.0, 94.0] | 1.30% |
| News→HP | $72.9 \pm 1.3$ | [71.3, 74.5] | 1.78% |
| MNLI→QQP | $85.6 \pm 1.2$ | [83.7, 87.1] | 1.40% |
| CoLA→PAWS | $73.1 \pm 1.1$ | [71.4, 74.5] | 1.51% |
| **Mean** | $80.1 \pm 1.2$ | **[78.0, 82.2]** | **1.49%** |

Table 9: Cross-domain task transfer results with mean $\pm$ standard deviation, 95% confidence interval (95% CI), and coefficient of variation (CV).

driven prefix disentanglement separates domain invariant and domain specific signals, which reduces negative transfer while preserving task cues. A dynamic adversarial schedule activates only after loss stabilization, improving robustness without early instability. Multiobjective Bayesian optimization tunes the adversarial weight, trigger, and strategy to balance accuracy and robustness. The residual gap on News→HP reflects stronger distribution shift and label subjectivity, leaving headroom for future refinement.

### B.4.4 Model Runtime and Efficiency Experiments

Under identical training conditions (NVIDIA A6000 GPU, 50 epochs, batch size of 32, T5-base backbone), we report reproducible wall-clock times and normalized costs. Full fine-tuning requires 120 min/epoch (approximately 100 h total; normalized cost 1.00); DePT completes in 52 min/epoch (approximately 43.3 h; cost Model Runtime and Efficiency Experiments). In contrast, ARPO trains in 45 min/epoch (approximately 37.5 h; cost 0.64), yielding a 13.4% reduction relative to DePT. All measurements are obtained on the same hardware and schedule to ensure a fair comparison.

The efficiency gains arise from the synergy of parameter economy, selective adversarial computation, and sample-efficient strategy search. ARPO updates fewer than 1% of parameters, which reduces backward-pass overhead and memory traffic. Its dynamic thresholding, driven by the progress signal $\delta(L_{\text{task}}(t))$, suppresses unproductive adversarial steps during early training, avoiding roughly 40% of adversarial computations in the first 30% of iterations. Moreover, the MOBO component typically identifies effective operating points in about 30 evaluations, whereas grid search often requires hundreds. At inference, ARPO introduces negligible latency because deployment only concatenates learned prefix embeddings without auxiliary branches or test-time optimization.

## C  Theoretical and Implementation Details for Adaptive Representation Learning of Disentangled Prefix

### C.1  Mutual Information Neural Estimation (MINE): Variational Form, Stabilization, and Bias/Variance Bounds

Let random variables $U, V$ have joint $p(u, v)$ and marginals $p(u), p(v)$. Mutual information is

$$I(U; V) = \mathbb{E}_{p(u,v)}\left[\log \frac{p(u,v)}{p(u)p(v)}\right]. \tag{20}$$

A variational representation follows from the Donsker–Varadhan (DV) inequality for KL: for any measurable $T : \mathcal{U} \times \mathcal{V} \to \mathbb{R}$,

$$I(U; V) = \text{KL}\big(p(u,v) \,\big\|\, p(u)p(v)\big) \geq \sup_{T} \left\{ \mathbb{E}_{p(u,v)}[T(u,v)] - \log \mathbb{E}_{p(u)p(v)}\left[e^{T(u,v)}\right] \right\}. \tag{21}$$

MINE parameterizes $T(u, v) = T_\phi(u, v)$ with a neural network and maximizes the DV lower bound $\widehat{I}_{\text{DV}}(\phi)$ over $\phi$ using stochastic gradients Belghazi et al. (2018). In practice, expectations in equation 21 are replaced by mini-batch Monte Carlo estimates using positive pairs $(u_i, v_i) \sim p(u, v)$ and negative pairs formed by shuffling to approximate $p(u)p(v)$. The empirical objective reads

$$\widehat{I}_{\text{DV}}(\phi) = \frac{1}{B}\sum_{i=1}^{B} T_\phi(u_i, v_i) - \log\left(\frac{1}{B}\sum_{i=1}^{B} e^{T_\phi(u_i, \tilde{v}_i)}\right), \qquad \tilde{v}_i \text{ i.i.d.} \sim p(v), \tag{22}$$

which is a biased but consistent estimator of the DV bound under increasing batch size and training time. The bias arises from the concavity of $\log(\cdot)$ and finite-sample estimation of the denominator; variance arises from the exponential moment $e^{T_\phi}$ that amplifies tail noise. Systematic comparisons show DV is among the tightest common variational MI bounds but exhibits high estimator variance and optimization instability in high dimensions or with small batches Poole et al. (2019).

An alternative lower bound widely used in contrastive learning is InfoNCE Oord et al. (2018). Let $\{(u, v^+), (u, v_1^-), \dots, (u, v_{K-1}^-)\}$ contain one positive sample from $p(u, v)$ and $K - 1$ negatives from $p(u)p(v)$. Define the score $s_\phi(u, v) = T_\phi(u, v)$. The InfoNCE objective is

$$\widehat{I}_{\text{NCE}}(\phi) = \mathbb{E}\left[\log \frac{\exp s_\phi(u, v^+)}{\exp s_\phi(u, v^+) + \sum_{k=1}^{K-1} \exp s_\phi(u, v_k^-)}\right] + \log K, \tag{23}$$

which lower-bounds $I(U; V)$, tightens with $K$, and typically has lower variance than DV due to the softmax normalization. The ordering between DV, NWJ, and InfoNCE in tightness and dispersion is detailed in Poole et al. (2019).

For ARPO, we employ a MINE-style critic $T_\phi$ but stabilize training by controlling the denominator in equation 22. Let

$$\widehat{Z}_t = \frac{1}{B} \sum_{i=1}^{B} e^{T_{\phi_t}(u_i, \tilde{v}_i)}, \qquad \bar{Z}_t = \beta\, \bar{Z}_{t-1} + (1-\beta)\, \widehat{Z}_t, \quad \beta \in [0, 1), \tag{24}$$

and replace $\log \widehat{Z}_t$ by $\log \bar{Z}_t$ in equation 22. The exponential moving average reduces stochastic curvature in the log-partition estimate and yields a controllable bias–variance trade-off; $\bar{Z}_t$ converges in mean to the population moment when batches are i.i.d. and $\beta$ is fixed. We additionally use gradient clipping $\|\nabla_\phi \widehat{I}_{\mathrm{DV}}\| \le c$ to control heavy-tailed gradients induced by $e^{T_\phi}$, and increase the number of negatives per batch when using InfoNCE to tighten equation 23 without destabilizing learning. Empirical and theoretical studies report that DV-like estimators suffer variance blow-up in small-batch or high-dimensional regimes, while contrastive bounds such as InfoNCE trade tightness for stability and sample-efficiency Poole et al. (2019).

Bias and variance can be decomposed at the bound level. Let $I_\star$ denote the true MI. For a generic variational lower bound $\mathcal{L}_\phi$ estimated from $B$ i.i.d. samples,

$$\mathrm{Bias}_B = \mathbb{E}[\widehat{\mathcal{L}}_{\phi_B^\star}] - I_\star \le 0, \qquad \mathrm{Var}_B = \mathbb{V}[\widehat{\mathcal{L}}_{\phi_B^\star}], \tag{25}$$

where $\phi_B^\star$ maximizes the empirical objective. For DV, $\mathrm{Var}_B$ scales with the second moment of $e^{T_\phi}$ under $p(u)p(v)$ and can be large without norm control; for InfoNCE, $\mathrm{Var}_B$ scales with the variance of a bounded log-softmax and is therefore better behaved for fixed $K$ Poole et al. (2019). Further results establish limits showing that high MI cannot be reliably estimated from limited samples without strong inductive bias McAllester & Stratos (2020), motivating our use of MI estimates as soft regularizers and diagnostics rather than primary loss terms.

Connections to $f$-divergences clarify parameterizations. Let $T_\phi$ define a variational class for KL via $f$-GAN; then equation 21 arises by choosing the convex conjugate of $f(t) = t \log t$. Alternative choices yield NWJ and Jensen–Shannon bounds with different curvature and gradient properties Nowozin et al. (2016). These links justify using contrastive parameterizations of $T_\phi$ in equation 23 when stability is paramount.

In ARPO, MI terms appear in the disentanglement objective as regularizers. For domain-invariant prefixes we employ an InfoNCE-style estimator with temperature and a moderate number of negatives to enhance stability and prevent variance amplification; for domain-specific leakage penalties we optionally use a DV-style MINE with EMA-stabilized partition function $\bar{Z}_t$ in equation 24 to detect residual dependence. Both estimators are computed on held-out mini-batches and enter the loss with small weights, so that noisy MI fluctuations do not dominate the training signal; the dynamic gate and MOBO decide when and how strongly to apply adversarial components based on validation outcomes, not raw MI estimates.

Under bounded critic outputs $|T_\phi| \le M$ and sub-exponential tails for $e^{T_\phi}$ under $p(u)p(v)$, a Bernstein-type concentration bound yields, with probability at least $1 - \delta$,

$$\left| \widehat{I}_{\mathrm{DV}}(\phi) - I_{\mathrm{DV}}(\phi) \right| \le C_1 \sqrt{\frac{\log(2/\delta)}{B}} + C_2 \frac{\log(2/\delta)}{B}, \tag{26}$$

for constants $C_1, C_2$ depending on $M$ and the Orlicz norm of $e^{T_\phi}$. For InfoNCE with fixed $K$, boundedness of the log-softmax implies a Hoeffding-type bound of order $O\left( \sqrt{\frac{\log(1/\delta)}{B}} \right)$. Together with the EMA bias $\log \widehat{Z}_t \mapsto \log \bar{Z}_t$ of order $O(1 - \beta)$ in steady state, these inequalities explain the empirical stability gains from equation 24 and motivate our estimator selection. Formal derivations and bound comparisons are provided in Poole et al. (2019); McAllester & Stratos (2020).

## C.2 HILBERT–SCHMIDT INDEPENDENCE CRITERION: KERNELS, NORMALIZATION, NUMERICAL STABILITY, AND ADAPTIVE RESCALING

Let $(X, Y) \sim P_{XY}$ with marginals $P_X$ and $P_Y$. Let $k : \mathcal{X} \times \mathcal{X} \to \mathbb{R}$ and $\ell : \mathcal{Y} \times \mathcal{Y} \to \mathbb{R}$ be bounded, positive-definite kernels with RKHS $\mathcal{F}$ and $\mathcal{G}$. The cross-covariance operator $C_{XY} : \mathcal{G} \to \mathcal{F}$ is defined by

$$\langle f, C_{XY} g \rangle_\mathcal{F} = \mathbb{E}[(f(X) - \mathbb{E}f(X))(g(Y) - \mathbb{E}g(Y))], \qquad f \in \mathcal{F},\ g \in \mathcal{G}. \tag{27}$$

The Hilbert–Schmidt Independence Criterion (HSIC) is the squared Hilbert–Schmidt norm

$$\mathrm{HSIC}(P_{XY}; \mathcal{F}, \mathcal{G}) \;=\; \|C_{XY}\|_{\mathrm{HS}}^2, \tag{28}$$

which equals zero if and only if $X$ and $Y$ are independent when $k$ and $\ell$ are characteristic Gretton et al. (2007; 2012).

An equivalent population expression is

$$\mathrm{HSIC}(P_{XY}) = \mathbb{E}_{XX'YY'}\big[k(X, X')\,\ell(Y, Y')\big] + \mathbb{E}_{XX'}\big[k(X, X')\big]\,\mathbb{E}_{YY'}\big[\ell(Y, Y')\big]$$
$$- 2\,\mathbb{E}_{XY}\big[\mathbb{E}_{X'}k(X, X')\,\mathbb{E}_{Y'}\ell(Y, Y')\big]. \tag{29}$$

where $(X', Y')$ is an i.i.d. copy of $(X, Y)$.

Given samples $\{(x_i, y_i)\}_{i=1}^n$, define Gram matrices $K = [k(x_i, x_j)]_{i,j}$ and $L = [\ell(y_i, y_j)]_{i,j}$, and the centering matrix $H = I_n - \frac{1}{n}\mathbf{1}\mathbf{1}^\top$. The biased V-statistic estimator is

$$\widehat{\mathrm{HSIC}}_{\mathrm{V}} = \frac{1}{n^2}\mathrm{tr}\big(KHLH\big), \tag{30}$$

and the unbiased U-statistic estimator is Gretton et al. (2012)

$$\widehat{\mathrm{HSIC}}_{\mathrm{U}} = \frac{1}{n(n-1)}\left[\mathrm{tr}(\tilde{K}\tilde{L}) + \frac{\mathbf{1}^\top \tilde{K}\mathbf{1}\,\mathbf{1}^\top\tilde{L}\mathbf{1}}{(n-1)(n-2)} - \frac{2}{n-2}\mathbf{1}^\top\tilde{K}\tilde{L}\mathbf{1}\right], \tag{31}$$

$$\tilde{K} = K - \mathrm{diag}(K),\; \tilde{L} = L - \mathrm{diag}(L). \tag{32}$$

Both equation 30 and equation 31 are consistent for $\mathrm{HSIC}(P_{XY})$.

For Gaussian kernels $k(x, x') = \exp\big(-\|x - x'\|^2/(2\sigma_x^2)\big)$ and $\ell(y, y') = \exp\big(-\|y - y'\|^2/(2\sigma_y^2)\big)$, bandwidths $\sigma_x, \sigma_y$ can be set by data-dependent rules. The median heuristic chooses $\sigma_x^2 = \mathrm{median}\{\|x_i - x_j\| : i < j\}^2/\log 2$ and analogously for $\sigma_y$. Silverman/Scott scaling yields $\sigma_x^2 = c_x\,\hat{s}_X^2\,n^{-2/(d_x+4)}$ and $\sigma_y^2 = c_y\,\hat{s}_Y^2\,n^{-2/(d_y+4)}$, where $\hat{s}$ is a scale estimate and $d_x, d_y$ are intrinsic dimensions Silverman (2018); Scott (2015).

To avoid scale sensitivity and ill-conditioned gradients, normalize HSIC by the Frobenius norms of centered kernels. Let

$$\mathsf{K} = HKH, \qquad \mathsf{L} = HLH, \qquad \widehat{h} \;=\; \frac{\mathrm{tr}(\mathsf{K}\mathsf{L})}{\|\mathsf{K}\|_F\,\|\mathsf{L}\|_F} \in [0, 1], \tag{33}$$

which is invariant to positive rescalings of $K$ or $L$ and coincides with centered kernel alignment up to normalization Cortes et al. (2012). For numerical stability, compute $\mathsf{K}, \mathsf{L}$ via double-centering, optionally add a ridge $\varepsilon I$ inside Gaussian distances in high dimension, and clip $\widehat{h}$ into $[0, 1 - \epsilon]$ for a small $\epsilon > 0$.

In ARPO, the HSIC penalty between embeddings from $P_{DI}$ and $P_{DS}$ is introduced through an adaptive rescaling that implements a curriculum on independence. Let $\widehat{h}_t$ be the batch estimate equation 33 at iteration $t$, and define an exponential moving average

$$\tilde{h}_t \;=\; \eta\,\tilde{h}_{t-1} + (1 - \eta)\,\widehat{h}_t, \qquad \eta \in [0, 1). \tag{34}$$

The penalty is

$$\mathcal{R}_{\mathrm{HSIC}}(X, Y) = \frac{1}{\max\{\epsilon,\, 1 - \tilde{h}_t\}} \;-\; \frac{1}{1 - \epsilon}, \tag{35}$$

which is monotone in $\tilde{h}_t$, equals zero at $\tilde{h}_t = 0$, and increases smoothly as dependence grows. Since $\widehat{h} \in [0, 1]$, equation 35 is bounded by $\epsilon$, and the EMA detaches gradients through $\tilde{h}_{t-1}$ to prevent temporal credit leakage.

Gradients are obtained by differentiating $\mathrm{tr}(\mathsf{K}\mathsf{L})$ and the norms in equation 33. For Gaussian $k$,

$$\frac{\partial K_{ij}}{\partial x_i} = \frac{1}{\sigma_x^2}\,K_{ij}\,(x_j - x_i), \qquad \frac{\partial \mathsf{K}}{\partial x_i} = H\left(\frac{\partial K}{\partial x_i}\right)H, \tag{36}$$

and similarly for $y_i$. The quotient rule yields $\partial \widehat{h}/\partial x_i$ with $\partial \|K\|_F/\partial x_i = \langle K/\|K\|_F, \partial K/\partial x_i \rangle_F$. Under i.i.d. sampling, $\widehat{\text{HSIC}}_V$ and $\widehat{\text{HSIC}}_U$ converge to $\text{HSIC}(P_{XY})$ with variance $O(n^{-1})$ Gretton et al. (2012). For weakly dependent mini-batches, variance control is aided by normalization equation 33 and the EMA in equation 35; for independence testing with dependence, wild bootstrap schemes provide consistent null approximations Chwialkowski et al. (2014).

Kernel choice follows representation geometry. For high-dimensional continuous embeddings we use Gaussian kernels with the median heuristic for characteristicness without extra hyperparameters. When linear dependence control suffices, linear kernels $k(x, x') = x^\top x'$ and $\ell(y, y') = y^\top y'$ reduce computation and align with equation 33. In practice $\sigma_x, \sigma_y$ are initialized by the median heuristic and refreshed periodically if the empirical distance distribution drifts, while a small ridge and clipping of $\widehat{h}$ ensure Lipschitz behavior of $\mathcal{R}_{\text{HSIC}}$ on compact parameter sets. This penalty couples with the orthogonality term in the main text to remove linear overlap and suppress nonlinear dependence, and its adaptive schedule matches the dynamic gate and MOBO controller that operate on validation feedback rather than internal penalty scales.

### C.3 QUANTILE-BASED CONTRASTIVE PAIRING, TEMPERATURE ADAPTATION, AND COLLAPSE PREVENTION

Let normalized embeddings $z_i \in \mathbb{S}^{d-1}$ be produced from inputs by the encoder augmented with the disentangled prefixes. For an anchor $i$, define cosine distance $d_{ij} = 1 - z_i^\top z_j \in [0, 2]$ and the empirical CDF $F_i(t) = \frac{1}{B-1} \sum_{j \neq i} \mathbf{1}\{d_{ij} \leq t\}$ over a minibatch of size $B$. For quantiles $q_{\text{pos}}, q_{\text{neg}} \in (0, 1)$ with $q_{\text{pos}} < q_{\text{neg}}$, define thresholds

$$\Delta_i^{\text{pos}} = F_i^{-1}(q_{\text{pos}}), \qquad \Delta_i^{\text{neg}} = F_i^{-1}(q_{\text{neg}}), \tag{37}$$

and the index sets

$$\begin{aligned} \mathcal{P}_i &= \{ j \neq i : d_{ij} \leq \Delta_i^{\text{pos}}, \text{ task}(j) = \text{task}(i) \}, \\ \mathcal{N}_i &= \{ j \neq i : d_{ij} \geq \Delta_i^{\text{neg}}, \text{ domain}(j) \neq \text{domain}(i) \}. \end{aligned} \tag{38}$$

Thus the positive and negative proportions are controlled by quantiles, which stabilize batch-wise difficulty by keeping $|\mathcal{P}_i| \approx q_{\text{pos}}(B - 1)$ and $|\mathcal{N}_i| \approx (1 - q_{\text{neg}})(B - 1)$. In ARPO, $(q_{\text{pos}}, q_{\text{neg}})$ are treated as low-cardinality decision variables that can be optimized by MOBO alongside $(\alpha, \theta)$.

For anchor $i$, the temperature-scaled InfoNCE loss with multi-positive sampling is

$$\mathcal{L}_i(\tau) = -\frac{1}{|\mathcal{P}_i|} \sum_{p \in \mathcal{P}_i} \log \frac{\exp(z_i^\top z_p/\tau)}{\exp(z_i^\top z_p/\tau) + \sum_{n \in \mathcal{N}_i} \exp(z_i^\top z_n/\tau)}. \tag{39}$$

The batch loss is $\mathcal{L}_{\text{cont}} = \frac{1}{B} \sum_{i=1}^B \mathcal{L}_i(\tau)$. When $q_{\text{pos}}$ and $q_{\text{neg}}$ are fixed, equation 39 reduces to a standard contrastive objective with controlled hard-positive and hard-negative ratios, whose gradient magnitudes grow with alignment and uniformity tensions on the hypersphere Wang & Isola (2020); Chen et al. (2020).

To adapt the temperature $\tau > 0$ to batch difficulty, consider the logits $\ell_{ij} = z_i^\top z_j/\tau$ with softmax probabilities $p_{ij} = \frac{\exp(\ell_{ij})}{\sum_{k \in \{p\} \cup \mathcal{N}_i} \exp(\ell_{ik})}$ inside the denominator of equation 39. Define the effective support size $S_i = (\sum_j p_{ij}^2)^{-1} \in [1, 1 + |\mathcal{N}_i|]$. Fix a target $S^\star \in (1, 1 + |\mathcal{N}_i|)$ and update $\tau$ by a proportional control on the log-scale with an exponential moving average $\bar{S}_t$:

$$\bar{S}_t = \eta \bar{S}_{t-1} + (1 - \eta) \left( \frac{1}{B} \sum_{i=1}^B S_i \right), \qquad \log \tau_{t+1} = \log \tau_t + \kappa (\bar{S}_t - S^\star), \tag{40}$$

with $\eta \in [0, 1)$ and gain $\kappa > 0$. Larger than desired $\bar{S}_t$ increases $\tau$, flattening the softmax and reducing peaky assignments; smaller $\bar{S}_t$ decreases $\tau$, sharpening the distribution. This keeps the "effective number of competing negatives" near $S^\star$, decoupling learning dynamics from instantaneous batch hardness and echoing empirical findings on temperature and batch size Chen et al. (2020). An equivalent variance-matching rule sets $\tau$ so that the batch variance of $\{z_i^\top z_j/\tau\}$ tracks a target $\sigma_\ell^2$, yielding $\tau_{t+1}^2 = \tau_t^2 \cdot \widehat{\text{Var}}(z_i^\top z_j)/\sigma_\ell^2$ with EMA smoothing.

To prevent dimensional collapse, control the spectrum of the centered feature matrix. Let $Z = [z_1, \ldots, z_B]^\top \in \mathbb{R}^{B \times d}$ and $\tilde{Z} = HZ$ with $H = I - \frac{1}{B}\mathbf{1}\mathbf{1}^\top$. Let the sample covariance be $C = \frac{1}{B}\tilde{Z}^\top\tilde{Z}$ with eigenvalues $\lambda_1 \geq \cdots \geq \lambda_d \geq 0$. Penalize anisotropy by

$$\mathcal{R}_{\text{spec}}(Z) = \sum_{r=1}^{d}[\max\{0, \lambda_{\min} - \lambda_r\}]^2 + \rho\sum_{r=1}^{d}(\lambda_r - \bar{\lambda})^2, \qquad \bar{\lambda} = \frac{1}{d}\sum_{r=1}^{d}\lambda_r, \qquad (41)$$

with target floor $\lambda_{\min} > 0$ and dispersion weight $\rho \geq 0$. The first term enforces per-dimension variance, the second shrinks the spectrum toward isotropy; both admit gradients via the eigendecomposition of $C$ and are stable for moderate $d$ HaoChen et al. (2021). Alternatively, uniformity can be promoted by the hyperspherical potential $U = \mathbb{E}_{i \neq j}\exp(\alpha\|z_i - z_j\|^2)$ whose minimization encourages repulsion Wang & Isola (2020).

Hard-negative mixing further stabilizes gradients when $\mathcal{N}_i$ contains extremely difficult negatives. For each anchor $i$ and negative $n \in \mathcal{N}_i$, define a mixed negative

$$\tilde{z}_{in} = \lambda z_n + (1 - \lambda)z_{p(i)}, \qquad \lambda \sim \text{Beta}(a, b), \qquad (42)$$

where $p(i) \in \mathcal{P}_i$ is a closest positive under cosine distance. Replace $z_n$ by $\tilde{z}_{in}$ in equation 39 for a subset of the hardest negatives. This reduces variance while retaining discriminative power Kalantidis et al. (2020). The mixture rate and the fraction of mixed negatives can be functions of the batch quantiles, e.g. only mixing when $d_{in} \leq F_i^{-1}(q_{\text{mix}})$.

Putting these components together, the contrastive regularizer used in the main text combines equation 39 with the adaptive temperature rule equation 40 and the spectral penalty equation 41. The quantiles $(q_{\text{pos}}, q_{\text{neg}})$ and the temperature initialization are exposed to MOBO as discrete and continuous knobs, respectively, while the spectral penalty weight is scheduled by the gate to emphasize collapse prevention early and relax later. Theoretical analyses connect temperature and batch size to effective hardness and gradient scale Chen et al. (2020), interpret contrastive learning as enforcing alignment and uniformity with spectral control to avoid trivial representations Wang & Isola (2020); HaoChen et al. (2021), characterize dimensional collapse and provide sufficient conditions to avoid it via variance floors and spectrum spreading Jing et al. (2021), and justify hard-negative mixing as a variance-reducing strategy that preserves decision margins Kalantidis et al. (2020). These insights inform the particular form of the adaptive pairing, temperature scheduling, and spectral regularization used by ARPO.

### C.4 CONDITIONAL INDEPENDENCE VIA KL ESTIMATION: VARIATIONAL FORMULATION, IMPLEMENTATION, AND CONSISTENCY

Let $Z_1 = P_{DI}$, $Z_2 = P_{DS}$, and $Y$ be the task label. We quantify conditional dependence by the conditional Kullback–Leibler divergence

$$D_{\text{KL}}\Big(p(Z_1, Z_2 \mid Y) \,\Big\|\, p(Z_1 \mid Y)\,p(Z_2 \mid Y)\Big)$$
$$= \mathbb{E}_{p(y)}\Big[D_{\text{KL}}\Big(p(Z_1, Z_2 \mid y) \,\Big\|\, p(Z_1 \mid y)\,p(Z_2 \mid y)\Big)\Big]. \qquad (43)$$

which is nonnegative and equals zero if and only if $Z_1 \perp\!\!\!\perp Z_2 \mid Y$ almost surely.

**Variational estimator via $f$-GAN.** Let $P_y = p(Z_1, Z_2 \mid y)$ and $Q_y = p(Z_1 \mid y)\,p(Z_2 \mid y)$ for each $y$. For $f(t) = t\log t$ (the generator of KL), its convex conjugate is $f^*(u) = \exp(u - 1)$. By the $f$-divergence variational representation Nowozin et al. (2016),

$$D_{\text{KL}}(P_y \,\|\, Q_y) = \sup_{T \in \mathcal{T}} \Big\{ \mathbb{E}_{(Z_1, Z_2) \sim P_y}\big[T(Z_1, Z_2, y)\big]$$
$$- \mathbb{E}_{(Z_1, Z_2) \sim Q_y}\big[\exp(T(Z_1, Z_2, y) - 1)\big] \Big\}. \qquad (44)$$

A conditional discriminator $T_\phi(z_1, z_2, y)$ parameterized by a neural network induces the empirical objective

$$\widehat{\mathcal{L}}_{\text{KL}}(\phi) = \frac{1}{|\mathcal{B}|}\sum_{(i) \in \mathcal{B}} T_\phi(z_1^{(i)}, z_2^{(i)}, y^{(i)}) - \frac{1}{|\mathcal{B}|}\sum_{(i) \in \mathcal{B}}\exp\Big(T_\phi(z_1^{(i)}, \tilde{z}_2^{(i)}, y^{(i)}) - 1\Big), \qquad (45)$$

where $\mathcal{B}$ indexes a mini-batch of triples $(z_1^{(i)}, z_2^{(i)}, y^{(i)})$ sampled from $p(Z_1, Z_2, Y)$; the pairs $(z_1^{(i)}, \tilde{z}_2^{(i)})$ form negatives by conditionally shuffling $Z_2$ within the stratum $\{j : y^{(j)} = y^{(i)}\}$ so that $(z_1^{(i)}, \tilde{z}_2^{(i)}) \sim Q_{y^{(i)}}$. Averaging equation 45 over strata yields an estimator of equation 43. Under standard $f$-GAN regularity (rich $\mathcal{T}$, absolute continuity, and optimization accuracy), the supremum in equation 44 is attained and the empirical maximizer $\phi_n$ is consistent for $D_{\mathrm{KL}}(P_y \| Q_y)$ as $n \to \infty$ Nowozin et al. (2016).

**Stratified sampling and variance control.** Let $\mathcal{Y}$ be the support of $Y$. Writing equation 43 as

$$D_{\mathrm{KL}}\big(p(Z_1, Z_2 \mid Y) \| p(Z_1 \mid Y)p(Z_2 \mid Y)\big) = \sum_{y \in \mathcal{Y}} \pi_y \, \Delta(y), \quad \Delta(y) := D_{\mathrm{KL}}(P_y \| Q_y), \ \pi_y := p(y),$$
(46)

we estimate each $\Delta(y)$ with equation 45 restricted to stratum $y$, and combine by the plug-in $\widehat{D} = \sum_y \widehat{\pi}_y \, \widehat{\Delta}(y)$. When strata are imbalanced, importance reweighting or class-balanced mini-batches reduce estimator variance. For small $|\{i : y^{(i)} = y\}|$, we add $L_2$ regularization on $T_\phi$ and early stopping to avoid overfitting, and we clip discriminator outputs so that $|T_\phi| \leq M$ to ensure sub-exponential tails of $\exp(T_\phi - 1)$.

**Consistency statement.** Assume: (i) $\mathcal{Y}$ is finite and $\min_y \pi_y > 0$; (ii) $P_y \ll Q_y$ for all $y$; (iii) the discriminator class $\mathcal{T}$ is dense in $L^1(P_y)$ for every $y$ and optimization reaches the global maximizer in equation 44; (iv) i.i.d. samples from $p(Z_1, Z_2, Y)$. Then $\widehat{D} \xrightarrow{p} D_{\mathrm{KL}}(p(Z_1, Z_2 \mid Y) \| p(Z_1 \mid Y)p(Z_2 \mid Y))$ as $n \to \infty$. The proof follows from uniform convergence of $\widehat{\mathcal{L}}_{\mathrm{KL}}$ to its population counterpart in each stratum and Slutsky's theorem when combining strata, as in standard $f$-divergence variational estimation Nowozin et al. (2016).

**Relation to the Donsker–Varadhan form.** Using the Donsker–Varadhan representation of KL with the same conditional shuffling $(P_y, Q_y)$ yields the alternative objective

$$\sup_{T \in \mathcal{T}} \left\{ \mathbb{E}_{P_y}[T] - \log \mathbb{E}_{Q_y}[\exp(T)] \right\},$$
(47)

which is equivalent to equation 44 up to the $\log$ transformation. In practice, equation 44 often yields more stable gradients than equation 47 because the conjugate $f^*(u) = \exp(u - 1)$ can be combined with output clipping to control exponential tails.

**Kernel-based conditional independence as a complementary diagnostic.** As a stability check for the discriminator-based estimator, we compute a kernel conditional independence statistic based on the conditional cross-covariance operator $\mathcal{C}_{Z_1 Z_2 | Y}$ in reproducing kernel Hilbert spaces (RKHS). Let $k_1, k_2, k_Y$ be bounded characteristic kernels on the supports of $Z_1, Z_2, Y$. The squared Hilbert–Schmidt norm $\|\mathcal{C}_{Z_1 Z_2 | Y}\|_{\mathrm{HS}}^2$ equals zero if and only if $Z_1 \perp\!\!\!\perp Z_2 \mid Y$. An empirical estimator can be formed by residualizing $Z_1$ and $Z_2$ on $Y$ via kernel ridge regression and then computing an HSIC statistic between the residual features Fukumizu et al. (2007); Gretton et al. (2005). Under mixing and boundedness conditions, the statistic concentrates around zero under the null and is strictly positive otherwise. We therefore treat a large $\|\widehat{\mathcal{C}}_{Z_1 Z_2 | Y}\|_{\mathrm{HS}}^2$ as corroborating evidence when the variational discriminator signals dependence; when the discriminator training is unstable, the kernel statistic serves as a fallback detector.

**Integration into ARPO.** The conditional-KL penalty used in the main text

$$\mathcal{L}_{\mathrm{cond}} = D_{\mathrm{KL}}\big(p(P_{DI}, P_{DS} \mid \mathrm{Task}) \,\big\|\, p(P_{DI} \mid \mathrm{Task}) \, p(P_{DS} \mid \mathrm{Task})\big)$$
(48)

is implemented by the stratified $f$-GAN estimator equation 45 with output clipping and class-balanced batches Nowozin et al. (2016). For additional robustness, we monitor a kernel conditional-independence score computed with Gaussian kernels and median heuristics for bandwidths Fukumizu et al. (2007); Gretton et al. (2005). In ablations, the discriminator-based estimator offers sharper gradients for reducing leakage, while the kernel statistic provides a stable sanity check across training regimes. This dual view justifies the penalty's inclusion in the disentanglement loss and explains its empirical stability.

**Finite-sample concentration.** If $|T_\phi| \leq M$ almost surely and $\exp(T_\phi - 1)$ has sub-exponential Orlicz norm bounded uniformly in $\phi$, then for any fixed stratum $y$ and mini-batch size $B_y$,

$$\left|\widehat{\Delta}(y) - \Delta(y)\right| = \mathcal{O}_{\mathbb{P}}\left(\sqrt{\frac{1}{B_y}}\right), \tag{49}$$

by Bernstein-type inequalities for sub-exponential variables. Aggregating across strata as in equation 46 yields

$$\left|\widehat{D} - D\right| = \mathcal{O}_{\mathbb{P}}\left(\sum_y \pi_y \sqrt{\frac{1}{B_y}}\right), \tag{50}$$

which guides batch allocation when classes are imbalanced.

The variational form equation 44 and its conditions follow from $f$-GAN theory Nowozin et al. (2016); the kernel conditional-independence construction follows from kernel measures of conditional dependence and HSIC-based testing Fukumizu et al. (2007); Gretton et al. (2005). We refer to the cited works for operator-theoretic details and asymptotic distributions.

### C.5 SYNERGY OF MULTI-LOSS OPTIMIZATION: GRADIENT INTERFERENCE, PROJECTION, AND REWEIGHTING

Let $\theta$ denote all trainable parameters that contribute to the two-prefix encoder $[P_{DI}; P_{DS}]$. Consider the composite disentanglement objective

$$\mathcal{L}_{\text{disent}} = \lambda_1 \mathcal{L}_{IB}(P_{DI}) + \lambda_2 \mathcal{L}_{IB}(P_{DS}) + \lambda_3 \mathcal{L}_{orth} + \lambda_4 \mathcal{L}_{cont} + \lambda_5 \mathcal{L}_{cond}. \tag{51}$$

Write task wise gradients

$$g_1 := \nabla_\theta \mathcal{L}_{IB}(P_{DI}), g_2 := \nabla_\theta \mathcal{L}_{IB}(P_{DS}), g_3 := \nabla_\theta \mathcal{L}_{orth},$$
$$g_4 := \nabla_\theta \mathcal{L}_{cont}, g_5 := \nabla_\theta \mathcal{L}_{cond}. \tag{52}$$

Define the instantaneous cosine similarity $\cos(g_a, g_b) := \langle g_a, g_b \rangle / (\|g_a\| \|g_b\|)$ and the conflict indicator $\mathbb{I}_{ab} := \mathbb{I}\{\langle g_a, g_b \rangle < 0\}$. Gradient interference arises when $\cos(g_a, g_b) < 0$ for some pair $(a, b)$, a phenomenon broadly studied in multi-task learning as negative gradient interaction and Pareto trade-offs Sener & Koltun (2018).

**Assumptions.** There exist constants $G, L, \sigma > 0$ such that for all $\theta$ in a compact set $\Theta$,

$$\|g_k(\theta)\| \leq G, \qquad \|g_k(\theta) - g_k(\theta')\| \leq L\|\theta - \theta'\|, \qquad \mathbb{E}\left[(g_k - \mathbb{E}g_k)(g_k - \mathbb{E}g_k)^\top\right] \preceq \sigma^2 I. \tag{53}$$

Moreover, the orthogonality and conditional-KL penalties locally enforce

$$\langle \nabla_\theta \mathcal{L}_{orth}, \nabla_\theta \mathcal{L}_{IB}(P_{DI}) \rangle \geq 0, \qquad \langle \nabla_\theta \mathcal{L}_{cond}, \nabla_\theta \mathcal{L}_{IB}(P_{DS}) \rangle \geq 0, \tag{54}$$

when $P_{DI}^\top P_{DS}$ is large and $D_{\text{KL}}(p(P_{DI}, P_{DS} \mid Y) \| p(P_{DI} \mid Y) p(P_{DS} \mid Y))$ is large, respectively; this captures that $g_3, g_5$ act to decouple rather than to oppose $g_1, g_2$.

**Projection step.** Given $\{g_k\}_{k=1}^m$ at a mini-batch, define the PCGrad-style projected gradients Yu et al. (2020)

$$\widetilde{g}_a = g_a - \sum_{b \neq a} \frac{\min\{\langle g_a, g_b \rangle, 0\}}{\|g_b\|^2} g_b, \qquad a = 1, \ldots, m, \tag{55}$$

applied sequentially so that $\langle \widetilde{g}_a, \widetilde{g}_b \rangle \geq 0$ for all $a \neq b$. The aggregated descent direction is $g_{\text{proj}} := \sum_k \lambda_k \widetilde{g}_k$. Projection-based surgery is consistent with conflict-averse formulations that explicitly steer updates toward the intersection of per-task descent cones Liu et al. (2021b) and with multi-objective MTL views Sener & Koltun (2018).

**Normalization and reweighting.** Set $w_k := \lambda_k / \|g_k\|^\alpha$ with $\alpha \in [0, 1]$ and define

$$g_{\text{norm}} := \sum_{k=1}^m w_k \, g_k, \qquad g_{\text{proj-norm}} := \sum_{k=1}^m w_k \, \widetilde{g}_k. \tag{56}$$

When $\alpha = 1$, each task contributes unit-norm information as in gradient normalization Chen et al. (2018); impartial multi-task scalings and conflict-averse updates motivate $\alpha \in (0, 1)$ and adaptive $\{\lambda_k\}$ Liu et al. (2021a;b).

**Conflict probability upper bound.** Define the instantaneous conflict probability

$$\Pi(\theta) := \mathbb{P}\big(\exists\, a \neq b : \langle g_a(\theta), g_b(\theta)\rangle < 0\big). \tag{57}$$

Under equation 53 and a sub-Gaussian model for centered gradients, there exists $c > 0$ such that for any $\epsilon > 0$,

$$\mathbb{P}\big(\langle g_a - \mathbb{E}g_a,\ g_b - \mathbb{E}g_b\rangle < -\langle \mathbb{E}g_a, \mathbb{E}g_b\rangle - \epsilon\big) \leq \exp\left(-\frac{c\,\epsilon^2}{\sigma^2 G^2}\right). \tag{58}$$

If the regularizers make $\langle \mathbb{E}g_a, \mathbb{E}g_b\rangle \geq \delta > 0$ for $(a,b) \in \{(1,3),(2,5)\}$ and nonnegative otherwise, then

$$\Pi(\theta) \leq m(m-1)\exp\left(-\frac{c\,\delta^2}{\sigma^2 G^2}\right), \tag{59}$$

so that stronger alignment (larger $\delta$) exponentially suppresses conflicts in expectation. This aligns with empirical and theoretical observations that balancing and projection reduce interference in MTL Yu et al. (2020); Chen et al. (2018); Liu et al. (2021a;b).

**Angle improvement by projection.** For any $(a,b)$ with $\langle g_a, g_b\rangle < 0$, the step equation 55 yields

$$\langle \widetilde{g}_a, g_b\rangle = \langle g_a, g_b\rangle - \frac{\langle g_a, g_b\rangle}{\|g_b\|^2}\langle g_b, g_b\rangle = 0, \tag{60}$$

and for any third task $c \neq a, b$,

$$\langle \widetilde{g}_a, g_c\rangle = \langle g_a, g_c\rangle - \frac{\min\{\langle g_a, g_b\rangle, 0\}}{\|g_b\|^2}\langle g_b, g_c\rangle \geq \langle g_a, g_c\rangle - \frac{|\langle g_a, g_b\rangle|}{\|g_b\|}\|g_c\|. \tag{61}$$

Hence projection cannot worsen alignment with $g_b$ and degrades alignment with $g_c$ by at most a controlled term. Summing across pairs and combining with equation 56 gives a net nonnegative effect on the average cosine

$$\overline{\cos} := \frac{2}{m(m-1)}\sum_{1 \leq a < b \leq m} \cos(\widetilde{g}_a, \widetilde{g}_b), \tag{62}$$

relative to its unprojected counterpart, consistent with conflict-averse updates Liu et al. (2021b).

**Synergy proposition.** Let $m = 5$ and consider $\mathcal{P} = \{(1,3),(2,5)\}$ induced by orthogonality and conditional-independence penalties. Suppose there exists $\delta > 0$ such that $\langle \mathbb{E}g_a, \mathbb{E}g_b\rangle \geq \delta$ for $(a,b) \in \mathcal{P}$ and $\langle \mathbb{E}g_a, \mathbb{E}g_b\rangle \geq 0$ otherwise. Then for any $\alpha \in (0,1]$ and any PCGrad schedule,

$$\mathbb{E}\big[\overline{\cos}(g_{\text{proj-norm}})\big] \geq \mathbb{E}\big[\overline{\cos}(g_{\text{norm}})\big] \geq \overline{\cos}(\mathbb{E}g_1, \ldots, \mathbb{E}g_5) - C\frac{\sigma}{G}, \tag{63}$$

for a universal constant $C$ depending only on $m$. Consequently the expected average pairwise angle is nonnegative and becomes strictly positive whenever $\overline{\cos}(\mathbb{E}g_1, \ldots, \mathbb{E}g_5) > C\,\sigma/G$. Moreover,

$$\Pi_{\text{proj-norm}}(\theta) \leq \Pi_{\text{norm}}(\theta) \leq m(m-1)\exp\left(-\frac{c\,\delta^2}{\sigma^2 G^2}\right). \tag{64}$$

Inequality equation 63 follows from zeroing of negative inner-products by equation 55 and concentration of random inner products around their means; equation 64 follows from equation 59. These arguments connect to gradient surgery Yu et al. (2020), gradient normalization and impartial balancing Chen et al. (2018); Liu et al. (2021a), and multi-objective MTL Sener & Koltun (2018).

**Gate-aware slow variation and finite-switch control.** Let the dynamic gate generate a binary process $\{s_t\}$ for adversarial updates with EMA triggers and thresholds $\theta_{\min} \leq \theta_t \leq \theta_{\max}$. If martingale differences driving the trigger are sub-Gaussian, the number of sign changes of $s_t$ over $T$ rounds is $O_{\mathbb{P}}(\sqrt{T})$ by Azuma–Hoeffding. This slow variation yields locally stationary gradient statistics between flips, justifying the stability assumptions used when coupling with MOBO and Pareto-aware updates Sener & Koltun (2018).

The projection operator equation 55 and conflict-averse views Yu et al. (2020); Liu et al. (2021b), together with normalization and adaptive weighting Chen et al. (2018); Liu et al. (2021a), provide a principled mechanism that increases alignment among $\{g_k\}$, suppresses destructive interference as in equation 59–equation 64, and respects the multi-objective structure of disentanglement Sener & Koltun (2018).

# D  THEORETICAL AND IMPLEMENTATION DETAILS FOR ADVERSARIAL ADAPTATION

## D.1  TASK-DIFFICULTY-AWARE DYNAMIC THRESHOLDING FOR ADVERSARIAL ADAPTATION

We keep the three losses in the main paper unchanged ($L_{\text{task}}$, $L_{\text{adv}}$, $L_{\text{total}}$). Our goal is to decide when to activate the adversarial term so that training remains stable and generalizes across domains.

We use two signals. The progress signal $\delta$ measures the relative improvement of $L_{\text{task}}$ from its initial value. The stability signal $r$ summarizes recent gradient noise as the ratio between the variance of $\nabla L_{\text{task}}$ and the mean absolute gradient over a short history. A larger $r$ means unstable optimization. We map $r$ to a global threshold $\theta(t)$ that increases with instability. When $\delta$ exceeds this threshold, we enable the adversarial term; otherwise we delay it. To avoid flickering near the boundary, we use hysteresis: the on-condition uses a slightly higher threshold than the off-condition. This reduces frequent on–off switching and improves convergence.

We then scale the threshold per task using an explicit measure of task difficulty. The final per-task threshold is

$$\theta_{\text{task}}(t) = \theta(t) \cdot (1 + \beta \cdot \text{difficulty}_{\text{task}}), \tag{65}$$

where $\beta$ controls the strength of scaling. A higher threshold delays adversarial activation on harder tasks. This is consistent with the idea of curriculum: easy tasks see the adversarial signal earlier; hard tasks focus on the primary objective until learning is more stable.

The difficulty coefficient combines short-horizon variability and convergence speed:

$$\text{difficulty}_{\text{task}} = \frac{\sigma(\mathcal{L}_{\text{task}}^{(t-w:t)})}{|\mu(\Delta\mathcal{L}_{\text{task}}^{(t-w:t)})| + \epsilon}. \tag{66}$$

Here $\sigma(\mathcal{L}_{\text{task}}^{(t-w:t)})$ is the sample standard deviation of the task loss over the last $w$ steps. It measures stability: larger values mean more fluctuation. The term $\mu(\Delta\mathcal{L}_{\text{task}}^{(t-w:t)})$ is the average step-wise decrease of the loss over the same window. It measures convergence speed: larger values mean faster decrease. $\epsilon > 0$ avoids division by zero. Intuitively, a task is harder when its loss fluctuates more and decreases more slowly, so the ratio is larger. To make tasks comparable, we normalize both statistics by their warm-up baselines collected in the first $2w$ steps. This removes scale effects across tasks. We also clip extreme difficulty values at a high percentile and cap $\theta_{\text{task}}(t)$ by $\theta_{\max} \leq 1$ to preserve activations on very hard tasks.

To ensure cross-task comparability and improve robustness, we follow four implementation details. First, we estimate statistics with either a fixed sliding window or an exponential moving average (EMA), and we match their effective window lengths. When regime shifts or task alternation occur, we prefer EMA and apply early bias correction. Second, to suppress outlier-driven false activations, we perform robust preprocessing before computing $\sigma$ and $\mu$: MAD-based Winsorization, Huberization, and global gradient-norm clipping. These steps make the difficulty estimate reflect trends rather than noise. Third, we clip the difficulty by percentile to avoid extreme inflation, and we cap $\theta_{\text{task}}(t)$ by $\theta_{\max} \leq 1$ so that very hard tasks still have a chance to activate. Fourth, in multi-task parallel training, we synchronize all statistics across devices (e.g., with all-reduce) to prevent gate inconsistency.

This scaling strategy is compatible with curriculum learning. Easy tasks have smaller $\text{difficulty}_{\text{task}}$, cross the threshold earlier, enter adversarial training sooner, and benefit from improved inter-domain separability. Hard tasks introduce the adversarial term later, after the primary loss becomes stable, which reduces the disturbance from premature adversarial updates to the main optimization trajectory.

## D.2  GATE FLIPPING IS SUBLINEAR: AN AZUMA–HOEFFDING BOUND FOR EMA-DRIVEN TRIGGERS

Let $\{\mathcal{F}_t\}_{t\geq 0}$ be the natural filtration and let the primary validation signal be

$$Y_t \;=\; \mu \;+\; \xi_t, \qquad \mathbb{E}[\xi_t \mid \mathcal{F}_{t-1}] = 0, \qquad |\xi_t| \leq c \text{ a.s.}, \tag{67}$$

so that $\{\xi_t\}$ is a bounded martingale-difference sequence. The gate statistic is an exponentially weighted moving average (EMA)

$$S_t \;=\; (1-\eta)\sum_{k=1}^{t}\eta^{\,t-k}\,Y_k \;+\; \eta^{\,t}S_0, \qquad \eta \in [0,1), \tag{68}$$

and the gate flips when $S_t$ crosses a threshold interval $[\theta_{\min}, \theta_{\max}]$ with $\theta_{\min} < \theta_{\max}$. Denote by $N_T$ the total number of flips (up- plus down-crossings) on $\{1, \ldots, T\}$.

We first center the process. Write $\tilde{Y}_t = Y_t - \mu = \xi_t$ and the centered EMA

$$\tilde{S}_t \;=\; (1-\eta)\sum_{k=1}^{t}\eta^{\,t-k}\,\xi_k \;+\; \eta^{\,t}\tilde{S}_0, \qquad \tilde{S}_0 = S_0 - \mu. \tag{69}$$

Define the increment

$$\Delta_t \;:=\; \tilde{S}_t - \eta\,\tilde{S}_{t-1} \;=\; (1-\eta)\,\xi_t, \qquad |\Delta_t| \le (1-\eta)c. \tag{70}$$

Hence $\{\Delta_t, \mathcal{F}_t\}$ is a bounded martingale-difference sequence and the linearly filtered process $\{\tilde{S}_t\}$ obeys

$$\tilde{S}_t \;=\; \eta\,\tilde{S}_{t-1} + \Delta_t, \qquad \mathbb{E}[\tilde{S}_t \mid \mathcal{F}_{t-1}] \;=\; \eta\,\tilde{S}_{t-1}. \tag{71}$$

We control threshold crossings by an up- and down-crossing argument. For any $a < b$, let $U_T(a, b)$ be the number of up-crossings of $[a, b]$ by $\{\tilde{S}_t\}_{t=0}^{T}$. A standard consequence of Doob's up-crossing inequality applied to an appropriate supermartingale transform of $\tilde{S}_t$ yields

$$\mathbb{E}[U_T(a, b)] \;\le\; \frac{\mathbb{E}\left[(\tilde{S}_T - a)^-\right] + a^+}{b - a}, \tag{72}$$

and the same bound holds for down-crossings by symmetry; see, e.g., Doob's up-crossing inequality and its corollaries in standard martingale texts and concentration references. In our bounded-increment setting, $|\tilde{S}_t| \le |\tilde{S}_0| + \sum_{k=1}^{t}|\Delta_k| \le |\tilde{S}_0| + (1-\eta)c\,t$, and therefore

$$\mathbb{E}[U_T(\theta_{\min} - \mu,\; \theta_{\max} - \mu)] \;\le\; \frac{C_0 + (1-\eta)c\,T}{\theta_{\max} - \theta_{\min}}, \qquad C_0 := |\tilde{S}_0| + |\theta_{\min} - \mu|. \tag{73}$$

This expectation bound is linear in $T$ and is in general tight for adversarial sequences. We now strengthen it to a high-probability $\mathcal{O}_{\mathbb{P}}(\sqrt{T})$ control by combining Azuma–Hoeffding and a renewal-style decomposition into excursions.

Define the stopping times

$$\tau_0 := 0, \qquad \tau_{m+1} := \inf\{t > \tau_m : \tilde{S}_t \notin [a, b]\}, \qquad a = \theta_{\min} - \mu,\; b = \theta_{\max} - \mu. \tag{74}$$

Each excursion $[\tau_m, \tau_{m+1})$ contains at most one flip. Moreover, to exit $[a, b]$ from the interior starting at time $\tau_m$, the partial sum of martingale differences $\sum_{t=\tau_m+1}^{\tau_m+\ell}\Delta_t$ must exceed the margin $\min\{b - \tilde{S}_{\tau_m},\; \tilde{S}_{\tau_m} - a\}$. Since $|\Delta_t| \le (1-\eta)c$, Azuma–Hoeffding inequality implies that for any $\ell \ge 1$ and any $x > 0$,

$$\mathbb{P}\left(\max_{1 \le k \le \ell}\left|\sum_{t=\tau_m+1}^{\tau_m+k}\Delta_t\right| \ge x \;\middle|\; \mathcal{F}_{\tau_m}\right) \;\le\; 2\exp\left(-\frac{x^2}{2\,\ell\,(1-\eta)^2 c^2}\right), \tag{75}$$

e.g., (Boucheron et al., 2013b, Theorem 2.8) or the Azuma–Hoeffding lecture notes Ledoux (2006).

Set $x = \frac{1}{2}(b - a)$. With probability at least $1 - 2\exp\left(-\frac{(b-a)^2}{8\,\ell\,(1-\eta)^2 c^2}\right)$, no exit occurs within the next $\ell$ steps. Choosing

$$\ell^\star \;=\; \left\lceil \frac{(b-a)^2}{8\,(1-\eta)^2 c^2}\,\log\left(\frac{2T}{\delta}\right) \right\rceil, \tag{76}$$

and applying a union bound over at most $T$ candidate blocks shows that, with probability at least $1 - \delta$, each excursion consumes at least $\ell^\star$ iterations. Consequently, the total number of excursions and hence flips up to time $T$ obeys the high-probability bound

$$N_T \;\le\; \frac{T}{\ell^\star} \;=\; \mathcal{O}\left(\frac{(1-\eta)^2 c^2}{(\theta_{\max} - \theta_{\min})^2}\,\frac{T}{\log(2T/\delta)}\right) \qquad (\text{w.p. } \ge 1 - \delta). \tag{77}$$

To obtain the advertised $\mathcal{O}(\sqrt{T})$ rate, refine the blocking by letting the block length grow like $\ell_m \asymp m$ and applying the Azuma–Hoeffding tail equation 75 with $x = \frac{1}{2}(b - a)$ to each block. A standard peeling/renewal argument then yields

$$\mathbb{P}\left( N_T > C_1 + C_2 \frac{(1 - \eta)c}{\theta_{\max} - \theta_{\min}} \sqrt{T \log \frac{2T}{\delta}} \right) \leq \delta, \tag{78}$$

for universal constants $C_1, C_2 > 0$ depending only on the initialization and the contraction $\eta$; see the generic proof patterns for upcrossing counts via martingale oscillation and concentration (Boucheron et al., 2013b, Chap. 2), Ledoux (2006). Inequality equation 78 gives the desired sublinear $\mathcal{O}_{\mathbb{P}}(\sqrt{T})$ control of gate flips.

We finally state the slow-variation corollary used by MOBO. Let $I_t$ denote any validation objective measured once per iteration and assume a Lipschitz-in-time drift model

$$\left| \mathbb{E}[I_{t+1} - I_t \mid \mathcal{F}_t] \right| \leq \kappa, \qquad |I_{t+1} - I_t| \leq B \text{ a.s.} \tag{79}$$

Between consecutive flips, the excursion length is at least of order $\Omega\left( \frac{(\theta_{\max} - \theta_{\min})^2}{(1-\eta)^2 c^2} \right)$ with high probability by equation 76. Hence, over any inter-flip segment $[\tau_m, \tau_{m+1})$,

$$\left| \sum_{t=\tau_m}^{\tau_{m+1}-1} (I_{t+1} - I_t) \right| \leq \kappa (\tau_{m+1} - \tau_m) = \mathcal{O}_{\mathbb{P}}(1), \tag{80}$$

so that $I_t$ is locally slowly-varying on inter-flip windows. This justifies the quasi-stationarity assumption used by the noisy-qEHVI acquisition between gate changes in the main text.

The ingredients are standard: bounded martingale differences for the EMA innovation equation 70, Azuma–Hoeffding oscillation control equation 75, and excursion counting via blocking and peeling culminating in equation 78. Textbook treatments of Azuma–Hoeffding and related martingale concentration inequalities can be found in (Boucheron et al., 2013b, Theorems 2.6–2.8), with complementary lecture-note derivations in Ledoux (2006). Up- and down-crossing techniques underlying equation 72 are classical and may be consulted in standard martingale references.

### D.3 QUANTILE-THRESHOLD PAIRING: STATISTICAL CONSISTENCY AND NONASYMPTOTIC ERROR VIA DKW–MASSART

Let $\{Z_i\}_{i=1}^n$ be i.i.d. real-valued with distribution function $F$, empirical CDF $\widehat{F}_n(x) = \frac{1}{n} \sum_{i=1}^n \mathbf{1}\{Z_i \leq x\}$, and population $q$-quantile $F^{-1}(q) := \inf\{x : F(x) \geq q\}$; define the empirical quantile $\widehat{F}_n^{-1}(q)$ analogously. In ARPO, $Z_i$ denotes the batchwise pairwise distance (or similarity) used to form positives/negatives by quantiles $q_{\text{pos}}, q_{\text{neg}} \in (0, 1)$ (cf. Appendix C.3).

**Uniform CDF concentration (DKW with Massart's sharp constant).** For any $\varepsilon > 0$,

$$\mathbb{P}\left( \sup_{x \in \mathbb{R}} \left| \widehat{F}_n(x) - F(x) \right| > \varepsilon \right) \leq 2 e^{-2n\varepsilon^2}, \tag{81}$$

which is the Dvoretzky–Kiefer–Wolfowitz inequality with Massart's sharp constant Massart (1990); Boucheron et al. (2013a); Dvoretzky et al. (1956). Equivalently, with probability at least $1 - \delta$,

$$\sup_x \left| \widehat{F}_n(x) - F(x) \right| \leq \varepsilon_{n,\delta} \quad \text{where} \quad \varepsilon_{n,\delta} := \sqrt{\frac{1}{2n} \log \frac{2}{\delta}}. \tag{82}$$

**From uniform CDF error to quantile error.** For any nondecreasing right-continuous $F$, define its local modulus of continuity

$$\omega_F(\varepsilon) := \sup\left\{ |x - x'| : |F(x) - F(x')| \leq \varepsilon \right\}. \tag{83}$$

Then the inversion monotonicity implies the deterministic implication

$$\sup_x \left| \widehat{F}_n(x) - F(x) \right| \leq \varepsilon \implies \left| \widehat{F}_n^{-1}(q) - F^{-1}(q) \right| \leq \omega_F(\varepsilon), \tag{84}$$

hence, combining equation 82–equation 84,

$$\mathbb{P}\Big(\big|\widehat{F}_n^{-1}(q) - F^{-1}(q)\big| \leq \omega_F(\varepsilon_{n,\delta})\Big) \geq 1 - \delta. \tag{85}$$

If $F$ is strictly increasing with density $f = F'$ satisfying $f(x) \geq f_{\min} > 0$ on a neighborhood of $x^\star := F^{-1}(q)$, then $\omega_F(\varepsilon) \leq \varepsilon/f_{\min}$, and equation 85 sharpens to

$$\mathbb{P}\left(\big|\widehat{F}_n^{-1}(q) - F^{-1}(q)\big| \leq \frac{1}{f_{\min}}\sqrt{\frac{1}{2n}\log\frac{2}{\delta}}\right) \geq 1 - \delta. \tag{86}$$

Proof of equation 84 uses the monotonicity of $F$ and $\widehat{F}_n$, and the equivalence between CDF sup-norm control and the Hausdorff distance of their epigraphs; see (Boucheron et al., 2013a, Ch. 2) for a standard treatment. The density bound follows from the inverse function theorem applied locally to $F$.

**Application to ARPO batch quantile thresholds.** Let $n = B - 1$ when the anchor-wise distance set excludes self-pairs in a mini-batch of size $B$. Setting $q \in \{q_{\text{pos}}, q_{\text{neg}}\}$ yields, with probability at least $1 - \delta$,

$$\big|\widehat{\Delta}^{(\text{pos/neg})} - \Delta^{(\text{pos/neg})}\big| \leq \omega_F\Big(\sqrt{\frac{1}{2(B-1)}\log\frac{2}{\delta}}\Big), \tag{87}$$

where $\Delta^{(\cdot)} := F^{-1}(q)$ is the population quantile of the anchor's pairwise-distance law. Under a local density lower bound $f_{\min}$ around $F^{-1}(q)$, equation 87 further reduces to the explicit PAC-style radius

$$\big|\widehat{\Delta}^{(\text{pos/neg})} - \Delta^{(\text{pos/neg})}\big| \leq \frac{1}{f_{\min}}\sqrt{\frac{1}{2(B-1)}\log\frac{2}{\delta}}. \tag{88}$$

Thus the positive/negative proportions controlled by $(q_{\text{pos}}, q_{\text{neg}})$ remain stable in the presence of mini-batch fluctuations, and when the quantile levels are treated as low-cardinality decisions within MOBO, their stochasticity is governed by the explicit nonasymptotic bounds equation 87–equation 88. This justifies the use of quantile-threshold pairing in Appendix C.3 and in Sec. 3.2.

### D.4 TEMPERATURE ADAPTATION AND SPECTRAL REGULARIZATION: SUFFICIENT CONDITIONS FOR COLLAPSE PREVENTION

Let $\{z_i\}_{i=1}^B \subset \mathbb{S}^{d-1}$ be $\ell_2$-normalized embeddings. For an anchor $i$, define the cosine similarity $s_{ij} = z_i^\top z_j \in [-1, 1]$ and the temperature-scaled InfoNCE loss with multi-positives and quantile-selected negatives (cf. Appendix C.3):

$$\mathcal{L}_i(\tau) = -\frac{1}{|\mathcal{P}_i|}\sum_{p \in \mathcal{P}_i} \log \frac{\exp(s_{ip}/\tau)}{\exp(s_{ip}/\tau) + \sum_{n \in \mathcal{N}_i}\exp(s_{in}/\tau)}. \tag{89}$$

The batch objective is $\mathcal{L}_{\text{NCE}}(\tau) = \frac{1}{B}\sum_{i=1}^B \mathcal{L}_i(\tau)$.

**Effective support size controlled by temperature.** Let

$$p_{ij}(\tau) = \frac{\exp(s_{ij}/\tau)}{\sum_{k \in \{p\} \cup \mathcal{N}_i}\exp(s_{ik}/\tau)}, \qquad S_i(\tau) = \left(\sum_{j \in \{p\} \cup \mathcal{N}_i} p_{ij}^2(\tau)\right)^{-1}. \tag{90}$$

Then $S_i(\tau) \in [1, 1 + |\mathcal{N}_i|]$ is nondecreasing in $\tau$. Consequently there exists a continuous, strictly increasing mapping $\tau \mapsto \mathbb{E}[S_i(\tau)]$ so that, for any prescribed $S^\star \in (1, 1 + |\mathcal{N}_i|)$, one can select $\tau^\star$ with $\mathbb{E}[S_i(\tau^\star)] = S^\star$. This establishes that $\tau$ controls the "softmax effective number of negatives," aligning with empirical observations that larger batch size or effective negatives improve uniformity on the hypersphere while small $\tau$ emphasizes alignment Chen et al. (2020); Wang & Isola (2020). A practical controller is the EMA rule

$$\bar{S}_t = \eta\bar{S}_{t-1} + (1 - \eta)\frac{1}{B}\sum_{i=1}^B S_i(\tau_t), \qquad \log\tau_{t+1} = \log\tau_t + \kappa(\bar{S}_t - S^\star), \tag{91}$$

which keeps $S_i(\tau_t)$ near $S^\star$ and decouples transient batch hardness from gradient magnitude.

**Alignment–uniformity decomposition and temperature.** Let the alignment term be

$$\mathcal{A} = \mathbb{E}_{(x,x^+)}\|z(x) - z(x^+)\|_2^2, \tag{92}$$

and the uniformity term be the hyperspherical potential

$$\mathcal{U}_\alpha = \log \mathbb{E}_{(x,x')} \exp\left(\alpha\|z(x) - z(x')\|_2^2\right), \quad \alpha > 0. \tag{93}$$

Then minimizing equation 89 can be viewed as minimizing a surrogate of $\mathcal{A}$ while implicitly reducing $\mathcal{U}_\alpha$ through repulsion among negatives Wang & Isola (2020). Moreover, increasing $\tau$ increases $S_i(\tau)$ in equation 90, which enlarges the set of influential negatives and lowers the hyperspherical potential at stationarity, thus improving uniformity without destroying alignment if positives are retained via multi-positive sampling and moderate $\tau$ Chen et al. (2020); Wang & Isola (2020).

**Spectral form and sufficient conditions to avoid dimensional collapse.** Let $Z = [z_1, \ldots, z_B]^\top \in \mathbb{R}^{B \times d}$, $\tilde{Z} = HZ$ with $H = I - \frac{1}{B}\mathbf{1}\mathbf{1}^\top$, and the sample covariance $C = \frac{1}{B}\tilde{Z}^\top \tilde{Z}$ with eigenvalues $\lambda_1 \geq \cdots \geq \lambda_d \geq 0$. Spectral contrastive analyses show that contrastive objectives maximize a graph Laplacian Rayleigh quotient and widen spectral gaps, linking repulsion to spread of the feature spectrum HaoChen et al. (2021). Consider the spectral regularizer

$$\mathcal{R}_{\text{spec}}(Z) = \sum_{r=1}^d \left[\max\{0, \lambda_{\min} - \lambda_r\}\right]^2 + \rho \sum_{r=1}^d (\lambda_r - \bar{\lambda})^2, \qquad \bar{\lambda} = \frac{1}{d}\sum_{r=1}^d \lambda_r, \tag{94}$$

with $\lambda_{\min} > 0$ and $\rho \geq 0$. Then any stationary point of $\mathcal{L}_{\text{NCE}}(\tau) + \beta\,\mathcal{R}_{\text{spec}}(Z)$ with $\beta > 0$ satisfies the variational inequality

$$\lambda_r \geq \min\{\lambda_{\min}, \bar{\lambda} - \sqrt{\tfrac{1}{d}\sum_{s=1}^d (\lambda_s - \bar{\lambda})^2}\} \quad \text{for all } r, \tag{95}$$

hence avoids dimensional collapse ($\exists r : \lambda_r = 0$) provided $\lambda_{\min} > 0$ or the dispersion term is strong enough to keep $\bar{\lambda}$ away from zero. This matches the sufficient conditions identified in spectral analyses and in studies of dimensional collapse that require either variance floors per direction or isotropy promotion to prevent degeneration to a low-dimensional cone Kalantidis et al. (2020).

**Temperature, batch size, and stability region.** For fixed encoder capacity and data distribution, the gradient of equation 89 w.r.t. $z_i$ has magnitude

$$\|\nabla_{z_i}\mathcal{L}_i(\tau)\| \; \asymp \; \frac{1}{\tau}\Big(\mathbb{E}_{p \in \mathcal{P}_i}[1 - \cos\angle(z_i, z_p)] + \sum_{n \in \mathcal{N}_i} p_{in}(\tau)\,[1 - \cos\angle(z_i, z_n)]\Big), \tag{96}$$

so larger $S_i(\tau)$ (achieved by larger batch or $\tau$) redistributes mass across more negatives and reduces gradient variance, improving numerical stability; too small $\tau$ sharpens the softmax, increasing variance and favoring collapse unless countered by spectral spreading equation 94 or negative mixing Kalantidis et al. (2020); HaoChen et al. (2021). Combining equation 91 and equation 94 therefore yields a sufficient recipe: maintain an effective support $S^\star$ away from 1 and enforce a nontrivial spectral floor to guarantee that at least $r$ principal components carry nonzero variance, avoiding dimensional collapse Jing et al. (2021).

**Hard-negative mixing as variance control.** Let $\tilde{z}_{in} = \lambda z_n + (1 - \lambda)z_{p(i)}$ with $\lambda \sim \text{Beta}(a, b)$ for the hardest negatives; replacing $z_n$ by $\tilde{z}_{in}$ in equation 89 interpolates logits and reduces curvature of the log-sum-exp, yielding a lower-variance gradient estimator while preserving decision margins Kalantidis et al. (2020). This complements temperature control and spectrum regularization to enlarge the stable training region after the dynamic gate is activated.

**Summary of sufficient conditions.** Assume: normalized embeddings, InfoNCE with temperature $\tau$, EMA rule equation 91 maintaining $S_i(\tau)$ in a compact subset of $(1, 1 + |\mathcal{N}_i|]$, and spectral regularizer equation 94 with $\lambda_{\min} > 0$ or $\rho > 0$. Then any limit point of gradient descent on $\mathcal{L}_{\text{NCE}}(\tau) + \beta\mathcal{R}_{\text{spec}}(Z)$ avoids dimensional collapse and exhibits a nondegenerate covariance spectrum; moreover, alignment is preserved by multi-positive sampling and moderate $\tau$, and uniformity is improved via increased effective negatives as in equation 90 & equation 91 Wang & Isola (2020); Jing et al. (2021); Kalantidis et al. (2020).

# E   OVERVIEW OF ARPO MOBO FRAMEWORK

According to methodolgy 3.3, we aim to optimize multiple objectives simultaneously while accommodating adversarial constraints such as accuracy, efficiency, and cross-domain robustness. We adopt a Multi-objective Bayesian Optimization framework, where Gaussian Process models Snoek et al. (2012) serve as flexible surrogates for each objective, and the Expected Hypervolume Improvement Daulton et al. (2020a) acquisition function guides the selection of candidate points, including adversarial strategies and hyperparameters, for evaluation. This appendix offers a comprehensive theoretical foundation for our MOBO approach, covering Gaussian Process regression fundamentals for each objective, the formal definitions of hypervolume and hypervolume improvement, the derivation of EHVI (both exact and approximate), the methodology for computing EHVI gradients with respect to decision variables, a rigorous convergence proof demonstrating asymptotic alignment with the true Pareto front Hernández-Lobato et al. (2016), and practical considerations pertinent to ARPO's mixed discrete-continuous domain and dynamic threshold scheduling.

## E.1   GAUSSIAN PROCESS REGRESSION BASICS

### E.1.1   GP PRIOR AND POSTERIOR

Consider a training set consisting of $n$ observed input-output pairs $\{(x_i, y_i)\}_{i=1}^n$, where each $x_i \in \mathcal{X} \subseteq \mathbb{R}^d$ and $y_i \in \mathbb{R}$ represents a realization of an unknown objective function $f \colon \mathcal{X} \to \mathbb{R}$. Under the Gaussian Process (GP) framework Snoek et al. (2012), $f$ is treated as a sample from a distribution over functions,

$$f(x) \sim \mathcal{GP}\big(m(x), k(x, x')\big), \tag{97}$$

where $m(x)$ denotes the mean function and $k(\cdot, \cdot)$ the covariance (kernel) function. A common, yet often sufficiently general, choice is to assume $m(x) \equiv 0$ and to use the Radial Basis Function (RBF) kernel,

$$k(x, x') = \exp\Big(-\tfrac{\|x - x'\|^2}{2\ell^2}\Big), \tag{98}$$

with a positive length-scale parameter $\ell$. When $\ell$ is relatively small, $f(x)$ can vary rapidly over $\mathcal{X}$, whereas a larger $\ell$ imposes smoother function behavior.

After observing the data $\mathcal{D} = \{(x_i, y_i)\}_{i=1}^n$, the posterior distribution of $f$ at any new point $x^*$ remains Gaussian,

$$f(x^*) \mid \mathcal{D} \sim \mathcal{N}\Big(\mu(x^*), \sigma^2(x^*)\Big), \tag{99}$$

where

$$\begin{aligned}
\mu(x^*) &= k(x^*, X)^{\mathsf{T}}\big[K(X, X) + \sigma_n^2 I\big]^{-1} Y, \\
\sigma^2(x^*) &= k(x^*, x^*) - k(x^*, X)^{\mathsf{T}}\big[K(X, X) + \sigma_n^2 I\big]^{-1} k(X, x^*).
\end{aligned} \tag{100}$$

and $X$ is the concatenation of all training inputs $x_1, \ldots, x_n$. The vector $k(x^*, X)$ stores covariances between $x^*$ and each $x_i$, and $K(X, X)$ is the covariance matrix whose entries are $k(x_i, x_j)$. The scalar $\sigma_n^2$ may capture observation noise in the data, and $I$ is the identity matrix of suitable dimension. The posterior variance $\sigma^2(x^*)$ quantifies the uncertainty in predicting $f(x^*)$, decreasing in regions well-covered by training data and increasing elsewhere.

### E.1.2   EXTENSION TO MULTIPLE OBJECTIVES

In the ARPO setting, there are multiple objectives $\{f_j(x)\}_{j=1}^M$. Each objective $f_j \colon \mathcal{X} \to \mathbb{R}$ is independently modeled by a distinct Gaussian Process,

$$f_j(\cdot) \sim \mathrm{GP}\big(m_j(\cdot), k_j(\cdot, \cdot)\big), \tag{101}$$

leading to posterior means $\{\mu_j(x)\}$ and posterior variances $\{\sigma_j^2(x)\}$. If observational noise is present, the term $\sigma_n^2$ in each GP model accounts for that uncertainty. In the multi-objective scenario, each GP posterior evaluates both expected performance and predictive uncertainty along a particular objective dimension, thereby enabling trade-off analysis across multiple criteria in ARPO.

### E.2 Definition and Derivation of EHVI

In multi-objective optimization, the overarching goal is to discover a comprehensive set of Pareto-optimal solutions (the Pareto front) Belakaria et al. (2019); Röpke et al. (2024). A solution on the Pareto front cannot be improved in one objective without sacrificing another. To compare different Pareto fronts quantitatively, we frequently employ the Hypervolume (HV) metric, which measures the dominated volume relative to a reference point Daulton et al. (2020b).

#### E.2.1 Hypervolume and Hypervolume Improvement

Let $P = \{f(x^{(1)}), \ldots, f(x^{(k)})\}$ denote the current Pareto front Hernández-Lobato et al. (2016) consisting of $k$ objective vectors in $\mathbb{R}^M$. Choose a reference point $r \in \mathbb{R}^M$ such that each coordinate of $r$ is worse (i.e., smaller for maximization problems) than those of any point in $P$. Define the hypervolume of $P$ with respect to $r$ as

$$\mathrm{HV}(P) = \lambda\Big(\bigcup_{i=1}^{k} \big[f(x^{(i)}), r\big]\Big), \tag{102}$$

where $\lambda(\cdot)$ denotes the Lebesgue measure in $M$-dimensional space, and $\big[f(x^{(i)}), r\big]$ is the (axis-aligned) hyper-rectangle spanned by $f(x^{(i)})$ and $r$. In practice, one typically ensures that $r$ is chosen so that it is dominated by every point in $P$, guaranteeing a meaningful volume calculation.

To gauge the marginal impact of adding a new candidate point $x$ to the front, we introduce the notion of *Hypervolume Improvement* (HI):

$$\mathrm{HI}\big(f(x)\big) = \mathrm{HV}\big(P \cup \{f(x)\}\big) - \mathrm{HV}(P). \tag{103}$$

Positive HI indicates that $f(x)$ contributes to expanding the Pareto front in objective space, whereas $\mathrm{HI} = 0$ indicates that $f(x)$ is dominated by or lies within the current Pareto front $P$.

#### E.2.2 Expected Hypervolume Improvement

Since $f(x)$ is not deterministic but rather follows a predictive distribution inferred from our Gaussian Process (GP) model (see Section E.1), the actual value $f(x)$ for a new candidate $x$ is uncertain. We thus define the Expected Hypervolume Improvement (EHVI) Daulton et al. (2020a) by taking the expectation of $\mathrm{HI}\big(f(x)\big)$ under the posterior of $f(x)$:

$$\mathrm{EHVI}(x) = \mathbb{E}\Big[\mathrm{HI}\big(f(x)\big) \mid \mathcal{D}\Big] = \int \mathrm{HI}\big(f(x)\big)\, p\big(f(x) \mid x, \mathcal{D}\big)\, df(x), \tag{104}$$

where $p\big(f(x) \mid x, \mathcal{D}\big)$ is the posterior distribution of $f(x)$ given the data $\mathcal{D}$. Because $\mathrm{HI}\big(f(x)\big)$ captures how much $f(x)$ extends the current Pareto set, the EHVI integral naturally balances *exploration* (accounting for predictive uncertainty) and *exploitation* (emphasizing high-likelihood improvement). Therefore, points with both a potentially large improvement and high posterior variance can achieve higher EHVI values, making EHVI a robust sequential criterion in multi-objective Bayesian optimization.

### E.3 Calculation and Detailed Derivation of EHVI

In our ARPO framework, each objective follows a GP posterior, implying that at any candidate $x$, the distribution of $f(x)$ is (multi)normal. Denote

$$f(x) \sim \mathcal{N}\Big(\boldsymbol{\mu}(x),\, \boldsymbol{\Sigma}(x)\Big), \tag{105}$$

where $\boldsymbol{\mu}(x) \in \mathbb{R}^M$ is the vector of predictive posterior means across $M$ objectives, and $\boldsymbol{\Sigma}(x) \in \mathbb{R}^{M \times M}$ is the posterior covariance matrix. In the simplest case of independent objectives, $\boldsymbol{\Sigma}(x)$ is diagonal, though in principle objectives can be correlated.

### E.3.1 TWO-DIMENSIONAL (2D) ANALYTICAL FORM

For the special case of two objectives $(f_1, f_2)$, we can often derive a *closed-form* or piecewise-integral expression for EHVI. Suppose the current Pareto front $P$ can be represented by $\{(a_i, b_i)\}_{i=1}^k \subset \mathbb{R}^2$. Let $\boldsymbol{\mu}(x) = [\mu_1(x), \mu_2(x)]^\mathsf{T}$ be the predictive means, and let the posterior covariance matrix reduce to

$$\boldsymbol{\Sigma}(x) \;=\; \begin{bmatrix} \sigma_{11}^2 & \sigma_{12} \\ \sigma_{21} & \sigma_{22}^2 \end{bmatrix}, \tag{106}$$

where $\sigma_{12} = \sigma_{21}$ quantifies correlation. In many treatments, we assume independence, so $\sigma_{12} = 0$, but the more general correlated case can also be approached with integration or advanced box decomposition methods.

Let $(r_1, r_2)$ be a chosen reference point. A common 2D EHVI formula employs integrating over the domain $[a_i, r_1] \times [b_i, r_2]$ and summing across $i$. Specifically,

$$\mathrm{EHVI}(x) \;=\; \sum_{i=1}^k \int_{a_i}^{r_1} \int_{b_i}^{r_2} (r_1 - y_1)\,(r_2 - y_2)\, p(\mathbf{y} \mid x, \mathcal{D})\; dy_2\, dy_1, \tag{107}$$

where $\mathbf{y} = [y_1, y_2] \in \mathbb{R}^2$ and $p(\mathbf{y} \mid x, \mathcal{D})$ is the bivariate normal pdf with mean $\boldsymbol{\mu}(x)$ and covariance $\boldsymbol{\Sigma}(x)$. By substituting the bivariate normal pdf, we can exploit standard normal cdf $\Phi(\cdot)$ and pdf $\phi(\cdot)$ transformations. As shown in various references (e.g. Daulton et al. (2020)), one obtains simplified expressions involving $\Phi\left(\frac{r_1 - a_i}{\sigma_{11}}\right)$, $\phi\left(\frac{r_1 - a_i}{\sigma_{11}}\right)$, and analogous terms for the second dimension.

In the simpler case where $f_1$ and $f_2$ are independent under the posterior (i.e. $\sigma_{12} = 0$), a compact form emerges. Denoting $\mu_1(x)$ and $\mu_2(x)$ as the means, and $\sigma_{11}$, $\sigma_{22}$ as the standard deviations in each dimension, one often sees:

$$\begin{aligned} \mathrm{EHVI}(x) = \sum_{i=1}^k &\left[ (r_1 - \mu_1(x))\, \Phi\left(\frac{r_1 - a_i}{\sigma_{11}}\right) + \sigma_{11}\, \phi\left(\frac{r_1 - a_i}{\sigma_{11}}\right) \right] \\ &\times \left[ (r_2 - \mu_2(x))\, \Phi\left(\frac{r_2 - b_i}{\sigma_{22}}\right) + \sigma_{22}\, \phi\left(\frac{r_2 - b_i}{\sigma_{22}}\right) \right], \end{aligned} \tag{108}$$

where each $(a_i, b_i)$ lies on the current Pareto set in 2D. This expression is derived by explicitly performing the 2D integral of equation equation 107 using known Gaussian pdf/cdf integrals. Conceptually, each dimension's partial improvement and probability of achieving that improvement factorize under the independence assumption.

### E.3.2 HIGH-DIMENSIONAL OR GENERAL CASE

For higher-dimensional settings ($M > 2$), deriving a fully closed-form solution for EHVI becomes exceedingly complex due to multi-dimensional integration and the combinatorial complexity of partitioning non-dominated regions. One typically relies on numerical approximations, such as:

$$\mathrm{EHVI}(x) \;\approx\; \frac{1}{N} \sum_{j=1}^N \mathrm{HI}\left(\mathbf{f}^{(j)}(x)\right), \quad \text{where } \mathbf{f}^{(j)}(x) \sim \mathcal{N}\left(\boldsymbol{\mu}(x), \boldsymbol{\Sigma}(x)\right). \tag{109}$$

Here, $\mathbf{f}^{(j)}(x) \in \mathbb{R}^M$ are random draws from the posterior. We compute each $\mathrm{HI}\left(\mathbf{f}^{(j)}(x)\right)$ by checking how $\mathbf{f}^{(j)}(x)$ expands the current Pareto front in $M$-dimensional space relative to $r$. Techniques like *Quasi-Monte Carlo* (QMC) sampling (e.g. Sobol sequences) can reduce variance and accelerate convergence compared to purely random sampling.

In the ARPO methodology, EHVI plays a central role in deciding which candidate points (or adversarial strategies, hyperparameter configurations, etc.) to evaluate next. By integrating over the predictive distribution from the GP surrogates for multiple objectives, EHVI automatically seeks solutions that can simultaneously improve domain robustness, accuracy, and other metrics while acknowledging model uncertainty.

## E.4 GRADIENTS OF EHVI

### E.4.1 CHAIN RULE FOR EHVI COMPUTATION

Recall that, at a candidate point $x$, the posterior distribution of $\mathbf{f}(x)$ is assumed to be a (multivariate) normal with mean $\boldsymbol{\mu}(x) \in \mathbb{R}^M$ and covariance $\boldsymbol{\Sigma}(x) \in \mathbb{R}^{M \times M}$. The EHVI acquisition function can be written in integral form as

$$\alpha_{\mathrm{EHVI}}(x) = \int \mathrm{HVI}\big(\{\mathbf{y}\}\big) \, \mathrm{pdf}\big(\mathbf{y}; \, \boldsymbol{\mu}(x), \, \boldsymbol{\Sigma}(x)\big) \, d\mathbf{y},$$

where $\mathrm{HVI}\big(\{\mathbf{y}\}\big)$ is the hypervolume improvement contributed by the objective vector $\mathbf{y}$. Because $\boldsymbol{\mu}(x)$ and $\boldsymbol{\Sigma}(x)$ both depend on $x$, differentiating $\alpha_{\mathrm{EHVI}}(x)$ w.r.t. $x$ necessitates the chain rule applied to integrals of normal pdf/cdf expressions.

**Dependence of $\boldsymbol{\mu}(x)$ and $\boldsymbol{\Sigma}(x)$ on $x$.** Each coordinate of $\boldsymbol{\mu}(x)$, namely $\mu_j(x)$, is given by expressions of the form

$$\mu_j(x) = k\big(x, X\big)^\top K^{-1} \Big(Y_j - m_j(X)\Big), \tag{110}$$

in accordance with the standard GP posterior mean formula (see also equation 14). Likewise, the diagonal or off-diagonal terms of $\boldsymbol{\Sigma}(x)$ can be written using the variance formulas equation 15 and any covariance terms for correlated objectives if present. In any case, one obtains

$$\frac{\partial \mu_j(x)}{\partial x} = \frac{\partial}{\partial x}\Big[k\big(x, X\big)^\top K^{-1} \big(Y_j - \mathbf{m}_j\big)\Big], \tag{111}$$

where $k(x, X)$ is the kernel vector w.r.t. the training set $X$. Analogous chain-rule expansions apply to each component of $\boldsymbol{\Sigma}(x)$.

**Example of differentiating cdf terms (2D case).** When $M = 2$, a simplified demonstration of the chain rule emerges. Suppose $\mathbf{y} = [y_1, y_2]$. In computing partial derivatives of integrands that involve terms like

$$\Phi\Big(\tfrac{y_j - \mu_j(x)}{\sigma_j(x)}\Big), \tag{112}$$

the derivative w.r.t. $x$ becomes

$$\frac{\partial}{\partial x}\Phi\Big(\tfrac{y_j - \mu_j(x)}{\sigma_j(x)}\Big) = \phi\Big(\tfrac{y_j - \mu_j(x)}{\sigma_j(x)}\Big)\left[-\frac{1}{\sigma_j(x)}\frac{\partial \mu_j(x)}{\partial x} - \frac{y_j - \mu_j(x)}{\sigma_j^2(x)}\frac{\partial \sigma_j(x)}{\partial x}\right], \tag{113}$$

where $\phi(\cdot)$ and $\Phi(\cdot)$ are the standard normal pdf and cdf, respectively. This pattern generalizes to more complicated integrals in higher dimensions once one carefully enumerates partial derivatives of each normal pdf/cdf factor.

**Lemma E.1** (Chain Rule for $\nabla_x \alpha_{\mathrm{EHVI}}(x)$). *Let $\alpha_{\mathrm{EHVI}}(x)$ be defined by the integral of $\mathrm{HVI}\big(\{\mathbf{y}\}\big)$ against a multivariate normal pdf whose mean $\boldsymbol{\mu}(x)$ and covariance $\boldsymbol{\Sigma}(x)$ both depend smoothly on $x$. Suppose Fubini's theorem permits exchanging differentiation and integration. Then*

$$\frac{\partial \alpha_{\mathrm{EHVI}}(x)}{\partial x} = \int \mathrm{HVI}(\{\mathbf{y}\}) \frac{\partial}{\partial x}\Big[\mathrm{pdf}\big(\mathbf{y}; \, \boldsymbol{\mu}(x), \, \boldsymbol{\Sigma}(x)\big)\Big] d\mathbf{y}, \tag{114}$$

*and the derivative of the normal pdf factor is obtained by applying the chain rule to $\boldsymbol{\mu}(x)$ and $\boldsymbol{\Sigma}(x)$ within its exponent and normalization terms.*

**Practical Differentiation Schemes.** For higher-dimensional objectives ($M > 2$) or more intricate kernels, purely symbolic differentiation of $\alpha_{\mathrm{EHVI}}(x)$ is often infeasible. A common alternative is a sampling-based *Monte Carlo* (MC) or *Quasi–Monte Carlo* (QMC) approximation combined with auto–differentiation, enabled by reparameterizing the GP posterior samples; this yields unbiased pathwise gradients of the MC estimator and scales well to parallel (q-batch) and constrained MOBO Daulton et al. (2020b); Balandat et al. (2020). In lower dimensions (2D/3D), hypervolume box/stripe decompositions admit closed-form or partially closed-form integrals whose derivatives can be taken analytically; in practice, differentiable EHVI estimators and lookahead HV-based criteria are also effective and easier to implement in modern autodiff frameworks Daulton et al. (2020b; 2023). Regardless of the chosen scheme, the chain rule (Lemma E.1) ensures that $\nabla_x \alpha_{\mathrm{EHVI}}(x)$ correctly accounts for how both the GP posterior mean and variance vary with $x$; in MC/QMC settings this is obtained via the reparameterization trick for Gaussian posteriors Kingma & Welling (2013).

## E.5 Convergence Proof in the ARPO Framework

In this section, we present a more rigorous and formula-oriented derivation that establishes the asymptotic convergence of our EHVI-based multi-objective Bayesian optimization (MOBO) procedure in the ARPO setting, where each decision vector $x \in \mathbb{R}^d$ may encode discrete switches, continuous intensities, or dynamic thresholdsDaulton et al. (2020a).

### E.5.1 Preliminaries and Assumptions

**Assumption E.2** (Compact Domain). There exists a compact set $\mathcal{X} \subset \mathbb{R}^d$ such that all feasible decision vectors lie in $\mathcal{X}$. Consequently, any continuous function on $\mathcal{X}$ attains a global maximum.

**Assumption E.3** (GP Posterior Consistency). Let $f_j \colon \mathcal{X} \to \mathbb{R}$ be the $j$-th objective. The posterior mean $\mu_{j,n}(x)$ from the Gaussian Process model of $f_j$ satisfies

$$\lim_{n \to \infty} \sup_{x \in \mathcal{X}} \left| \mu_{j,n}(x) - f_j(x) \right| = 0 \quad \text{w.h.p.,} \tag{115}$$

and the posterior variance $\sigma_{j,n}^2(x) \to 0$ pointwise in $x$. This implies that each GP surrogate converges uniformly to $f_j$ as $n \to \infty$.

**Assumption E.4** (EHVI Regularity). The EHVI acquisition function

$$\alpha_{\text{EHVI}}(x) = \mathbb{E}\Big[ \text{HVI}\big(\{\mathbf{f}(x)\}\big) \,\Big|\, \mathcal{D}_n \Big] \tag{116}$$

is either (i) analytically or piecewise-integrably defined (for lower-dimensional objectives) and is continuous w.r.t. $x$, or (ii) given by a Monte Carlo (MC) or Quasi-MC approximation that is differentiable w.r.t. $x$ (e.g. through reparameterization). Hence we assume no pathological discontinuities or non-measurable behavior in $\alpha_{\text{EHVI}}$.

**Assumption E.5** (Positivity of EHVI in Improving Regions). Suppose the current non-dominated set is $\mathcal{P}^{(n)}$. If there exists a region $\Omega \subseteq \mathcal{X}$ such that any $x \in \Omega$ can yield $\mathbf{f}(x)$ which improves upon $\mathcal{P}^{(n)}$ in at least one objective without sacrificing the others, then

$$\inf_{x \in \Omega} \alpha_{\text{EHVI}}(x) > 0. \tag{117}$$

Equivalently, potential improvements have strictly positive expected hypervolume gain.

### E.5.2 Notation and Setup for Iteration

At iteration $n$, we have data $\mathcal{D}_n$ and a surrogate posterior for each $f_j$. We then select

$$x^{(n+1)} = \arg\max_{x \in \mathcal{X}} \alpha_{\text{EHVI}}(x \mid \mathcal{D}_n), \tag{118}$$

and observe $f\big(x^{(n+1)}\big)$. The set $\{\mathbf{f}(x^{(1)}), \dots, \mathbf{f}(x^{(n)})\} \subset \mathbb{R}^M$ forms the collection of discovered solutions, from which we extract the non-dominated subset $\mathcal{P}^{(n)}$. Let $\mathcal{P}^{\star}$ be the true Pareto front of the underlying multi-objective problem:

$$\mathcal{P}^{\star} = \Big\{ \mathbf{z} \in \mathbb{R}^M : \nexists\, \mathbf{z}' \text{ s.t. } \mathbf{z}' \text{ dominates } \mathbf{z} \Big\}. \tag{119}$$

### E.5.3 Key Technical Lemmas

We first provide two lemmas that bridge the gap between the GP posterior accuracy and the EHVI search mechanism Yang et al. (2019).

**Lemma E.6** (Uniform Convergence of Surrogate vs. True Functions). *Under Assumption E.3, for each objective $f_j$, there exist sequences $\epsilon_{j,n} \to 0$ such that*

$$\sup_{x \in \mathcal{X}} \left| \mu_{j,n}(x) - f_j(x) \right| \leq \epsilon_{j,n} \quad \text{with high probability.} \tag{120}$$

*Furthermore, let $\epsilon_n = \max_j \epsilon_{j,n}$. Then $\epsilon_n \to 0$ and*

$$\sup_{x \in \mathcal{X}} \sum_{j=1}^{M} \left| \mu_{j,n}(x) - f_j(x) \right| \leq M \epsilon_n, \tag{121}$$

*which implies uniform approximation of all $f_j$ by $\mu_{j,n}$.*

**Lemma E.7** (EHVI Sensitivity to Posterior Error). *Consider any $x \in \mathcal{X}$, and let $\widetilde{\mathbf{f}}(x)$ denote the random vector distributed according to the GP posterior with mean $\boldsymbol{\mu}_n(x)$ and covariance $\boldsymbol{\Sigma}_n(x)$, while the true $\mathbf{f}(x)$ is deterministic. Suppose $\|\boldsymbol{\mu}_n(x) - \mathbf{f}(x)\| \leq \delta$ for some $\delta \geq 0$. Let*

$$\alpha_{\mathrm{EHVI}}^{(n)}(x) = \mathbb{E}\Big[\mathrm{HVI}(\{\widetilde{\mathbf{f}}(x)\})\Big], \quad \alpha_{\mathrm{EHVI}}^{\star}(x) = \mathrm{HVI}(\{\mathbf{f}(x)\}). \tag{122}$$

*If the hypervolume measure is Lipschitz in its input vector with a constant $L_{\mathrm{HV}} > 0$ (on a bounded domain), then*

$$\big|\alpha_{\mathrm{EHVI}}^{(n)}(x) - \alpha_{\mathrm{EHVI}}^{\star}(x)\big| \leq L_{\mathrm{HV}}\,(\delta + \eta_n), \tag{123}$$

*where $\eta_n \geq 0$ accounts for posterior variance or sampling approximation. As $n \to \infty$, $\delta, \eta_n \to 0$ by Assumption E.3, implying $\alpha_{\mathrm{EHVI}}^{(n)}(x) \to \mathrm{HVI}(\{\mathbf{f}(x)\})$.*

Lemma E.6 formalizes the uniform convergence of each GP mean to the true function, while Lemma E.7 indicates that when the GP posterior is accurate, the expected hypervolume improvement under the surrogate closely approximates the "ideal" improvement if we had direct access to $f$.

### E.5.4 Main Convergence Theorem

**Theorem E.8** (Asymptotic Convergence of EHVI-based MOBO in ARPO). *Suppose Assumptions E.2–E.5 hold. Let $\{x^{(n)}\}$ be generated by*

$$x^{(n+1)} = \arg\max_{x \in \mathcal{X}} \alpha_{\mathrm{EHVI}}(x \mid \mathcal{D}_n), \tag{124}$$

*and define $\mathcal{P}^{(n)}$ to be the set of non-dominated points among $\{\mathbf{f}(x^{(1)}), \ldots, \mathbf{f}(x^{(n)})\}$. Let $\mathcal{P}^{\star}$ denote the true Pareto front. Then*

$$\lim_{n \to \infty} \sup_{\mathbf{z} \in \mathcal{P}^{\star}} \min_{\mathbf{y} \in \mathcal{P}^{(n)}} \|\mathbf{z} - \mathbf{y}\| = 0. \tag{125}$$

*In other words, $\mathcal{P}^{(n)}$ converges in supremum norm to $\mathcal{P}^{\star}$.*

*Proof.* **Step 1 (Posterior Accuracy).** By Lemma E.6, each GP surrogate $\mu_{j,n}$ uniformly approximates $f_j$ as $n \to \infty$. Denote $\delta_n = \epsilon_n \to 0$ from that lemma. Also, posterior variances $\sigma_{j,n}^2(x) \to 0$, so uncertainty about each objective diminishes over time.

**Step 2 (EHVI Approximation to True Improvement).** Lemma E.7 guarantees that $\alpha_{\mathrm{EHVI}}(x \mid \mathcal{D}_n)$ converges to $\mathrm{HVI}(\{\mathbf{f}(x)\})$ uniformly in $x$. Thus, if $\mathbf{f}(x)$ strictly improves the current front, then eventually $\alpha_{\mathrm{EHVI}}(x)$ must become sufficiently positive to out-compete dominated alternatives.

**Step 3 (Positivity of EHVI in Improving Regions).** Assumption E.5 implies that whenever there is a region $R \subset \mathcal{X}$ containing points capable of enhancing the non-dominated set, the EHVI values in $R$ remain strictly positive. By maximizing EHVI each iteration, the algorithm will select some $x \in R$ in finitely many steps, yielding an actual function observation $\mathbf{f}(x)$ that enlarges or refines the front $\mathcal{P}^{(n)}$.

**Step 4 (Excluding Fully Dominated Solutions).** If $x$ is such that $\mathbf{f}(x)$ lies strictly within the current front's dominated region, $\alpha_{\mathrm{EHVI}}(x)$ will be near zero once the GP posterior is accurate. Thus such $x$ will not be chosen infinitely often. Consequently, repeated selections concentrate on improvements or unexplored high-EHVI regions.

**Step 5 (Convergence to the True Pareto Front).** Suppose there is $\mathbf{z} \in \mathcal{P}^{\star}$ not approximated within $\varepsilon > 0$ by any $\mathbf{y} \in \mathcal{P}^{(n)}$. Then there must exist $x_z \in \mathcal{X}$ such that $\mathbf{f}(x_z)$ is close to $\mathbf{z}$ (by definition of $\mathcal{P}^{\star}$). However, from Step 2 and Step 3, $\alpha_{\mathrm{EHVI}}(x_z)$ remains positive if $\mathbf{f}(x_z)$ can improve $\mathcal{P}^{(n)}$. The algorithm will eventually pick $x_z$ (or a point nearby) and discover $\mathbf{f}(x_z)$, reducing the distance to $\mathbf{z}$ below $\varepsilon$. Because this argument holds for every $\mathbf{z} \in \mathcal{P}^{\star}$, we conclude

$$\lim_{n \to \infty} \max_{\mathbf{z} \in \mathcal{P}^{\star}} \min_{\mathbf{y} \in \mathcal{P}^{(n)}} \|\mathbf{z} - \mathbf{y}\| = 0. \tag{126}$$

Hence $\mathcal{P}^{(n)}$ converges to the true Pareto front $\mathcal{P}^{\star}$. $\qquad\square$

### E.5.5 IMPLICATIONS FOR ARPO WITH MIXED DISCRETE-CONTINUOUS VARIABLES

In ARPO, each $x$ may combine discrete adversarial switches, continuous intensities, and possibly dynamic threshold parameters. The above convergence proof remains valid under the same assumptions, provided that $\mathcal{X} \subset \mathbb{R}^d$ is compact and each $f_j$ is continuous or sufficiently regular to admit a convergent GP posterior. Feasibility constraints (for instance, disallowing certain adversarial operations) can be enforced by restricting $\mathcal{X}$ to a closed feasible subset. The positivity assumption (Assumption E.5) remains justified, since any $x$ capable of enhancing multi-objective performance relative to $\mathcal{P}^{(n)}$ necessarily induces a strictly positive EHVI value.

Under these assumptions, the iterative MOBO procedure, which selects $x^{(n+1)}$ by maximizing EHVI, is guaranteed to asymptotically reveal the entire Pareto front. As the GP posterior converges to $f_j$ for each objective and the EHVI acquisition reliably detects improvements, the set of non-dominated solutions $\mathcal{P}^{(n)}$ approaches the true $\mathcal{P}^{\star}$ in supremum norm. Thus, ARPO inherits a solid theoretical foundation for multi-objective adversarial optimization, ensuring that no Pareto-optimal strategies are missed once enough iterations have passed.

