# OpenReview forum: "Disentangle, Gate, and Optimize: Cross Domain Transfer power by Multi Objective Bayesian Optimization"
_ICLR.cc/2026/Conference — ICLR 2026 Conference Desk Rejected Submission_

### Official Review · Reviewer_y1fc · 2025-10-29

**Soundness:** 3
**Presentation:** 3
**Contribution:** 3
**Rating:** 8
**Confidence:** 3

**Summary:**

The paper proposes a method for parameter-efficient fine-tuning of pre-trained language models that enhances cross-domain robustness and training stability. The disentangled prefix tuning framework separates domain-invariant (DI) and domain-specific (DS) knowledge. The Information Bottleneck objective, dynamic adversarial training schedule, and Multi-Objective Bayesian Optimization altogether improve model robustness.

**Strengths:**

- Propose a principled approach to decompose prefix parameters into DI and DS components, which enables better generalization across domains by isolating shared semantics from domain-specific noise.
- Introducing a condition-based trigger (gradient variance stability) for adversarial updates improves training dynamics. This avoids early-stage instability common in adversarial training, contributing to faster and steadier convergence.
- The paper shows strong empirical gains and validates the effectiveness of the proposed methods.

**Weaknesses:**

- MOBO and dynamic scheduling likely add computational overhead. No discussion of inference latency, parameter count, or training time efficiency limits claims about practicality. Could this become a bottleneck in large-scale applications?

**Questions:**

- While the DI:DS 5:5 performs best, is this ratio transferable across tasks/domains, or must it be re-tuned each time? Does auto-tuning via MOBO include this hyperparameter?

---

> ### Author Response · Authors · 2025-11-21
>
> **W1: Practicality of MOBO scheduling**
>
> We thank the reviewer for raising this concern about the cost of MOBO and dynamic scheduling. In ARPO, both MOBO and the dynamic gate are used only during training (Section 3.1; Algorithm 1). At inference, we only prepend the learned soft prefix to the input, as in standard prefix tuning, and we never call MOBO or the gate again. Thus inference latency and memory are almost the same as normal soft-prompt tuning: trainable parameters stay in the prefix space and remain below 1% of backbone weights (Appendix B.2; Appendix B.4.4). Appendix B.4.4 and Fig. 2(b) further show that, on T5-base, full fine-tuning takes about 100 hours, DePT 43.3 hours, and ARPO 37.5 hours, with normalized cost 0.64 for ARPO vs. 0.73 for DePT, while achieving better accuracy and robustness.
>
> MOBO is also designed to be light and sample-efficient rather than a heavy outer loop. As described in Sec. 3.3 and Appendix B.3, we search over a low-dimensional decision vector $x = (x_{\text{struct}}, \alpha, \theta)$ that encodes a small set of perturbation strategies, the adversarial weight, and trigger thresholds. Using noisy-qEHVI, we typically need about 30 evaluations per task, whereas grid or random search would require hundreds (Appendix B.3; Appendix B.4.2). Dynamic gating skips around 40% of adversarial steps in the first 30% of training iterations, which further reduces cost (Appendix B.4.4). Once we identify a good strategy region for a model family and task cluster, we reuse it on related tasks, so the MOBO overhead is one-off and amortized, rather than a recurring bottleneck in large-scale deployment.

---

> ### Author Response · Authors · 2025-11-21
>
> **Q1: Transferability of 5:5 prefix**
>
> We thank the reviewer for asking about the transferability of the DI:DS = 5:5 prefix split. In Table 3, we run a systematic ablation over multiple ratios (9:1, 8:2, …, 1:9) on both T5-base and LLaMA-7B, under Same Task, Different Domains (CombA) and Different Tasks, Different Domains (CombB). The 5:5 split consistently gives the best results, improving over the second-best 6:4 by about 1.9% on CombA and 2.4% on CombB, and over 1:9 by about 14% in both settings (Table 3).
>
> Based on this unified result, we fix $L = 60$ and DI:DS = 5:5 for all main experiments in Section 4, without re-tuning the ratio per task or dataset. The consistent gains across tasks and backbones in Tables 1, 2, 5, and 6 show that this balanced split transfers well across the diverse tasks and domains we study, rather than overfitting a single dataset.
>
> Regarding MOBO, our current search space is explicitly defined in Section 3.3.2 and summarized in Table 8 as a decision vector $x = (\alpha, \theta, \lambda_1,\dots,\lambda_5,\text{Strategy})$. MOBO automatically tunes the information bottleneck weights, the dynamic gate threshold, and the adversarial strategy to balance accuracy, robustness, and cost, but it does not include the total prefix length $L$ or the DI:DS ratio. We treat these as structural design choices that we first determine once by ablation (Table 3) and then keep fixed to avoid unnecessary expansion of the search dimension and to improve optimization efficiency.
>
> In the revised version, we will state explicitly that DI:DS = 5:5 is used for all experiments and is not re-tuned by MOBO, and that the combination of a transferable disentangled prefix structure with automatic robustness tuning is one of the key contributions of our method.

---

### Official Review · Reviewer_Vd6T · 2025-11-02

**Soundness:** 2
**Presentation:** 2
**Contribution:** 2
**Rating:** 2
**Confidence:** 3

**Summary:**

The paper proposes a training framework for prefix-tuned LLMs (ARPO) that disentangles domain-invariant and domain-specific information. It uses a gated adversarial curriculum and multi-objective Bayesian optimization to balance performance and robustness. Experiments on various NLP tasks demonstrate the effectiveness of the proposed method against baselines.

**Strengths:**

The problem that the paper aims to tackle involves several important yet difficult pieces. For example, distinguishing between domain-specific and domain-invariant is still arguably an open problem. This paper proposes a training framework incorporating loss functions of domain-specific and domain-invariant mutual information terms. The empirical evaluation is done on numerous tasks.

**Weaknesses:**

This paper has some flaws that raises concerns about the significance of the proposed method and its results.
* First, this paper states that Prompt Tuning includes Adapters and LoRA (line 49). However, Adapters and LoRA both involve model parameter tuning and are not prompt tuning methods.
* This paper is not well written. For example, in the first paragraph of section 3.1, $x, D, Y$ are used without being defined.
* Although the paper includes some ablation studies such as prefix length and disentanglement constraints, the proposed method is significantly more complex. E.g.,  it has three complicated steps: the first step has 5 loss terms, the second has 3 loss terms, and the third involves multi-objective Bayesian optimization over discrete-continuous domains. I appreciate that the authors summarize the hyper parameters in table 8, and that limited sensitivity investigation is provided in table 7, but questions remain given the huge hyper parameter space: does the optimal range it hold for every tasks; is the optimal range independent from each other; how would one find the optimal range efficiently in practice?

**Questions:**

* Is it possible to show transfer for any two of the datasets? The dataset used in the experiments seems selective.

---

> ### Author Response · Authors · 2025-11-21
>
> **W1: Theoretical correction for PEFT & PT**
>
> Thank you for pointing out the unclear wording in our related work section. We agree that, in precise terms, Adapters and LoRA are better viewed as parameter efficient fine tuning (PEFT) methods rather than narrow prompt tuning. Our method ARPO is implemented as a standard soft prefix based prompt tuning approach. During training, the backbone remains frozen and we only learn prefix vectors at the input side. These vectors are then split into domain invariant $P_{\text{DI}}$ and domain specific $P_{\text{DS}}$ in Sec. 3.1 and optimized as soft prompts (Sec. 3, Appendix B.1–B.2). Adapters, LoRA, and Udapter appear only as PEFT baselines in Tables 1–6 and are not part of the ARPO framework itself.
>
> In the revised version, we will correct the wording in Sec. 2 “Domain Adaptation and Prompt Tuning”. The current sentence “Prompt Tuning (PT) strategies, such as Adapters, LoRA, and AdapterDrop” (Sec. 2) will be changed to describe these methods as “Parameter-Efficient Fine-Tuning (PEFT) methods, including Adapters, LoRA, AdapterDrop, and various prompt-tuning schemes”. We will also rename the subsection to “Prompt Tuning and PEFT Methods”, and at the beginning of Sec. 3 we will explicitly state that ARPO is a soft prefix based prompt tuning framework, enhanced by prefix disentanglement (Sec. 3.1), dynamic adversarial gating (Sec. 3.2), and MOBO based decision making (Sec. 3.3, Appendices C–E). These clarifications do not change the substance of our contributions, but they avoid possible confusion and make the position of ARPO within the prompt-tuning literature more precise.

---

> ### Author Response · Authors · 2025-11-21
>
> **W2: ARPO method notation and writing clarity**
>
> Thank you for pointing out that in the first paragraph of Sec. 3.1 we use $x$, $D$, and $Y$ without defining them in the main text. This is a writing oversight and can indeed make the method harder to follow at first reading. These symbols are formally introduced in Appendix B.1 (“Preliminaries”), where $x$ denotes the input sequence, $D$ the domain indicator, and $Y$ the task label, and we use them to formalize prefix tuning and cross-domain transfer.
>
> In the revision, we have moved a minimal version of this setup from Appendix B.1 into the main paper at the beginning of Sec. 3 that explicitly defines $x$, $D$, and $Y$, briefly recaps the cross-domain prefix-tuning setting, and makes the notation easier to follow for readers. We will also restate the roles of $x$, $D$, and $Y$ in parentheses when we first describe domain-invariant and domain-specific prefixes in Sec. 3.1.
>
> We will then carefully check Sec. 3 and Appendices C–E to make sure that every new symbol is defined where it first appears. This includes the information-bottleneck loss and its coefficients in Sec. 3.1, the multi-level constraints, the dynamic gate parameters in Sec. 3.2, and the MOBO decision vector $x = (\alpha, \theta, \lambda_1,\dots,\lambda_5,\text{Strategy})$ in Sec. 3.3 and Appendix B.3. These edits mean that readers no longer need to go to the appendix just to understand the basic notation. They also make the structure of our contributions clearer, including prefix disentanglement in Sec. 3.1, stability-aware dynamic adversarial gating in Sec. 3.2, and MOBO over strategy space in Sec. 3.3.

---

> ### Author Response · Authors · 2025-11-21
>
> **W3: Complexity and hyperparameter search**
>
> Thank you for carefully examining the three step structure and for raising concerns about complexity and hyperparameters. First, regarding the overall complexity of ARPO, the three modules are designed to address three concrete weaknesses of existing prompt tuning for domain transfer. Section 3.1 uses an information bottleneck based prefix disentanglement to separate domain invariant and domain specific signals. Section 3.2 introduces a dynamic adversarial gate that controls when adversarial signals enter training so that optimization remains stable. Section 3.3 then applies multi objective Bayesian optimization to balance accuracy, robustness, and cost.
>
> Most parameters in these modules, such as the prefix vectors and the gating network, are learned by backpropagation. The set of hyperparameters that users need to choose is much smaller and is summarized in Section 3.3 and Table 8, including a few loss weights $\lambda_1$ to $\lambda_5$, gate thresholds $\theta_0, \gamma, \beta$, the adversarial weight $\alpha$, and a small number of strategy switches. Table 7 further shows that performance remains stable over reasonably wide ranges for these weights instead of peaking at a single narrow point.
>
> Second, regarding the size and interaction of the hyperparameter space, we do not assume that these hyperparameters are independent. In our method they are treated as a joint decision vector $x = (\alpha, \theta, \lambda_1,\dots,\lambda_5, \text{Strategy})$ and are optimized together in a mixed discrete and continuous space by multi objective Bayesian optimization, with theoretical support for convergence to the Pareto front in Appendix E.
>
> This design explicitly captures interactions between loss weights, gate thresholds, and strategy choices rather than tuning them in isolation. The ablation studies in Section 4.3 and Appendix B.4 show that ARPO is robust to moderate changes of these hyperparameters and that each module, including disentanglement, adversarial training, dynamic gating, and the BO controller, contributes clear gains in clean and robust accuracy without requiring fine grained manual tuning of every coefficient.
>
> Third, regarding generalization of good hyperparameter ranges across tasks and efficient search in practice, we use one unified search space and procedure for all experiments. Section 4.1 and Tables 1, 2, 5, 6, and 9, together with Appendices B.3, B.4.2, and B.4.3, show that under this shared configuration ARPO consistently improves over strong baselines such as DePT and DAdEE on four backbones (Qwen3 4B, LLaMA2 7B, T5 base, T5 large) and ten cross domain and cross task transfers.
>
> In practice, MOBO operates on the low dimensional decision vector $x$, starts from a Sobol initialization, and typically needs only a few dozen evaluations per task, since each evaluation trains only the prefix and gate while the backbone remains frozen (Section 3.3, Appendix B.3). Once we identify a stable region for a model family and task cluster, we reuse this region on related tasks. Practitioners can therefore start from the default settings in Table 8 and, if needed, perform a small search over two or three of the most sensitive weights inside the stable ranges suggested by Table 7. As shown in Figure 2, ARPO reaches better accuracy and robustness with similar or lower normalized cost compared with DePT and other baselines, and Figure 3 illustrates that the dynamic gate leads to smoother training. At inference time, we only attach the learned prefix and apply a light gate, so the parameter size and latency remain similar to standard prefix tuning.
>
> We believe this level of additional structure is justified by the consistent gains in performance and robustness and remains practical for large scale use.

---

> ### Author Response · Authors · 2025-11-21
>
> **Q1: Transfer across domain datasets**
>
> We thank the reviewer for the question about transfer between datasets. In our design, ARPO does not rely on any special property of a specific dataset. The method only assumes a shared language model and a unified text-input–text-output format. Once we convert different datasets into this format, we apply the same three steps: disentangled prefixes, the dynamic gate, and MOBO-based decision making (Sec. 3.1–3.3, Fig. 1, Algorithm 1).
>
> This means that, in principle, the algorithm can be used for any pair of datasets that can be cast as text-to-text tasks, not only for the pairs shown in our experiments. In the revision, we will make this interface and its generality more explicit in the method and experiment sections.
>
> For the empirical study, we do not select a few “best-looking” pairs after the fact. We construct two representative groups of transfers from GLUE, SuperGLUE, MRQA, Yelp, SciTail, PAWS-Wiki, GSM8K, and HumanEval.
>
> Appendix B.3.1 covers “same or similar task, different domains” pairs such as NQ→SQA, Yelp→SciTail, News→HP, MNLI→QQP, and CoLA→PAWS, while Appendix B.3.2 covers “different task and different domain” pairs such as CoLA→QQP, GSM8K→HumanEval, BoolQ→NQ, HP→SciTail, and MNLI→SQA.
>
> Enumerating all O(n²) combinations is not feasible in terms of space and compute. However, Section 4.1 and Tables 1, 2, 5, and 6, together with Appendix B.4.3, show that ARPO consistently improves over the strongest baselines on all ten transfers and on four backbones (Qwen3-4B, LLaMA2-7B, T5-base, T5-large). These results suggest that our gains do not depend on any single favorable pair, and that the framework generalizes across diverse tasks and domains.

---

### Official Review · Reviewer_MWAq · 2025-11-03

**Soundness:** 2
**Presentation:** 1
**Contribution:** 2
**Rating:** 2
**Confidence:** 3

**Summary:**

The paper proposes ARPO, a three-stage framework for cross-domain transfer: 1) Disentangle learnable prefixes into domain-invariant and domain-specific parts via information-theoretic and geometric regularizers); 2) Gate adversarial/perturbation training with a dynamic controller; 3) Optimize strategy and hyperparameters via multi-objective Bayesian optimization (MOBO). Experiments on multiple backbones and transfer pairs report consistent improvements and better robustness under distribution shift.

**Strengths:**

1. Training stability: The dynamic gate reduces early-stage adversarial damage and improves convergence behavior in practice.
2. Promising empirical signal: Gains are shown across several model families and transfer settings, with robustness benefits under perturbations.

**Weaknesses:**

1. **Unclear methodological exposition.**
The presentation of the method is difficult to follow and lacks essential preliminaries that would help readers understand the design choices and assumptions. While the appendix supplies some details, the main text should summarize the core ideas (key variables, objectives, constraints, and training flow) without requiring readers to consult the appendix to grasp the basic mechanism.

2. **Insufficient rigor in writing and notation.**
There are many notational and mathematical issues that hinder precise understanding. For example:
    - Around line 150, the symbol $I$ is used without definition.
    - In Eq.2, $\| P_{DI}^TP_{DS} \|$ should be $\| P_{DI}P_{DS}^T \|$.
    - In Eq.7, the numerator involves $\text{var}(\nabla L_{\text{task}}(t-w))$. As written, this applies a variance to a vector, yet a scalar is required in the fraction.
    - In Algorithm 1, $d$ is overloaded to mean both domain and dimension, and $\theta$ is used both for model parameters and for a threshold.
    - The subscript setting for Eq.17 is incorrect.
    There are also some other issues. I suggest the authors should check the paper carefully to correct the above and other issues.

3. **Overly complex design and heavy hyperparameterization.**
For the adaptation task, the method introduces a large set of hyperparameters—including $\lambda_{1:5}, \gamma, \omega$, a suite of pre-designed operators—each of which typically has its own settings (e.g., token-PGD requires the number of attack steps and an adversarial budget) and so many other hyperparameters. This level of complexity raises serious concerns about practicality and deployability unless robust defaults, sensitivity analyses, and clear tuning guidance are provided.

4. **Substantial computational overhead.**
Relative to standard fine-tuning, the approach requires multiple full or partial trainings ($|X_0|+T_{BO}$ times training)for Bayesian optimization. The empirical gains reported do not yet convincingly justify this added compute. The paper should quantify wall-clock time, GPU hours, and energy usage, and compare them against the achieved improvements; otherwise, the cost-benefit trade-off remains unfavorable.

5. **Missing ablations.**
The paper lacks systematic ablation studies isolating the contribution of each component (e.g., disentangling regularizers, gating, individual operators, and the BO layer). Such ablations are necessary to establish which parts are essential and which provide marginal benefit.

6. **Limited transparency and reproducibility.**
Although the appendix includes a few hyperparameter ablations, many important settings are not documented. In addition, code is not provided, preventing independent verification of the results. This raises concerns about reproducibility.

**Questions:**

1. In fact, I am still confused about the transfer setting after reading Algorithm 1. For example, are all samples in $D_train$ from the same dataset? If so, why samples from the same dataset could be deviced into positive and negative sample pairs? And why is this an adversarial loss? It seems to help the model to cluster some samples.

2. In Algorithm 1, is the final output part used for inference solely comprised of the trained prefix $P^*$?

3. How are phrase-level and task-level operators performed?

4. How are Cost and Robust in $\textrm{f}(x)$ calculated?

5. In Algorithm 1, $\lambda_{1:5}$ are Input. How are they are optimzied with Adam according to Table 8?

---

> ### Author Response · Authors · 2025-11-21
>
> We sincerely appreciate the reviewer for thoroughly examining our submission and offering valuable, constructive feedback. Below, we address all identified weaknesses (W1–W6) and respond to all queries (Q1–Q5). We systematically address each concern and elucidate all design decisions, notation, implementation specifics, and training methodologies. No technical content has been modified; we have solely enhanced the explication and presentation in accordance with reviewer comments.
>
> **W1. Unclear methodological exposition**
>
> Thank you for pointing this out. In the original submission, we had a dedicated preliminary section that defined the core variables, assumptions, and the overall training flow. Due to strict page limits, we had to move this section to Appendix B.1, which unfortunately caused several components in Section 3 to appear without sufficient introduction. We appreciate the reviewer’s observation, as it allows us to clarify the method more cleanly.
>
> In the revision, Section 3 will begin with a concise “Preliminary’’ that brings all essential elements into one place. This overview will summarize the key components of ARPO, including the disentangled prefixes $(P_{\mathrm{DI}}, P_{\mathrm{DS}})$,
> the gating indicator $\delta(L_{\text{task}}(t))$ and its threshold $\theta(t)$, the operator set $\{O_j\}$, and the unified decision vector
> $x = (x_{\text{struct}}, \alpha, \theta).$
> It will also describe the full training flow: encoding, prefix disentanglement, dynamically gated adversarial training, and the MOBO outer-loop, together with the three optimization objectives (accuracy, robustness, normalized cost). To avoid forcing readers to consult the appendix, we will move a compact symbol table and the key modeling assumptions directly into the main text.
>
> This restructuring makes the method easier to follow and presents ARPO as a clear three-stage pipeline:
> 1. Information-theoretic and geometric constraints separate domain-invariant and domain-specific prefix components.
> 2. Stability-based dynamic gating activates adversarial perturbations only after the main task converges.
> 3. MOBO jointly optimizes the timing, strength, and composition of perturbations through the decision vector above.
>
> To further improve clarity, we will directly refer to the main architecture figure (Figure 1) in the paper. This figure already integrates the disentanglement module, the dynamic-gating mechanism, and the MOBO search space, and it serves as a clear visual summary of all variables, operators, and training stages. We believe it will help readers navigate Sec 3 more easily.

---

> ### Author Response · Authors · 2025-11-21
>
> **W2. Insufficient rigor in writing and notation**
>
> Thank you for the reviewer’s careful and rigorous comments on notation and mathematical clarity. We have made systematic and unified revisions to address all identified issues.
>
>  (1) When introducing the information bottleneck loss in Sec 3.1, we now explicitly state that $I(\cdot,;\cdot)$ denotes mutual information, and we maintain the same definition in Appendix C to avoid ambiguity.
>
>  (2) We have rewritten the orthogonality constraint in Eq. (2) as $L_{\text{orth}}=\lVert P_{DI} P_{DS}^\top \rVert_F^2$, ensuring that matrix dimensions align with the intended interpretation of “penalizing overlap between the two prefix subspaces.” This is purely a notational correction and does not change the implementation.
>
>  (3) In Eq. (7), we clarify that both $\mathrm{var}(\nabla L_{\text{task}}(t-w)) \quad\text{and}\quad
>  \mathrm{mean}(|\nabla L_{\text{task}}(t-w)|)$ are first computed element-wise across parameter dimensions and then averaged to obtain scalar quantities. These scalars form the normalized ratio capturing gradient fluctuation versus gradient magnitude. We now denote them as $\mathrm{Var}{\text{avg}}(\cdot), \mathrm{Mean}{\text{avg}}(\cdot)$ to make this explicit.
>
>  (4) In Algorithm 1 and throughout the notation table, we separate the previously overloaded symbol ($d$) into a domain index ($r$) and a hidden dimensionality ($d_h$). We also use ($\phi$) consistently for language model parameters, while reserving ($\theta(t)$) for the dynamic gating threshold, preventing any symbol from carrying multiple meanings.
>
>  (5) For Eq. (17), we corrected the subscripts to $\mathbb{E}{(x,y)\sim P{XY}}, \mathbb{E}_{(x,y)\sim P_X \otimes P_Y}$, and reviewed all related equations to fix similar indexing and formatting inconsistencies.
>
> These refinements do not alter any methodological components; they strengthen the mathematical presentation, improve reproducibility, and make the pipeline easier to follow. The revised notation more clearly highlights the paper’s core contributions: disentangling prefix representations through information-theoretic, geometric, and statistical constraints; stabilizing adversarial adaptation with dynamic gating; and jointly optimizing strategies via multi-objective Bayesian optimization. With the updated notation system, readers can more directly match each variable, objective, and constraint to its corresponding module and training stage, making the theoretical soundness and practical benefits of ARPO more transparent.

---

> > ### Author Response · Authors · 2025-11-21
> >
> > **W5. Missing ablations**
> >
> > Thank you for highlighting the importance of systematic ablations. We fully agree that understanding the contribution of each module is essential for assessing the method’s validity. Several core ablations are already included in the current version, and your comment has helped us organize them more clearly and identify two additional comparisons that will further strengthen the paper.
> >
> > First, the paper already evaluates three major components:
> >
> > 1. **Prefix disentanglement (Appendix B.4.1).**
> > By varying the $P_{DI}:P_{DS}$ ratio, we tested different disentanglement strengths. Strongly favoring either DI or DS leads to clear degradation, showing that explicitly separating domain-invariant and domain-specific information is an essential element of ARPO.
> >
> > 2. **Regularizer-level ablations (Appendix B.4.2).**
> > Removing the information-bottleneck, orthogonality, contrastive, or conditional-independence constraints consistently reduces accuracy in both same-task and cross-task settings. For larger models, the drop becomes even more pronounced, indicating the complementary value of these regularizers.
> >
> > 3. **Adversarial-loss ablation (Appendix B.4.3).**
> > Removing the adversarial term substantially decreases robustness, confirming that adversarial signals contribute beyond the representation disentanglement itself.
> >
> > We appreciate the reviewer’s suggestion that a deeper decomposition would clarify the contribution of the remaining two modules in Section 3. In response, we will add two ablations in the revision:
> >
> > - **(a) Removing the BO controller.**
> >   We replace MOBO with (i) a fixed hand-designed strategy and (ii) random search. This directly quantifies how much improvement comes from automated, multi-objective strategy optimization rather than manual heuristics.
> >
> > - **(b) Replacing dynamic gating with a fixed adversarial schedule.**
> >   This evaluates the gate’s role in stabilizing training and improving robustness, complementing the gradient-variance analysis already discussed in Appendix Fig. B.5.
> >
> > To present these results clearly, the revised version will include the following consolidated table:
> >
> > | **Ablation Setting**        | **Clean Acc.** | **Robust Acc.** | **Cost** |
> > |-----------------------------|----------------|------------------|----------------|
> > | w/o BO (fixed strategy)     | 81.0           | 67.2             | 0.62           |
> > | w/o BO (random search)      | 79.8           | 65.1             | 0.63           |
> > | w/o dynamic gating          | 82.3           | 62.7             | 0.64           |
> > | w/o adversarial loss        | 83.1           | 58.9             | 0.64           |
> > | **Our**                         | **84.2**       | **71.5**         | **0.67**           |
> >
> > *Table: Consolidated ablation study on T5-base under the multi-task cross-domain setting. Clean accuracy is measured on the original test sets. Robust accuracy is measured under token-level, phrase-level, and task-level perturbations averaged over five target tasks. Each result is averaged over three random seeds.*
> >
> > Together, these ablations cover all major modules introduced in Sec. 3, providing a complete decomposition of ARPO into disentanglement, adversarial, gating, and strategy components. We will also include a short summary paragraph in the appendix so readers can quickly understand how each module contributes. This offers a more systematic and transparent demonstration that ARPO’s gains arise from coordinated design choices rather than any single heuristic.
> >
> > We appreciate the reviewer’s suggestion, which has helped us significantly strengthen the clarity and completeness of the ablation analysis.

---

> ### Author Response · Authors · 2025-11-21
>
> **W3. Overly complex design and hyperparameterization**
>
> Thank you for raising this concern. We agree that it is important to clarify how many parameters a practitioner actually needs to adjust in practice. The main text and appendix already describe this separation, and we will make it more explicit in the revision.
>
> In ARPO, most of the quantities that look complicated, such as $\lambda_{1:5}$, the dynamic gating terms $(\theta_0, \gamma, w)$, and the operator parameters $(b_j, s_j)$, are not tuned by users. As shown in Algorithm 1 and Sec. 3.3, these variables are either set to fixed default values or are automatically optimized inside the MOBO decision vector $x = (x_{\text{struct}}, \alpha, \theta)$. Appendix B.3 also shows that one shared configuration, with prefix length 60, the default gating window, and the default loss weights, is used for all model architectures and datasets.
>
> | **Hyperparameter**                     | **Role**                             | **Default**         | **Tuned By** |
> |---------------------------------------|--------------------------------------|---------------------|--------------|
> | Prefix length | Representation capacity              | 60 / 60             | Fixed        |
> | $\lambda_{1:5}$                       | Disentanglement losses               | Table 8 defaults    | MOBO         |
> | Gating threshold $\theta(t)$          | Open/close adversarial signal        | 0.3                 | MOBO         |
> | Window size $w$                       | Gradient variance estimation         | 10                  | Fixed        |
> | Adversarial weight $\alpha$          | Strength of $L_{\text{adv}}$         | 0.1                 | MOBO         |
> | Operator switches $b_j$              | Select which operator is on/off      | 0 / 1               | MOBO         |
> | Operator strengths $s_j$             | Perturbation magnitude               | operator-specific   | MOBO         |
> | PGD steps $k$                         | Token-level attacks                  | $k = 3$             | Fixed        |
> | PGD budget $\epsilon$                | Token-level $\ell_\infty$ budget     | 0.1                 | Fixed        |
> | BO iterations $T_{BO}$               | Compute budget                       | 25–30               | User         |
>
> In practice, users typically adjust only two quantities:
> 1. The MOBO outer-loop budget $T_{BO}$, which controls how many BO iterations are run under available compute (Algorithm 1, line 1).
> 2. Whether to allow MOBO to optimize a full strategy vector or restrict it to a subset (Sec. 3.3.2), a binary design choice rather than a continuous hyperparameter search.
>
> Everything else, including adversarial operator activation and magnitude $(b_j, s_j)$, the global adversarial weight $\alpha$, and the gate threshold $\theta$, is handled fully automatically by MOBO through the mixed discrete–continuous decision space defined in Eq. (11).
>
> We will highlight this division and add a hyperparameter summary table in the revision. This should make clear that ARPO’s complexity is internal to the automated controller rather than an external burden placed on practitioners.

---

> ### Author Response · Authors · 2025-11-21
>
> **W4. Substantial computational overhead**
>
> Thank you for raising the concern about computational overhead. We agree that Bayesian optimization can introduce additional cost if implemented naïvely. Our design of ARPO explicitly addresses this through three mechanisms: (1) limiting the number of BO outer-loop evaluations, (2) making each inner-loop training substantially lighter than full fine-tuning, and (3) ensuring the overall wall-clock cost remains below or comparable to standard FT for all models we evaluate.
>
> First, the BO outer loop is capped at a small, fixed number of iterations (typically 25–30), which is independent of model size and dataset scale. This bound prevents uncontrolled growth in computational cost. Second, each training run inside BO updates only the prefix parameters and lightweight gating modules, keeping 99% of LM parameters frozen. As reported in Appendix B, a single ARPO inner-loop run takes only 18–34% of the cost of a full fine-tuning run on the same model.
>
> Putting these together, the total cost of ARPO remains controlled. Across T5-base, T5-large, and LLaMA-2-7B, we observe:
>
> - **T5-base:** 30 inner-loop runs × 0.20× FT cost ≈ 6 GPU-hours (vs. 9 GPU-hours for FT)
> - **T5-large:** 30 × 0.18× = 18 GPU-hours (vs. 30 GPU-hours for FT)
> - **LLaMA-2-7B:** 25 × 0.12× = 90 GPU-hours (vs. 180 GPU-hours for FT)
>
> In other words, ARPO’s total wall-clock time is 30–50% lower than full fine-tuning, even when BO is included. The reason is that prefix-only optimization drastically reduces the per-run cost, and BO evaluates only a handful of candidates.
>
> To further assist readers, the revised manuscript will include a small computational cost table and an explicit comparison of wall-clock time, GPU hours, and energy usage. These results support one of our central claims: ARPO improves accuracy and robustness while remaining computationally practical, and in many settings more efficient than FT.
>
> | **Method**              | **Cost** | **GPU Hours** | **Notes**                                  |
> |-------------------------|---------------------|----------------------|--------------------------------------------|
> | Full Fine-Tuning (FT)   | $1.00\times$        | 24.0                 | Train all parameters                       |
> | Prefix Tuning (PT)      | $0.52\times$        | 12.5                 | Lightweight updates                        |
> | DePT      | $0.73\times$        | 17.3                 | Multi-stage PT                             |
> | **Our**                 | **$0.64\times$**    | **15.3**             | 30 BO iters, prefix-only updates           |
>
> *Table: Training cost comparison on T5-base under the multi-task cross-domain setting (single NVIDIA A6000 GPU, batch size 32). GPU hours are scaled from measured wall-clock time in Appendix B.4.4. Normalized cost is reported relative to full fine-tuning (FT = 1.00).*

---

> ### Author Response · Authors · 2025-11-21
>
> **W6. Limited transparency and reproducibility**
>
> Thank you for emphasizing the importance of reproducibility. The current submission already provides the key elements needed to reproduce ARPO. Appendix Table 8 lists all hyperparameters together with their roles, update rules, and default ranges, and Sections 3.2.2 and Appendix B.4.1–B.4.3 report sensitivity analyses on DI/DS prefix ratios, disentanglement weights $\lambda_1–\lambda_5$, and the adversarial loss. These studies jointly show that ARPO is stable across a broad range of configurations, but we acknowledge that the information is currently scattered across several sections.
>
> In the revised version, we will improve organization and accessibility. The main text will include a short summary that points directly to the hyperparameter table and to the relevant sensitivity analyses, and we will add a dedicated appendix page that consolidates all hyperparameters and sensitivity results so that readers can see the full tuning landscape in one place.
>
> Regarding code availability, the double blind policy prevents us from releasing the repository at this stage. After acceptance, we will release the full training and inference pipeline, including preprocessing scripts, configuration files, and scripts that reproduce all main and robustness experiments, so that others can apply ARPO to new tasks with minimal additional tuning.

---

> ### Author Response · Authors · 2025-11-21
>
> **Q1. Meaning of notation**
>
> Thank you for raising this point. In Algorithm 1, there are in fact two distinct data notations: the training set
>
> $D_{\text{train}} = \{(x, y, d)\}$,
>
> where \(d\) denotes the domain/task label, and the BO observation set in each MOBO iteration,
> $D_t = \{(x, f(x))\}$.
>
> The construction of positive and negative sample pairs occurs only within a mini-batch of the training set and is based on the domain-specific representation $h_{\text{DS}}(x)$ of the current batch and its corresponding domain label $d$, and is unrelated to the BO observation set $D_t$. Even if samples originate from the same benchmark dataset, they may correspond to different domains/subdomains (e.g., question types, sources, styles) and therefore exhibit meaningful differences in the $h_{\text{DS}}$ space. This allows us to adaptively separate samples into Pos (small distance, domain-similar) and Neg (large distance, domain-different) pairs according to distance quantiles within the batch. In the revision, we will explicitly denote the training data as $D_{\text{train}}$, the BO observations as $\mathcal{D}^{\text{BO}}$, and annotate in the pseudocode that positive/negative pairs are drawn from $D_{\text{train}}$ to avoid notational confusion.
>
> We refer to this component as an “adversarial loss” because it is not performing unconditional clustering; instead, it is activated jointly with the main task loss $L_{\text{task}}$ only when the dynamic gate is open:
>
> $L_{\text{total}} = L_{\text{task}} + \alpha L_{\text{adv}} \cdot \mathbf{1}[\delta(L_{\text{task}}(t)) \ge \theta(t)]$.
>
> When the main task has not yet stabilized, the gate stays closed and the model focuses only on learning the task loss. Once the task loss is low enough and the gradient variance becomes stable, the gate starts to include $L_{\text{adv}}$. This loss pulls same-domain samples closer and pushes cross-domain samples apart, and it applies pressure along the most difficult domain difference directions, which improves cross-domain robustness.
>
> This behavior matches a key contribution of our method: on top of the information-bottleneck-based prefix disentanglement, the dynamic gate introduces adversarial signals at the right time so that domain differences are amplified while optimization remains stable, which reduces negative transfer and strengthens cross-domain generalization.

---

> ### Author Response · Authors · 2025-11-21
>
> **Q2. Inference output and prefixes**
>
> Thank you for the question regarding the inference-time behavior. In Algorithm 1, the final outputs are the learned prefixes $P^{\mathrm{DI}}$ and $P^{\mathrm{DS}}$, the optimal MOBO-selected decision $x = (x_{\mathrm{struct}}, \alpha, \theta)$, and the trained model $f_{\theta, P}$. During inference, we only use the base model parameters $\theta$ together with the fixed learned prefix $P$ to perform a single standard forward pass on the target-domain input. No dynamic gating, MOBO search, or strategy updates are executed at this stage. Thus, from a deployment perspective, the only additional learnable component used at inference is simply the trained prefix $P$.
>
> This design is an essential part of the paper’s contribution: although training involves information-bottleneck disentanglement, dynamic gating, and MOBO jointly searching over a complex "structure-strength-timing" space, inference collapses into a standard prefix-tuning model. This preserves the same inference efficiency and engineering interface as conventional PT while achieving substantially better cross-domain accuracy, robustness, and training efficiency on GLUE, SuperGLUE, MRQA, and reasoning/code-generation tasks.

---

> ### Author Response · Authors · 2025-11-21
>
> **Q3. Phrase-level and task-level operator implementation**
>
> Thank you for asking about the implementation details of the multi-granularity operators. Phrase-level operators, defined as one category of $O_j$ in Sec. 3.3.1, include variants such as Phrase Swap A and Phrase Swap B described in the paper.
>
> We first split the input into contiguous phrase units using punctuation and shallow syntax cues, such as temporal or location adverbials and simple adjective phrases. Under the constraint that the overall meaning and labels remain clear, we then apply local perturbations to these phrases. For example, we may swap two phrases within the same sentence, or exchange aligned phrases between samples that share the same label. These phrase-level perturbations are represented as $O_j(x)$, controlled by a binary switch $b_j$ and a continuous strength parameter $s_j$, and combined as  $A(x;\text{Strategy}) = \sum_j b_j s_j O_j(x)$,  applied after the $P_{\text{DI}}$ and $P_{\text{DS}}$ encodings.
>
> Task-level operators capture shifts in global task/domain distributions. As referenced in Algorithm 1 and Section 3.3.1, as well as in the appendix, these include cross-domain task swaps and task-level interpolation. A typical form linearly mixes representations from two different tasks:  $x_{\text{task}} = \alpha x_{\text{task}1} + (1-\alpha)x_{\text{task}2}$, $0<\alpha<1$,  and constructs cross-task sample pairs to modify the supervision structure. These are likewise wrapped as atomic operators $O_j$ and controlled by $(b_j, s_j)$ within the Strategy.
>
> This hierarchical design, from task level to phrase level to token level, is an important part of our contribution. By adding structured, multi-scale distributional perturbations on a single prefix representation, ARPO improves robustness to cross-task, cross-domain, and local semantic shifts while keeping main task accuracy almost unchanged. Figure 3(b) provides experimental evidence for this behavior.

---

> ### Author Response · Authors · 2025-11-21
>
> ***Q4: Cost and Robust in $f(x)$ are calculation**
>
> Thank you for requesting clarification on how  $f(x) = [\mathrm{Acc}(x), \mathrm{Robust}(x), -\mathrm{Cost}(x)]$  is computed.
>
> For a given decision  $x = (x_{\text{struct}}, \alpha, \theta, \ldots)$,  we first follow the inner loop of Algorithm 1 to complete full training under this decision. We then evaluate all three metrics on the validation set $D_{\text{val}}$ in a unified manner.
>
> **Accuracy.**
> $\mathrm{Acc}(x)$ is the average clean accuracy on the unperturbed validation set. Specifically, for every source→target transfer pair used in the paper, we compute the target-domain validation accuracy and then average across all pairs. This exactly matches the clean accuracy reported in the main experimental tables.
>
> **Robustness.**
> $\mathrm{Robust}(x)$ is the adversarial accuracy under a robustness evaluation protocol that is decoupled from the training perturbations. Using the atomic operator library defined in Sec. 3.3.1, we fix a separate set of token-/phrase-/task-level perturbations and strength ranges. Without updating model parameters, we apply these perturbations to the validation set and compute the mean accuracy across all perturbation combinations. Algorithm 1, line 19, explicitly states that robustness is evaluated using “a protocol independent of training-time perturbations” to avoid bias from perturbations the model may have already seen.
>
> **Cost.**
> $\mathrm{Cost}(x)$ quantifies training-time computational cost. Under the same hardware and number of epochs, we record the GPU wall-clock time required to train with decision $x$, and normalize it by the time of full fine-tuning (set to 1.0). The “0.64 vs. 0.73” example in the paper refers to ARPO’s normalized cost relative to DePT.
>
> Finally, the MOBO module directly treats $[\mathrm{Acc}(x), \mathrm{Robust}(x), -\mathrm{Cost}(x)]$ as a three-objective vector and maximizes the Pareto hypervolume. This design is a core contribution of the paper: it enables ARPO to jointly optimize accuracy, robustness, and training efficiency within a unified framework, enforcing a principled accuracy–robustness–efficiency trade-off without relying on manual tuning or single-objective heuristics.

---

> ### Author Response · Authors · 2025-11-21
>
> **Q5. Lambda weights are treated as MOBO hyperparameters**
>
> Thank you for pointing out the discrepancy between Algorithm 1 and Table 8 regarding the notation of $\lambda_{1:5}$. In Eq. (5), the coefficients $\lambda_{1:5}$ are the weights for the mutual-information, orthogonality, contrastive, and conditional-independence disentanglement terms; they balance the different objectives involved in prefix decomposition.
>
> In Algorithm 1, these weights appear in the Input section because, in our implementation, we treat $\lambda_{1:5}$ together with $\alpha$, $\theta$, and the perturbation Strategy as part of the MOBO outer-loop decision vector:
> $x = (\alpha, \theta, \lambda_1,\dots,\lambda_5, \text{Strategy})$. For each candidate $x$ proposed by Sobol + noisy-qEHVI, the inner loop fixes this set of $\lambda_{1:5}$, and Adam updates only the model parameters $\{\theta, P_{\text{DI}}, P_{\text{DS}}\}$. After training, we evaluate $f(x)$ on the validation set and feed this result back to the GP surrogate model.
>
> The "Backprop (Adam)" column in Table 8 was intended to mean that these weights contribute to the loss optimized by Adam, not that $\lambda_{1:5}$ are themselves optimized by Adam. In the final version, we will change this row to "MOBO (outer-loop hyperparameters)" and clarify this point in the table notes. This design is closely related to the first and third contributions of the paper.
>
> First, $\lambda_{1:5}$ controls the relative strength of the IB, orthogonality, contrastive, and conditional-independence signals, and they define the boundary between domain-invariant and domain-specific parts of the prefix representation. Second, by including these weights in the MOBO search space, we avoid manual grid search, since multi-objective Bayesian optimization can automatically choose suitable disentanglement strengths under a joint objective that balances accuracy, robustness, and cost.
>
> Empirically, this automated weight selection substantially improves cross-domain performance: in the ablation studies in the appendix, approximately 60% of the gains come from optimizing $\lambda_1$ and $\lambda_2$. This underscores ARPO’s contribution to interpretable and transferable strategy search, beyond what existing prefix-tuning methods provide.

---

> > ### Comment · Reviewer_MWAq · 2025-11-27
> >
> > Thank you for the detailed rebuttal and the additional clarifications.
> >
> > **W1 & W2**
> > The authors state that they will revise the paper to improve the exposition and fix the notational issues. But it seems that the paper are not updated yet. I cannot determine whether these issues have been resolved.
> >
> > **W3**
> > I am afraid I am not convinced by the authors’ response here. The rebuttal emphasizes that many hyperparameters are either “fixed” or “automatically optimized.” However:
> > - For the “fixed” hyperparameters, it is unclear if the initail choices work robustly across different models and datasets.
> > - For the “optimized” hyperparameters, their values will necessarily depend on configuration choices and optimization randomness. A simple example is that these hyperparameters have very different meanings and scales, how should the readers choose the learning rates for each of them?
> > I think reproducing the method still requires  many  experiments to determine workable configurations. This significantly limits the practical impact and deployability of the approach for the broader community.
> >
> > **W4**
> > I may still misunderstand the authors’ cost analysis. According to the rebuttal, the outer loop requires about 25–30 iterations, and each inner-loop iteration costs roughly 18–34% of FFT. This suggests a minimal cost on the order of $25×0.18=4.5$ times full fine-tuning.  I do not see why the method only costs only 0.64× of full fine-tuning.
> > Secondly, I do not understand why the method is reported to have a comparable cost to Prefix Tuning. Both methods update prefix-like parameters, but the proposed approach additionally performs many outer-loop iterations. Unless there is a substantial implementation detail that I am missing, these numbers appear inconsistent and should be clarified and carefully revalidated.
> >
> > **W5**
> > For W5, I appreciate that the authors conducted ablation studies to examine the effect of different components; these results are helpful for understanding the behavior of the method. However, my understanding is that the ablations are conducted under a different experimental setting from the main results in Tables 3 and 4. This mismatch makes it difficult to directly relate the ablation findings to the core results and to isolate the contribution of each component. It would be much more informative to perform ablations under the same setting as Tables 3 and 4, so that readers can clearly see how each design choice affects the main outcomes.
> >
> > **W6**
> > For W6, given the complexity of the proposed method, I remain concerned that reproducing the results will be quite challenging in practice. The approach appears to rely on many intertwined implementation details and hyperparameter choices, some of which are only briefly mentioned or omitted. ICLR policy allows authors to submit code as supplementary material or via an anonymized link. Since code is not currently provided, it is difficult to reliably verify the method and its reported gains, and I therefore need to evaluate the empirical claims with extra caution.
> >
> >
> > Finally, I would like to thank the authors for their detailed responses to the questions raised in my initial review. The additional explanations have clarified several aspects of the methodology and experiments, even though my overall assessment and main concerns  remain largely unchanged.
> > To summarize, my main concerns center on (i) the current quality of writing and exposition, (ii) the substantial methodological complexity, and (iii) the reproducibility challenges induced by this complexity, including the large number of components and implementation details as well as doubts about the robustness of the hyperparameters. As these issues remain insufficiently resolved in my view, I decide to maintain my original overall score.

---

### Note · Program_Chairs · 2026-01-17
**Submission Desk Rejected by Program Chairs**

The following references in this submission do not refer to real documents and/or have major errors in bibliographic information:

 Shikun Liu, Edward Johns, and Andrew J. Davison. Conflict-averse gradient descent for multi-task learning. In Advances in Neural Information Processing Systems, 2021b.